# Spatiotemporal brain hierarchies of auditory memory recognition and predictive coding

L. Bonetti [1,2,3,4] ✉, G. Fernández-Rubio[1], F. Carlomagno [1,5], M. Dietz [6], D. Pantazis [7], P. Vuust[1] & M. L. Kringelbach [1,2,3]

Our brain is constantly extracting, predicting, and recognising key spatiotemporal features of the physical world in order to survive. While neural processing of visuospatial patterns has been extensively studied, the hierarchical brain mechanisms underlying conscious recognition of auditory sequences and the associated prediction errors remain elusive. Using magnetoencephalography (MEG), we describe the brain functioning of 83 participants during recognition of previously memorised musical sequences and systematic variations. The results show feedforward connections originating from auditory cortices, and extending to the hippocampus, anterior cingulate gyrus, and medial cingulate gyrus. Simultaneously, we observe backward connections operating in the opposite direction. Throughout the sequences, the hippocampus and cingulate gyrus maintain the same hierarchical level, except for the final tone, where the cingulate gyrus assumes the top position within the hierarchy. The evoked responses of memorised sequences and variations engage the same hierarchical brain network but systematically differ in terms of temporal dynamics, strength, and polarity. Furthermore, induced-response analysis shows that alpha and beta power is stronger for the variations, while gamma power is enhanced for the memorised sequences. This study expands on the predictive coding theory by providing quantitative evidence of hierarchical brain mechanisms during conscious memory and predictive processing of auditory sequences.

The spatiotemporal features of hierarchical processing in the brain are essential to fully grasp the neural substrates of human perception and cognition, as suggested by the predictive coding theory (PCT)[1–7]. To elucidate such brain mechanisms, much research has focused on the visual system[8,9], which primarily relies on the recognition of patterns arranged in space. Conversely, the auditory system extracts information from patterns and sequences that develop over time[10], providing unique opportunities to understand the temporal hierarchies of the brain.

Extensive research spanning decades has established the hierarchical organisation of auditory perception, with a particular emphasis on the processing of elementary auditory stimuli. This hierarchical progression starts at the peripheral level within the cochlea and moves forward to the auditory pathway, which encompasses the brainstem, pons, trapezoid body, superior olivary complex, lateral lemniscus, inferior and medial geniculate nucleus of the thalamus, and ends in the primary auditory cortex[11,12]. However, how

[1]Center for Music in the Brain, Department of Clinical Medicine, Aarhus University & The Royal Academy of Music, Aarhus/Aalborg, Denmark. [2]Centre for Eudaimonia and Human Flourishing, Linacre College, University of Oxford, Oxford, United Kingdom. [3]Department of Psychiatry, University of Oxford, Oxford, United Kingdom. [4]Department of Psychology, University of Bologna, Bologna, Italy. [5]Department of Education, Psychology, Communication, University of Bari Aldo Moro, Bari, Italy. [6]Center of Functionally Integrative Neuroscience, Department of Clinical Medicine, Aarhus University, Aarhus, Denmark. [7]McGovern Institute for Brain Research, Massachusetts Institute of Technology (MIT), Cambridge, MA 02139, USA. ✉e-mail: leonardo.bonetti@psych.ox.ac.uk

auditory information is integrated from the auditory cortex to the whole brain has not been fully established yet.

Aiming to bridge this gap, prior investigations focused on automatic predictive processes within the framework of PCT. These studies assessed auditory automatic prediction error, often relying on well-established event-related potential/field (ERP/F) components such as N100, mismatch negativity (MMN), and error-related negativity (ERAN)[13–21]. They demonstrated that such components were automatically evoked in response to auditory stimuli, deviations from expected sound features, likelihood of occurrence of musical tones (N100, MMN), and varied harmonic properties (ERAN). Additional studies employed dynamic causal modelling (DCM) to investigate the brain hierarchical architecture during automatic predictive processes, as indexed by MMN. This research has yielded quantitative evidence demonstrating the flow of information from the primary auditory cortex to the superior temporal gyrus and the inferior frontal gyrus[22].

Expanding upon these investigations, Rocchi and colleagues provided evidence of hierarchical organisation in the auditory system using direct electrical brain stimulation. They showed effective connectivity from the auditory cortex to regions in the medial temporal lobe and prefrontal cortex, including the hippocampus and ventrolateral prefrontal cortex[23]. Yet, it remains uncertain how the integration of auditory information across the whole brain relates to complex cognitive functions, such as the conscious recognition and prediction of auditory sequences evolving over time.

To address this question, a unique perspective has arisen through the integration of musical memory paradigms with magnetoencephalography (MEG). Music is a complex artform acquiring meaning through the combination of its constituent elements extended over time[24], while MEG, particularly when used in combination with MRI, allows to track the brain activity with excellent temporal resolution in the order of milliseconds and acceptable spatial accuracy[25]. It is precisely due to these attributes that this combination offers a unique framework for investigating the rapid memory and predictive processes of temporal information in the human brain[7,26].

Indeed, recent studies employing MEG and musical paradigms investigated the brain mechanisms associated with perception, manipulation and recognition of sound sequences. As an example, Albouy and colleagues[27] explored the brain activity underlying memory retention, showing that theta oscillations in the dorsal stream of the participants' brain predicted their auditory WM performance. Along this line, we recently uncovered a vast network of brain areas responsible for encoding and recognising previously memorised and novel musical sounds. This network extended from the auditory cortex to the medial cingulate, inferior temporal cortex, insula, frontal operculum, and hippocampus[28–30]. We also observed that the complexity of music[31] and individual cognitive differences[32] influenced the activity recorded in this network.

In our previous studies, we utilised relatively rapid musical tones with a duration of 250 ms. However, this approach prevented us from closely tracking the neural dynamics associated with each sound within the sequences, leading to frequent overlap with the brain responses to subsequent tones. For this reason, we were also unable to study the rapid hierarchies between the brain regions that we revealed, which is necessary to understand the neurophysiology of conscious recognition of auditory sequences.

Moreover, while our previous works showed differences in brain activity when comparing previously memorised versus completely novel auditory sequences, we did not investigate the scenario in which an original sequence was systematically varied. This prevented us from exploring how the brain generates prediction errors within the framework of a conscious long-term memory recognition task, as opposed to the several instances of automatic prediction errors described in the literature[13–21].

To address these questions, in the present study we made two essential modifications to our previously employed experimental paradigm: (i) we compared the original five-tone melodies with systematic variations occurring at the second, third, fourth, or fifth tone; (ii) we increased the duration of the tones from 250 ms to 350 ms. These seemingly minor modifications were of great importance, as they enabled us to directly address our three main hypotheses, stated as follows. First, the conscious recognition of previously memorised auditory sequences relied on the hierarchical recruitment of a widespread brain network including auditory cortex, hippocampus and cingulate gyrus. We expected to observe that these regions responded to each tone forming the sequence, with the auditory cortex preceding the hippocampus and cingulate gyrus. Second, the detection of the varied auditory sequences elicited a prediction error, originating in the auditory cortex and then propagating to hippocampus, anterior cingulate and ventromedial prefrontal cortex. While we expected a consistent response in the auditory cortex to the introduction of varied sounds at any point in the sequence, we hypothesised that the anterior cingulate gyrus and ventromedial prefrontal cortex exhibited their strongest response exclusively when the variation in the sequence was introduced. Third, sequence recognition relied on a brain hierarchy characterised by feedforward connections from auditory cortices to hippocampus and cingulate gyrus, simultaneous with feedback connections in the opposite direction. Moreover, recognition of both previously memorised and varied sequences relied on the same hierarchy, while the temporal dynamics, strength and polarity of brain responses differed.

These hypotheses were confirmed by the results of the experiment, which showed that the brain distinctively responds to each tone forming the previously memorised sequence, with the auditory cortex preceding hippocampus and anterior and medial cingulate gyrus. In addition, the detection of the sequence variations elicits a conscious prediction error in the brain, originating in the auditory cortex and propagating to hippocampus, anterior and medial cingulate. While the auditory cortex has a constant response to the varied sounds introduced at any point in the sequence, the hippocampus and anterior and medial cingulate gyrus exhibit their strongest response exclusively to the sound which introduced the variation in the sequence. Finally, we provide quantitative evidence of the brain functional hierarchies during long-term auditory recognition. Here, as hypothesised, feedforward connections originate from the auditory cortices and extend to the hippocampus, anterior cingulate gyrus, and medial cingulate gyrus, along with feedback connections in the opposite direction. Notably, throughout the sequence, the hippocampus and cingulate gyrus maintain the same hierarchical level, except for the final tone, where the cingulate gyrus assumes the top position within the hierarchy.

## Results
### Experimental design and behavioural results
Eighty-three participants performed an old/new auditory recognition task during MEG recordings. After learning a short musical piece (Supplementary Fig. S1), participants were presented with 135 five-tone musical sequences lasting 1750 ms each and were requested to state whether each sequence belonged to the original music ('memorised' sequence (M), old) or it was a varied sequence ('novel' sequence (N), new) (Fig. 1a).

Twenty-seven sequences were extracted from the original musical piece and 108 were variations of the original melodies. We organised these variations in four categories depending on whether changes involved every musical tone of the sequence after the first (NT1), second (NT2), third (NT3) or fourth (NT4) tone (Fig. 1b and Supplementary Fig. S2).

We performed statistical analyses on the MEG task behavioural data (see Table 1 for descriptive statistics). We computed two

independent Kruskal-Wallis H tests to assess whether the five categories of temporal sequences (M, NT1, NT2, NT3, and NT4) differed in terms of response accuracy and reaction times (Supplementary Fig. S3).

The Kruskal-Wallis H test for response accuracy was significant ($H(4) = 36.38$, $p < 0.001$), indicating a difference between categories in

the number of correct responses. The Tukey–Kramer correction for multiple comparisons highlighted that NT4 trials were correctly identified with a lower frequency than M ($p = 0.001$), NT1 ($p = 0.001$), NT2 ($p = 0.0003$), and NT3 trials ($p = 0.0001$).

The Kruskal-Wallis H test for the reaction times was also significant ($H(4) = 22.53$, $p = 0.0002$). The Tukey-Kramer correction for

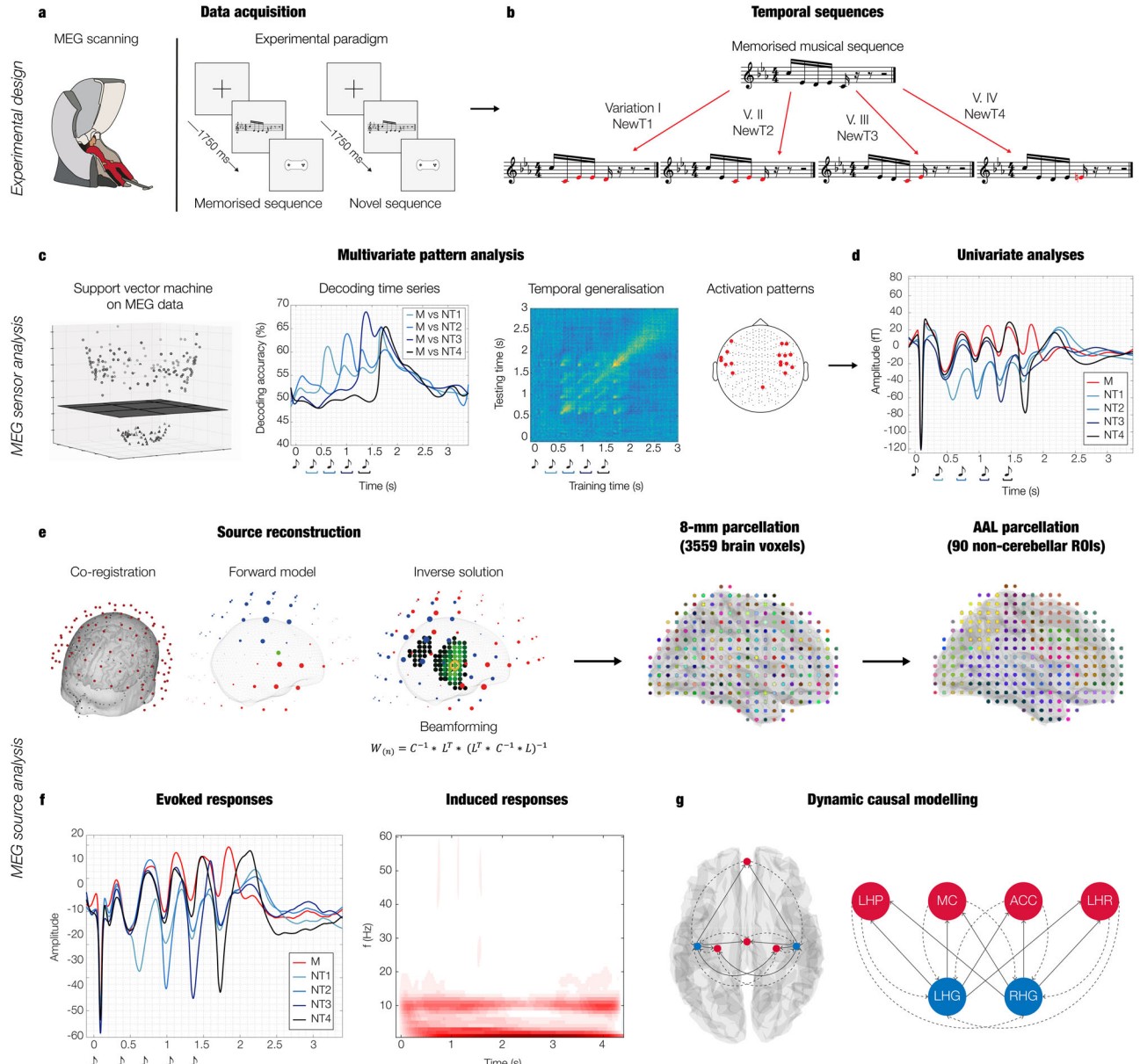

**Fig. 1 | Experimental design, stimuli, and analysis pipeline. a** The brain activity of 83 participants was collected using magnetoencephalography (MEG) while they performed an auditory old/new recognition task. One at a time, five-tone auditory sequences were presented in randomised order and participants were instructed to respond with button presses whether they were old (memorised musical sequences, M) or new (novel musical sequences, N). **b** Five types of auditory sequences (M, NT1, NT2, NT3, NT4) were used in the study (see Supplementary Fig. S2 for the full set of sequences). The N sequences were created through systematic variations (V.) of the M sequences. This procedure consisted of changing every musical tone of the sequence after the first (NT1), second (NT2), third (NT3) or fourth (NT4) tone, as illustrated by the red musical tones. **c** After MEG data pre-processing, multivariate pattern analysis was used to assess whether it was possible to discriminate the experimental conditions based on the neural activity recorded with MEG. **d** Univariate analyses on the MEG channels that primarily contributed to the decoding algorithm were performed using t-tests, independently for each time point, to compare the M condition with each of the N conditions (i.e. M versus NT1, M versus NT2, etc.). **e** The MEG data was co-registered with the individual anatomical magnetic resonance imaging (MRI) data, and source reconstructed using a beamforming algorithm. This resulted in one time series for either the 3559 reconstructed brain sources (8-mm parcellation) or for the 90 non-cerebellar automated anatomical labelling (AAL) regions of interest (ROIs). The reconstructed brain activity was contrasted across experimental conditions (M versus each category of N). **f** Evoked and induced responses were computed for the AAL ROIs and contrasted across experimental conditions. **g** Dynamic causal modelling was applied to a restricted set of AAL ROIs: left Heschl's gyrus (LHG), right Heschl's gyrus (RHG), left hippocampus (LHP), right hippocampus (RHP), anterior cingulate gyrus (ACC), and medial cingulate gyrus (MC), to assess the functional hierarchical organisation of the brain during the recognition of auditory sequences.

**Table 1 | MEG task behavioural results**

| Behavioural variables | M | NT1 | NT2 | NT3 | NT4 |
|---|---|---|---|---|---|
| Correct recognition | 22.33 ± 5.30 | 22.36 ± 4.27 | 21.58 ± 5.31 | 21.66 ± 5.34 | 17.04 ± 7.12 |
| Reaction times (ms) | 2426 ± 226 | 2407 ± 284 | 2431 ± 282 | 2415 ± 272 | 2578 ± 259 |

Mean and standard deviations across participants of number of correctly recognised trials and reaction times (ms) for the five experimental conditions (previously memorised [M], novel T1 [NT1], novel T2 [NT2], novel T3 [NT3], novel T4 [NT4]). Participants were presented with 27 M, 27 NT1, 27 NT2, 27 NT3, and 27 NT4 trials.

multiple comparisons highlighted that NT4 trials were correctly identified with greater reaction times than M ($p = 0.0016$), NT1 ($p = 0.0013$), NT2 ($p = 0.0054$), and NT3 trials ($p = 0.0008$).

## Multivariate pattern analysis on the MEG channels

Using a support vector machine (SVM) classifier (see details in Methods, and Fig. 1c), we performed multivariate pattern analysis to decode different neural activity associated with the recognition of M versus N. Specifically, we computed four independent analyses, decoding M versus each of the four categories of novel sequences (i.e., M versus NT1, M versus NT2, M versus NT3, and M versus NT4).

As shown in Fig. 2a and Supplementary Fig. S4, each of these analyses returned a decoding time series showing how the neural activity differentiated the pair of experimental conditions. Overall, the results showed that the SVM was able to detect significant differences between M and N sequences. As illustrated in Fig. 2a, decoding M versus NT1 returned the following main significant time windows: 0.53–0.73 s; 0.91–0.95 sec; 1.27–1.30 s; 1.62–1.88 s ($p < 0.012$, false-discovery rate [FDR]-corrected). Decoding M versus NT2 gave rise to the following main significant time windows: 0.89–1.18 s; 1.26–1.42 s; 1.54–1.89 s ($p < 0.012$, FDR-corrected). Decoding M versus NT3 returned one main significant time window: 1.25–2.07 s ($p < 0.012$, FDR-corrected). Finally, decoding M versus NT4 showed the following main significant time window: 1.64–2.07 ($p < 0.012$, FDR-corrected). The contribution of each MEG channel to the decoding algorithm in the significant time windows is depicted in Fig. 2b. Detailed statistical results are reported in Supplementary Data 1 and illustrated in Supplementary Fig. S4.

To evaluate the persistence of discriminable neural information over time, we used a temporal generalisation approach by training the SVM classifier at a given time point $t$ and testing it across all time points. This was calculated for the four pairs of experimental conditions described above. The signed-rank permutation test against chance level and cluster-based Monte-Carlo simulations[29,30,32,33] (MCS; $\alpha = 0.05$, MCS $p$-value = 0.001) showed that the performance of the classifier was significantly above chance even a few hundreds of milliseconds beyond the diagonal, for all pairs of conditions. Notably, the neural difference between M and N was comparable across diverse musical tones, as shown by the recurrent patterns depicted in Fig. 2c and highlighted by the red graphs. Detailed statistical results are reported in Supplementary Data 2.

## Univariate analysis on the MEG channels

Pairwise decoding is a powerful technique, but it can only tell if two conditions are characterised by significantly different brain activity and does not provide information about which condition presents the strongest brain activity. To answer this question, we computed an additional analysis focusing on the MEG channels showing the strongest activation patterns (one standard deviation plus the mean) in the decoding algorithm (Fig. 1d). This was done independently for magnetometers and gradiometers and returned 16 magnetometers and 32 gradiometers (see Methods for the channels number).

These MEG channels were then averaged based on the channel type (magnetometers or gradiometers) and the polarity of the signal (positive or negative orientation of the N100 to the first

sound of the sequence). Using this procedure, four distinct groups were created: (i) magnetometers negative N100 (Fig. 2d, left), (ii) magnetometers positive N100 (Fig. 2d, right), (iii) gradiometers negative N100 (Fig. 2e, left), (iv) gradiometers positive N100 (Fig. 2e, right) (further details are provided in the Methods section).

Then, independently for each group, we computed one two-sided t-test for each time point and each combination of M versus Ns (i.e., M versus NT1, M versus NT2, M versus NT3, M versus NT4). We corrected for multiple comparisons using a one-dimensional (1D) cluster-based MCS (MCS $\alpha = 0.05$, MCS $p$-value = 0.001). As reported in detail in Supplementary Data 3, the results showed significant differences between experimental conditions which aligned closely in timing with the decoding analyses reported in the previous section, indicating consistency between the two methods.

To provide full disclosure of our data, we complemented these computations with the same univariate analysis calculated independently for each MEG channel. As shown in Supplementary Fig. S5a–z2 and Supplementary Data 4, the significant differences between experimental conditions were coherent and spread across a large array of MEG channels.

## Neural sources of the MEG channels peak activity

First, we computed source reconstruction by employing a single-shell forward model and a beamforming approach as inverse solution in an 8-mm grid (corresponding to 3559 brain voxels), using magnetometers only (Fig. 1e). This procedure returned a time series for each of the reconstructed bran voxel, which is commonly referred to as neural activity index (see Methods for extensive details on this standard procedure).

Second, we extracted the time indices of the minimum and maximum value recorded by the magnetometers shown in Figs. 2d and 3 in a 400 ms time window following the onset of each tone. The duration of this time window reflects the period during which independent responses to each sound occurred. This procedure returned the time index of two neural peaks for each experimental condition and each tone of the sequence (with exclusion of the 1st one which did not differ across conditions). This revealed that one peak of the neural activity occurred around 350 ms after the onset of each tone (highlighted in yellow in Fig. 3a, b). With regards to this peak, condition M showed the strongest response among all conditions. The other peak occurred in response to the tone which introduced the variation in the musical sequence (highlighted in purple in Fig. 3a, b). In this case, the N condition which introduced the variation reported the strongest response among all other conditions. This occurred approximately at 250 ms after the varied tone (i.e. NT1 showed the strongest response to tone two among all conditions, NT2 to tone three, NT3 to tone four and NT4 to tone five). Extensive details on these neural activity peaks are reported in Table 2.

Third, for each of the defined peaks, we computed contrasts in MEG source space between M and N. More specifically, with regards to the peaks occurring approximately at 350 ms after each tone, we contrasted M versus NT1 (illustrated in the yellow boxes in Fig. 3). This was done because the peaks occurring 350 ms after each tone were always greater for M, and NT1 was the only condition comprising sounds that were always different from M. Similarly, with regards to

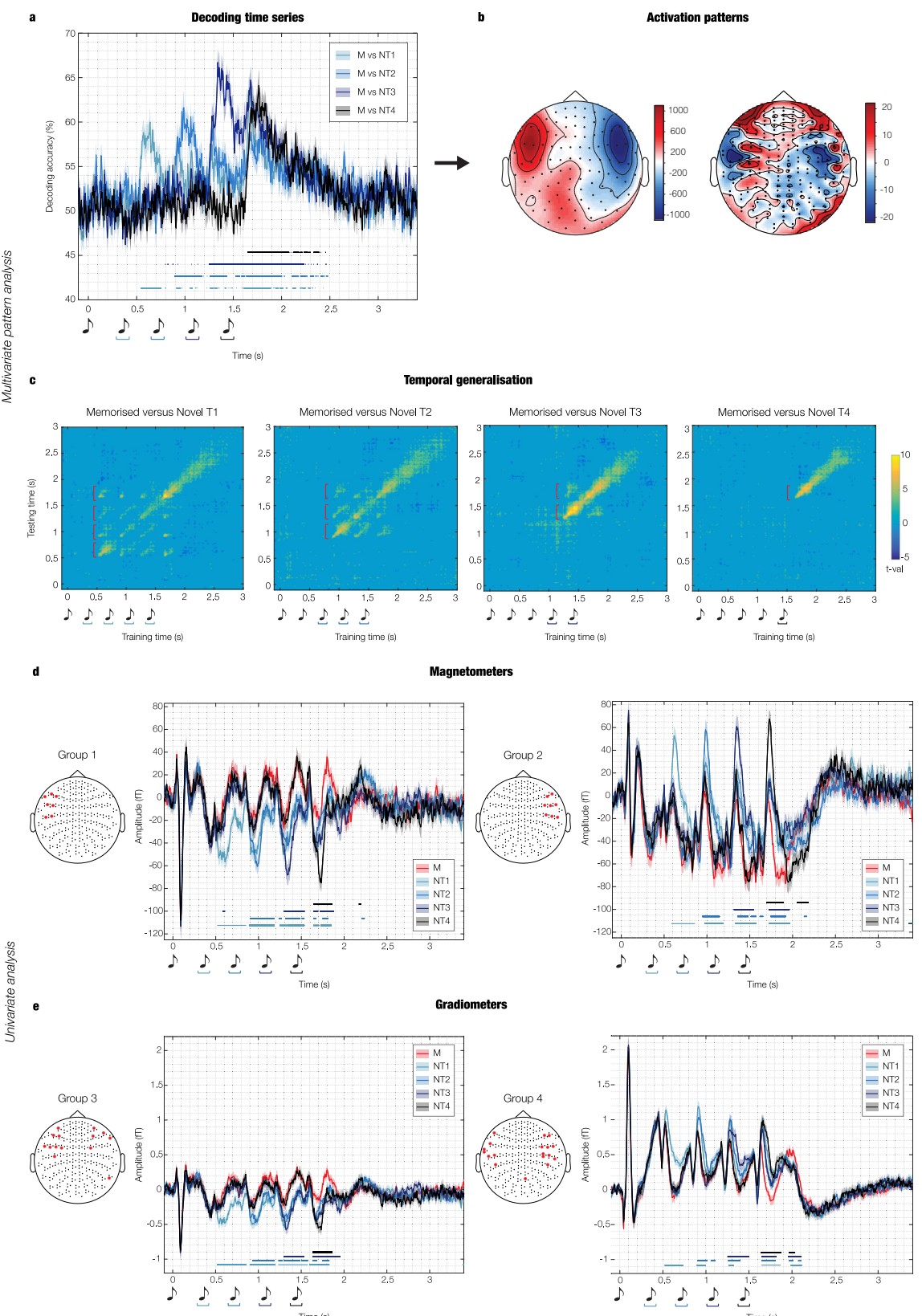

the peaks occurring approximately at 250 ms after the onset of the varied tones, we contrasted the N condition which showed the strongest peak against M (illustrated in the purple boxes in Fig. 3). Here, for each condition in the pair, we averaged the absolute value of the activity of each of the 3559 reconstructed brain voxels in the ±20 ms time window around the correspondent time index. Then, we computed two-sided t-tests for each of the 3559 brain voxels by contrasting the pairs of conditions and FDR-corrected for multiple comparisons. The results of these analyses are shown in Fig. 3 and reported in detail in Table 2 and Supplementary Data 5. In addition, we reported the neural activity recorded by the MEG channels in topographic maps in Supplementary Fig. S6.

**Fig. 2 | Multivariate pattern analysis and cluster-based univariate analysis on MEG channels. a** Multivariate pattern analysis decoding the different neural activity associated with memorised versus novel musical sequences. The plot shows the decoding time series (50% chance level) for four rounds of pairwise decoding (memorised [M] versus novel T1 [NT1], M versus novel T2 [NT2], M versus novel T3 [NT3], and M versus novel T4 [NT4]). The sketch of the musical tones represents the onset of the sounds forming the auditory sequences. The blue-black lines indicate the significant time points when the algorithm successfully decoded the experimental conditions (signed-rank permutation test against chance level and false-discovery rate [FDR] correction for multiple comparisons resulting in adjusted $p < 0.012$). **b** Activation patterns estimated by the decoding weights (without unit of measure). They represent the relative contribution of the MEG channels to the decoding algorithm when it successfully distinguished the two experimental conditions from the neural data (as indicated in **a**). **c** Temporal generalisation of pairwise decoding results, tested against a 50% chance level. Plots display the significant time points (either yellow or dark blue) where conditions

were successfully distinguished after cluster-based Monte-Carlo simulations [MCS] correction for multiple comparisons. **d** Univariate analysis computed on the magnetometers which primarily contributed (one standard deviation plus the mean over magnetometers) to the decoding algorithm. After averaging the magnetometers channels in two independent groups (see Methods for details), two-sided t-tests were conducted for each time point and M versus NTs pair (i.e., M versus NT1, M versus NT2, M versus NT3, M versus NT4), and corrected for multiple comparisons using cluster-based MCS (MCS, α = 0.05, MCS p-value = 0.001). Blue-black lines highlight the temporal extent of the significant differences between M and N conditions. Different hues of blue-black show specific M versus N condition comparisons. **e** Same procedure observed for **d** was performed on gradiometers. All the time series illustrated in this figure correspond to averages over participants (n = 83), while shaded areas indicate standard errors. For all plots, musical tone sketches indicate sounds sequence onset. Source data are provided as a Source Data file.

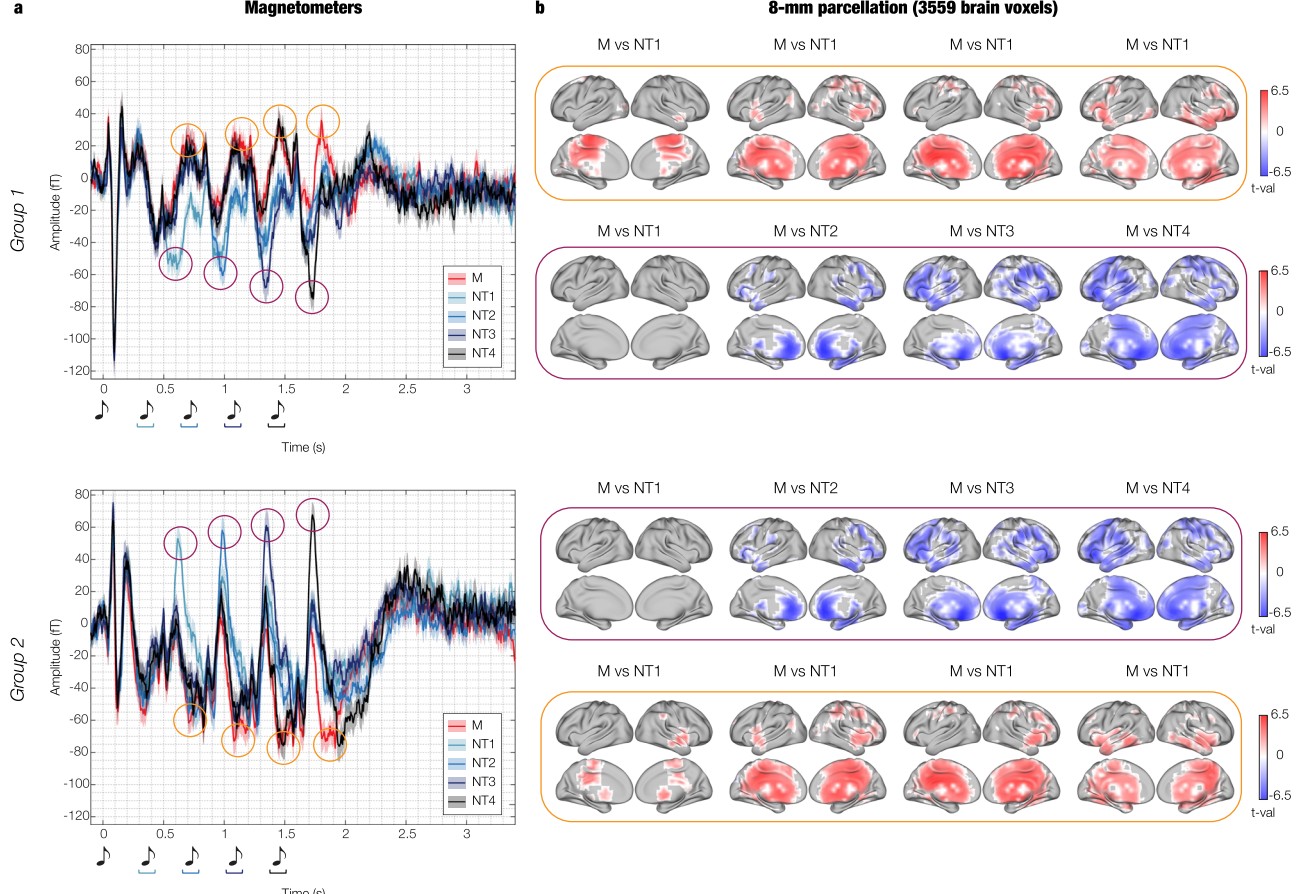

**Fig. 3 | Neural activity peaks recorded by magnetometers and contrasts in magnetoencephalography (MEG) 8-mm source space. a** Exactly as illustrated in Fig. 2, the waveform plot displays the time series averaged over participants (n = 83) for the following groups of MEG channels: (i) magnetometers negative N100, and (ii) magnetometers positive N100. Shaded areas indicate the standard errors. In addition, here we have marked the maximum and minimum peaks of neural activity following the onset of each tone (excluding the first tone). These peaks occurred approximately at 250 ms (purple circle) and 350 ms (yellow circle) after each tone's onset. **b** Within a time window of ±20 ms around each peak, we averaged the brain activity across all 3559 reconstructed brain voxels for each condition. In the yellow

boxes, we represent the significant brain voxels obtained by contrasting the activity associated with memorised (M) versus novel T1 (NT1). These contrasts were done by using two-sided t-tests and false discovery rate (FDR) to correct for multiple comparisons. Purple boxes depict the same analysis, but for comparisons involving M versus NT1 (tone 2), M versus novel T2 (NT2, tone 3), M versus novel T3 (NT3, tone 4), and M versus novel T4 (NT4, tone 5). In both cases, the red colour shows the brain voxels (t-values) characterised by a stronger activity for M versus NTs, while the blue colour indicates the opposite pattern. For a detailed statistical report, consult Supplementary Data 5. Source data are provided as a Source Data file.

Furthermore, the brain activity recorded by the magnetometers for each peak and hemisphere was correlated with the measure of participants' musical training provided by the Goldsmith Musical Sophistication Index (GOLD-MSI)[34] and corrected for multiple comparisons using FDR (Table 2).

## Automated anatomical labelling (AAL) time series

After evaluating the neural sources of the different brain activity associated with the experimental conditions using the fine-grained 8-mm parcellation of the brain (Fig. 1e, 8-mm parcellation), we focused on a set of anatomically defined regions of interest (ROIs) (Fig. 1e, AAL

**Table 2 | Neural activity peaks recorded by magnetometers, musical expertise and contrasts in MEG 8-mm source space**

| Tone # | Tone 2 | | Tone 3 | | Tone 4 | | Tone 5 | |
|---|---|---|---|---|---|---|---|---|
| **Peak** | **Max** | **Min** | **Max** | **Min** | **Max** | **Min** | **Max** | **Min** |
| Magnetometers in the left hemisphere | | | | | | | | |
| Time (s) | 0.708 (M) | 0.612 (NT1) | 1.092 (M) | 0.992 (NT2) | 1.452 (M) | 1.340 (NT3) | 1.804 (M) | 1.736 (NT4) |
| Musical expertise (FDR-corrected) | $r = 0.093$ $p = 0.473$ | $r = -0.218$ $p = 0.084$ | $r = 0.239$ $p = 0.066$ | $r = -0.186$ $p = 0.132$ | $r = 0.286$ $p = 0.040$ | $r = -0.261$ $p = 0.050$ | $r = 0.344$ $p = 0.014$ | $r = -0.034$ $p = 0.768$ |
| *Contrasts in MEG source space* | | | | | | | | |
| Conditions | M versus NT1 | M versus NT1 | M versus NT1 | M versus NT2 | M versus NT1 | M versus NT3 | M versus NT1 | M versus NT4 |
| Max t-val | 5.79 | n.s | 5.45 | −6.09 | 5.81 | −6.56 | 5.1 | −6.03 |
| FDR-adjusted $p$-value | 0.003 | n.s | 0.009 | 0.008 | 0.01 | 0.01 | 0.01 | 0.02 |
| MNI coordinates | −6 −22 64 | n.s | −6 −30 40 | 2 34 0 | −6 −38 32 | 2 18 −16 | 26 −30 −8 | −6 26 −16 |
| AAL region | Supp Motor Ar | n.s | Cingulum Mid | Cingulum Ant | Cingulum Mid | Subgenual | Hippocampus | Front Med Orb |
| Magnetometers in the right hemisphere | | | | | | | | |
| Time (s) | 0.620 (NT1) | 0.832 (M) | 0.988 (NT2) | 1.088 (M) | 1.352 (NT3) | 1.456 (M) | 1.732 (NT4) | 1.920 (M) |
| Musical expertise (FDR-corrected) | $r = 0.242$ $p = 0.061$ | $r = 0.051$ $p = 0.819$ | $r = 0.428$ $p < 0.001$ | $r = -0.034$ $p = 0.819$ | $r = 0.459$ $p < 0.001$ | $r = -0.050$ $p = 0.819$ | $r = 0.257$ $p = 0.050$ | $r = -0.026$ $p = 0.819$ |
| *Contrasts in MEG source space* | | | | | | | | |
| Conditions | M versus NT1 | M versus NT1 | M versus NT2 | M versus NT1 | M versus NT3 | M versus NT1 | M versus NT4 | M versus NT1 |
| Max t-val | n.s | 4.41 | −6.14 | 5.50 | −7.13 | 5.61 | −6.17 | 5.24 |
| FDR-adjusted $p$-value | n.s | 0.001 | 0.008 | 0.01 | 0.02 | 0.01 | 0.02 | 0.01 |
| MNI coordinates | n.s | 58 −14 −16 | 2 34 −8 | −6 −30 40 | 2 18 −16 | −6 −38 32 | −6 26 −16 | 10 −46 48 |
| AAL region | n.s | Temporal Mid | Cingulum Ant | Cingulum Mid | Subgenual | Cingulum Mid | Front Med Orb | Precuneus |

The table shows the time index of the maximum and minimum peaks recorded by the magnetometers (see Fig. 3) for each tone (excluding the first one). Additionally, the table indicates the condition for which the strongest peaks were observed (in parenthesis). The table also illustrates the correlations between the absolute value of the neural peaks and the participants' musical training (r = Pearson's correlation coefficient, *p*-values are adjusted by the false discovery rate [FDR] correction for multiple comparisons). Finally, the table provides details of the MEG source contrasts (two-sided t-tests) in 8-mm space. This includes the conditions which were contrasted, t-value, FDR-adjusted *p*-value, Montreal Neurological Institute (MNI) coordinates, and automated anatomical labelling (AAL) region of the brain voxel which showed the strongest difference between the experimental conditions.

parcellation). Here, we used a standard anatomical parcellation method known as automated anatomical labelling (AAL)[35] and calculated a time series for each of the 90 non-cerebellar ROIs of AAL (Fig. 1f, left, Evoked responses).

As specified above, the source reconstruction was conducted in an 8-mm space, resulting in 3559 brain voxels, each with its own time series. Afterwards, for each of the 90 ROIs, we identified the corresponding brain voxels and then averaged the time series of these associated voxels. This allowed us to obtain a final time series for each AAL ROI.

Then, we computed one two-sided t-test for each AAL ROI, each time point and each combination of M versus Ns, and we corrected for multiple comparisons using 1D cluster-based MCS (MCS, α = 0.05, MCS *p*-value = 0.001). Confirming the results from the 8-mm brain parcellation shown in Fig. 3, this analysis returned several significant clusters of differential brain activity over time between M and Ns. These were primarily localised in auditory cortices (e.g. Heschl's gyrus, superior temporal gyrus), hippocampus and para-hippocampal gyrus, inferior temporal cortex, anterior, medial and posterior cingulate gyrus, and ventromedial prefrontal cortex. Supplementary Fig. S7a–g and Supplementary Data 6 and 7 show all AAL ROIs and the significant differences between conditions. Figure 4 and Supplementary Fig. S8 show instead a selected array of ROIs which were particularly relevant for this study. Here, we identified the two ROIs that showed the maximum activity in absolute value among auditory regions (i), medial temporal lobe (ii) and cingulate and prefrontal cortices (iii). We selected these broad brain regions based on the cognitive processes involved in the experimental task used in this study: audition (i), memory (ii), evaluation and decision-making (iii). As shown by Fig. 4 and Supplementary Fig. S8, the selected AAL ROIs were left and right Heschl's gyrus (LHG, RHG), left and right hippocampus (LHP, RHP), anterior cingulate gyrus (ACC, left and right averaged together) and medial cingulate gyrus (MC, left and right averaged together), respectively.

In essence, the contrasts showed that M versus N was characterised by stronger activity in RHG, LHP, RHP and ACC at 350–450 ms after the onset of each tone. Similarly, M presented stronger negative activity than N in the MC at 400–500 ms after the onset of each tone (Fig. 4). Conversely, late N100 responses localised in LHG were stronger for N versus M. Moreover, LHP, RHP and ACC showed a stronger response for N versus M occurring at 250–300 ms after altering the original sequences (Fig. 4). Table 3 reports detailed statistics of the largest significant clusters, while complete statistical results of these six ROIs are described in Supplementary Data 7.

In addition, we applied the multivariate source leakage correction method proposed by Colclough and colleagues[36] (see Methods for details) to the ROIs time series and computed the same statistical analysis described above. As shown in Supplementary Fig. S9 and Supplementary Data 8, the results aligned with our initial findings, thereby corroborating their robustness.

Finally, as a further reliability measure, we reported in supplementary materials the description of an alternative, functional parcellation derived from the data (Supplementary Fig. S10 and Supplementary Data 9). As shown by Supplementary Figs. S7a–g and S11 and Supplementary Data 9, the contrasts across experimental conditions when using the functional parcellation returned results which were highly comparable with AAL, demonstrating that the significance of our findings did not depend on the chosen parcellation. All the subsequent analyses reported in the manuscript were computed on the AAL parcellation. Moreover, we complemented them by performing the same analyses using the functional parcellation, as reported in the supplementary materials (Supplementary Figs. S15–S18, Supplementary Table S1 and Supplementary Data 13, 14).

**Prediction error across brain regions of interest**
After differentiating the brain activity of M versus N, we conducted an additional analysis to investigate the peak responses elicited by the

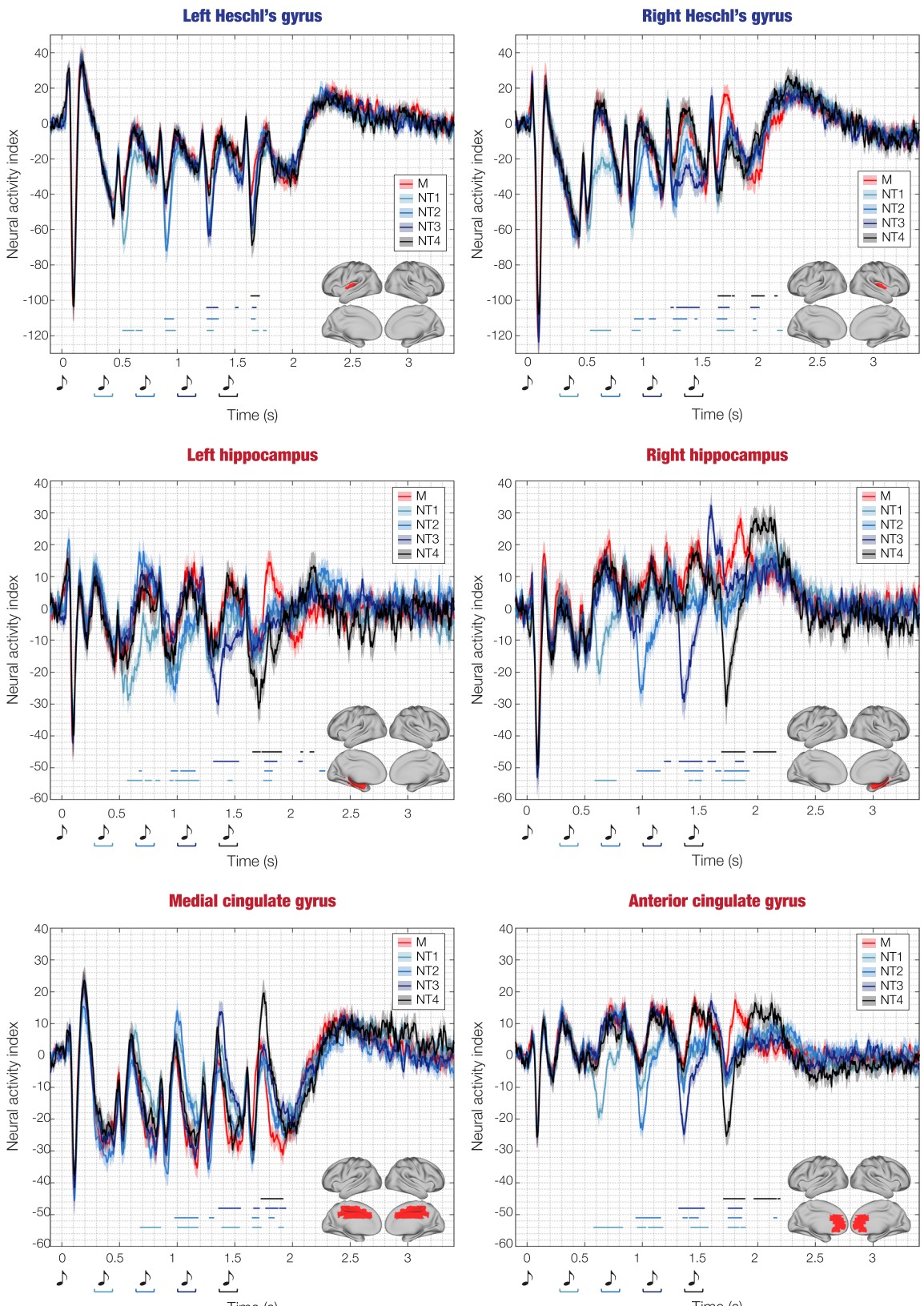

varied tones within the N conditions. Regarding Heschl's gyrus (left and right), the peaks corresponded to the late N100, occurring approximately at 150 ms after the onset of each varied tone. Regarding the hippocampus (left and right), anterior and medial cingulate gyrus, the peaks occurred approximately at 250 ms after the onset of each varied tone (these responses were negative for the hippocampus and anterior cingulate gyrus and positive for the medial cingulate gyrus). We computed the average neural activity within a ±20 ms time window centred around the designated time index (depicted in Fig. 5 by red circles). Subsequently, we performed separate one-sided analyses of variance (ANOVAs) for each ROI and selected condition, with the sequential neural peaks as independent variable and the neural activity

**Fig. 4 | Source-localised differences in evoked responses across experimental conditions.** This figure illustrates the source-localised brain activity averaged over participants ($n = 83$) for each experimental condition (memorised [M], novel T1 [NT1], novel T2 [NT2], novel T3 [NT3], novel T4 [NT4]) within six selected automated anatomical labelling (AAL) regions of interest (ROIs): left Heschl's gyrus (LHG), right Heschl's gyrus (RHG), left hippocampus (LHP), right hippocampus (RHP), anterior cingulate gyrus (ACC), and medial cingulate gyrus (MC). The shaded areas indicate standard errors. Musical tone sketches indicate the onset of the sounds forming the sequences. Brain templates depict the spatial extent of the selected ROIs. Blue-black lines highlight the temporal extent of the significant

differences between M versus each category of Ns (i.e. M versus NT1, M versus NT2, M versus NT3, M versus NT4). Here, the contrasts were computed using two-sided t-tests, independently for each time point, and correcting for multiple comparisons using one-dimensional Monte-Carlo simulations (MCS; MCS, $\alpha = 0.05$, MCS p-value = 0.001) Different hues of blue-black correspond to specific M versus N condition comparisons. For a comprehensive depiction of all AAL ROIs, please refer to Supplementary Fig. S7a–g. For a detailed statistical report on significant differences between experimental conditions in all ROIs, consult Supplementary Data 6. Source data are provided as a Source Data file.

as the dependent variable. Since this approach led to a total of 18 ANOVAs (six ROIs for NT1, NT2, NT3), we applied FDR correction to account for multiple comparisons. Moreover, the Tukey-Kramer correction was applied to the post-hoc analysis computed for each ANOVA (Supplementary Data 10). Results revealed that the responses in Heschl's gyri to subsequent varied tones were not significantly different: $LHG_{NT1}$: $F(3, 331) = 0.87$, $p = 0.51$; $LHG_{NT2}$: $F(2, 248) = 1.33$, $p = 0.37$; $LHG_{NT3}$: $F(1, 165) = 0.68$, $p = 0.51$; $RHG_{NT1}$: $F(3, 331) = 0.90$, $p = 0.51$; $RHG_{NT2}$: $F(2, 248) = 0.18$, $p = 0.83$; $RHG_{NT3}$: $F(1, 165) = 0.33$, $p = 0.60$. In contrast, for the hippocampus and cingulate gyrus, the responses to different varied tones were significantly different, highlighting the stronger activity following the initial tone which

introduced the sequence variation: $LHP_{NT1}$: $F(3, 331) = 2.82$, $p = 0.05$; $LHP_{NT2}$: $F(2, 248) = 7.29$, $p = 0.002$; $LHP_{NT3}$: $F(1, 165) = 7.86$, $p = 0.01$; $RHP_{NT1}$: $F(3, 331) = 7.40$, $p = 2.9e{-}04$; $RHP_{NT2}$: $F(2, 248) = 14.33$, $p = 1.3e{-}06$; $RHP_{NT3}$: $F(1, 165) = 27.16$, $p = 5.56e{-}07$; $ACC_{NT1}$: $F(3, 331) = 4.98$, $p = 0.005$; $ACC_{NT2}$: $F(2, 248) = 11.54$, $p = 1.6e{-}05$; $ACC_{NT3}$: $F(1, 165) = 17.55$, $p = 2.05e{-}04$; $MC_{NT1}$: $F(3, 331) = 3.21$, $p = 0.02$; $MC_{NT2}$: $F(2, 248) = 8.40$, $p = 2.9e{-}04$; $MC_{NT3}$: $F(1, 165) = 6.24$, $p = 0.02$.

## Dynamic causal modelling

In this study, DCM was used to test our hypothesised model of brain hierarchies during recognition of memorised and varied musical sequences against five competing models, which are described in detail in the Methods section and illustrated in Figs. 1g and 6. These models comprised the six AAL ROIs described in the previous section and depicted in Supplementary Fig. S8. Our hypothesised model consisted of feedforward connections from LHG and RHG to LHP, RHP, ACC, and MC, and feedback connections from LHP, RHP, ACC and MC to LHG and RHG (Fig. 6a, red box).

Since DCM for MEG is usually computed on relatively short time windows (e.g. 350–400 ms) and for one experimental perturbation (i.e. sound) at a time[37,38], we computed a series of DCM analyses independently for each tone of the sequences, in a 350 ms time window from each tone onset. Specifically, for each participant, we computed an independent DCM for tones two, three, four and five for condition M. In relation to the N conditions, we were interested in the tone that introduced the variation in the sequences. Thus, we computed four DCMs, organised as follows: tone two for NT1; tone three for NT2; tone four for NT3; tone five for NT4.

To compare alternative models, we estimated the free energy $F$ associated with the computation of each model (model evidence). To study the consistency of $F$ across the population, we used a random-effects Bayesian model selection (RFX BMS) procedure, which provides the posterior probability and protected exceedance probability of each model, indicating which of the alternative models has the strongest model evidence within the whole population. In addition, we computed the Bayesian omnibus risk (BOR), which represents the probability that the protected exceedance probability associated with each model comparison is attributable to chance within the sample of participants (Fig. 6b, c)[37]. This procedure was observed independently for the eight DCM analyses described above (four for M and four for Ns).

Results indicated that our hypothesised model was the one with the highest evidence for all the DCM analyses of tones two, three and four (Fig. 6). This result showed a high probability of not being driven by chance, as demonstrated by the BOR parameter: $BOR_{M\text{-}tone2} = 2.91e{-}07$; $BOR_{M\text{-}tone3} = 4.33e{-}04$; $BOR_{M\text{-}tone4} = 0.008$; $BOR_{NT1\text{-}tone2} = 4.49e{-}12$; $BOR_{NT2\text{-}tone3} = 0.010$; $BOR_{NT3\text{-}tone4} = 0.003$.

Regarding tone five, the model with the highest evidence for M showed a hierarchical architecture which consisted of the auditory cortex leading to the hippocampus and anterior cingulate gyrus and subsequently to the medial cingulate gyrus ($BOR_{M\text{-}tone5} = 0.006$). For NT4, the model with the highest evidence presented a hierarchy which started in the auditory cortex and led to the hippocampus and medial cingulate gyrus and then to the anterior cingulate gyrus ($BOR_{NT4\text{-}tone5} = 0.036$).

### Table 3 | Largest clusters of stronger activity of memorised (M) versus novel sequences (Ns)

| Contrast | ROI | Temporal extent of the largest clusters from the 1st tone of the sequence (ms) | Peak t-value | P-value |
|---|---|---|---|---|
| *Positive activity* | | | | |
| M versus NT1 (onset deviation NT1: 350 ms) | ACC | 580–830 | 7.45 | <0.001 |
| | LHG | 520–630 | 7.17 | <0.001 |
| | RHP | 580–780 | 7.66 | <0.001 |
| M versus NT2 (onset deviation NT2: 700 ms) | ACC | 940–1160 | 7.48 | <0.001 |
| | LHG | 890–970 | 6.26 | <0.001 |
| | RHP | 950–1160 | 6.24 | <0.001 |
| M versus NT3 (onset deviation NT3: 1005 ms) | ACC | 1310–1540 | 6.13 | <0.001 |
| | RHG | 1660–1760 | 5.37 | <0.001 |
| | RHP | 1320–1520 | 6.72 | <0.001 |
| M versus NT4 (onset deviation NT4: 1400 ms) | ACC | 1700–1890 | 6.54 | <0.001 |
| | LHG | 1640–1720 | 6.58 | <0.001 |
| | RHP | 1680–1890 | 6.56 | <0.001 |
| *Negative activity* | | | | |
| M versus NT1 (onset deviation NT1: 350 ms) | MC | 680–860 | −6.06 | <0.001 |
| M versus NT2 (onset deviation NT1: 700 ms) | MC | 980–1180 | −6.17 | <0.001 |
| M versus NT3 (onset deviation NT1: 1005 ms) | MC | 1360–1550 | −5.96 | <0.001 |
| M versus NT4 (onset deviation NT1: 1400 ms) | MC | 1720–1920 | −6.17 | <0.001 |

Largest clusters of significantly stronger activity of M versus Ns computed for the six selected automated anatomical (AAL) regions of interest (ROIs): left Heschl's gyrus (LHG), right Heschl's gyrus (RHG), left hippocampus (LHP), right hippocampus (RHP), anterior cingulate gyrus (ACC), and medial cingulate gyrus (MC). The table shows the contrast (two-sided t-tests), the onset of the first varied note in the N sequence, the correspondent ROI, the temporal extent (in ms) of the largest cluster, the peak t-value of the cluster and the associated Monte-Carlo simulations (MCS) p-value. The MC shows stronger negativity since the polarity of the MC signal was primarily negative, thus stronger activity in the MC for M versus N was indicated by a more pronounced negativity for M compared to N. All the other significant clusters, for both the selected six and the remaining AAL ROIs, are reported in detail in Supplementary Data 6 and 7.

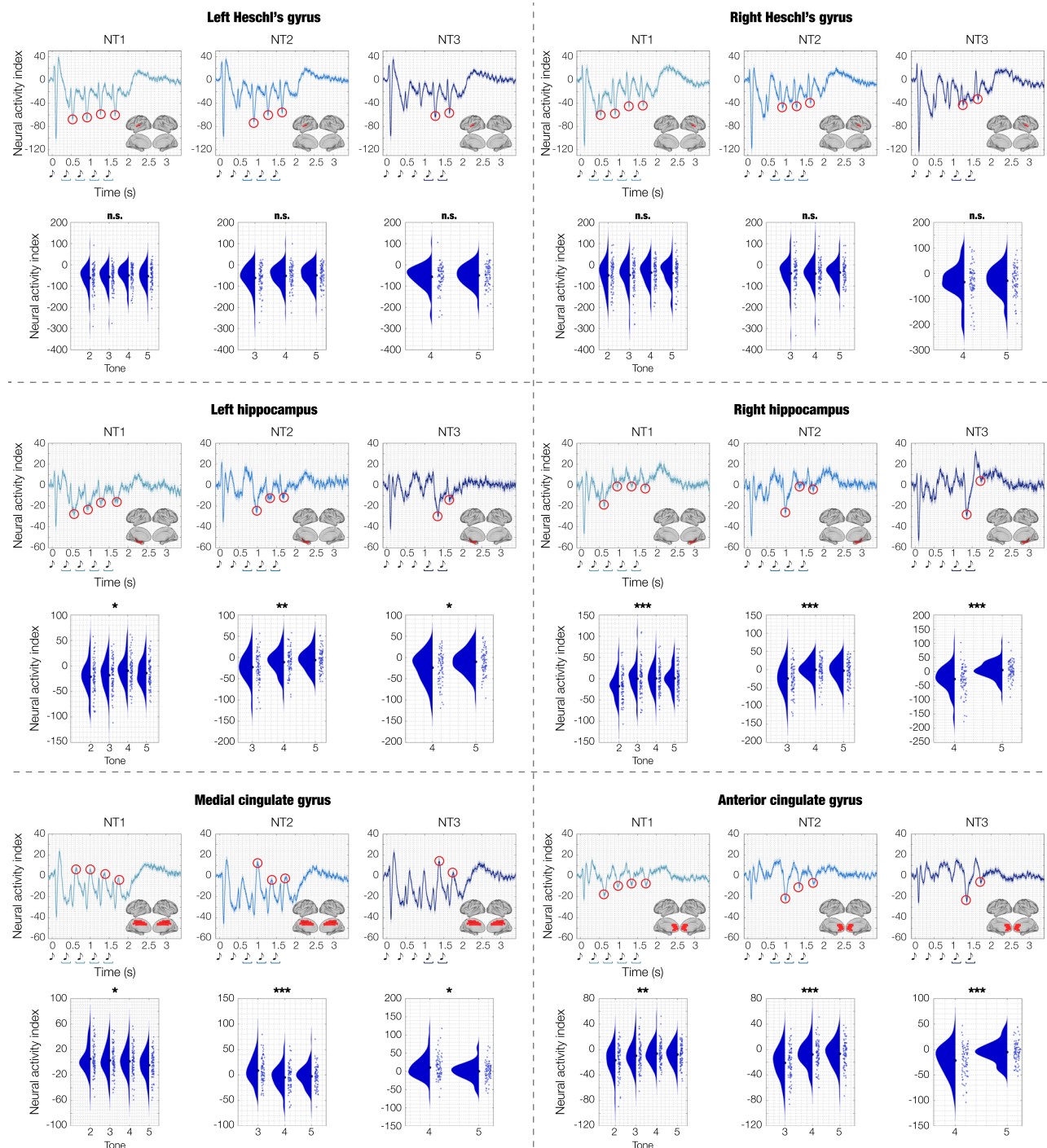

**Fig. 5 | Focus on prediction error: differentiating Heschl's gyrus from hippocampus and cingulate gyrus.** This figure focuses on prediction error responses in three novel (N) experimental conditions (novel T1 [NT1], novel T2 [NT2], novel T3 [NT3]) within six selected automated anatomical labelling (AAL) regions of interest (ROIs): left Heschl's gyrus (LHG), right Heschl's gyrus (RHG), left hippocampus (LHP), right hippocampus (RHP), anterior cingulate gyrus (ACC), and medial cingulate gyrus (MC). The figure shows the time series averaged over participants (*n* = 83). Shaded areas indicate standard errors. Musical tone sketches indicate the onset of the sounds forming the sequences. Brain templates illustrate the spatial extent of the selected ROIs. For each ROI, the circles highlight the peak responses indexing the prediction error to each of the varied tones in the sequences (occurring approximately at 150 ms after tone onset for LHG and RHG, and at 250 ms for LHP, RHP, ACC, MC. To be noted, LHP, RHP and ACC exhibit negative peaks, while MC displays positive peaks). Beneath each waveform, violin plots display the distribution of the amplitude of the peaks highlighted by the circles (each dot represents a participant). The different amplitude of these peaks was statistically tested using one-sided analyses of variance (ANOVAs), independently for each ROI. The stars indicate the significant ANOVAs after false discovery rate (FDR) correction for multiple comparisons (FDR-adjusted *p*-values: *<0.05; **<0.01; ***<0.001; n.s. not significant). Source data are provided as a Source Data file.

## Time-frequency analysis for induced responses

We computed time-frequency analysis using complex Morlet wavelet transform (from 1 to 60 Hz with 1-Hz intervals). This analysis was conducted for induced responses. First, we estimated independent time-frequency decompositions for each trial and each voxel of the six ROIs described above (LHG, RHG, LHP, RHP, ACC, MC) plus left and right occipital superior lobe for comparison purposes. Here, baseline correction was applied by subtracting, for each frequency, the average

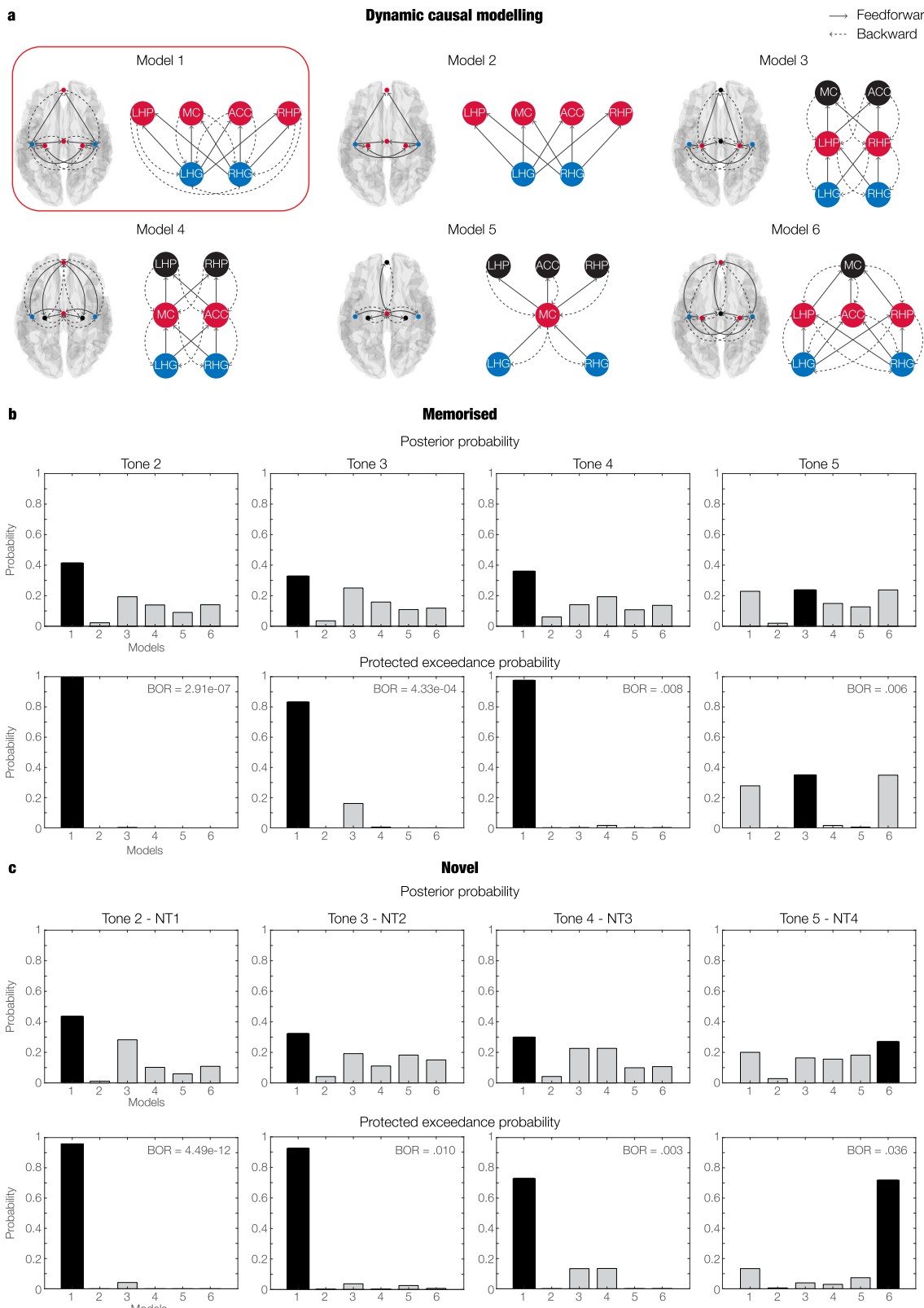

**a** Dynamic causal modelling

**b** Memorised

**c** Novel

power in the baseline interval from the power at all time points[39]. Then, the computed power spectra were averaged over voxels and over trials within each ROI. Finally, in line with the previous analyses, we calculated four contrasts (M versus NT1, M versus NT2, M versus NT3, and M versus NT4). Specifically, we computed a two-sided t-test for each frequency and time point and corrected for multiple comparisons using 2D cluster-based MCS (MCS, $\alpha = 0.05$, MCS $p$-value = 0.001). As

shown in Fig. 7, results were similar across ROIs and displayed a generalised stronger power for N versus M between 2 and 20 Hz (corresponding to, approximately, theta, alpha, and beta bands), in the time window 1.0–3.0 s ($p < 0.001$). In addition, a few significant clusters showed a moderate yet significant increase of power for M versus N ($p < 0.001$) for frequencies higher than 30 Hz, which are normally referred to as gamma[40]. Detailed statistical results about this

**Fig. 6 | Brain hierarchies during recognition of auditory sequences revealed by Dynamic Causal Modelling (DCM). a** Graphical depiction of the six alternative models employed in DCM analysis. The connections between the six selected automated anatomical labelling (AAL) regions of interest (ROIs) (left Heschl's gyrus [LHG], right Heschl's gyrus [RHG], left hippocampus [LHP], right hippocampus [RHP], anterior cingulate gyrus [ACC], medial cingulate gyrus [MC]) are illustrated within a brain template, and through a graphical representation. Our hypothesised model of brain hierarchies during recognition of memorised and varied musical sequences is enclosed in a red box. Blue, red and black circles illustrate subsequently higher levels of brain hierarchy. **b** Posterior probability, protected exceedance probability and Bayesian omnibus risk (BOR) indicating the model with the highest evidence across the population ($n = 83$; inference computed using random-effects Bayesian model selection) for the memorised (M) sequence. We conducted four independent DCM analyses, one for each tone (excluding the first tone, which was common across all experimental conditions). Our hypothesised model (model one) had the highest evidence for tones two, three and four, while model three reported the highest evidence for tone five. **c** Posterior probability, protected exceedance probability and BOR indicating the model with the highest evidence across the population ($n = 83$; inference computed using random-effects Bayesian model selection) for the novel (N) conditions. Here, we focused on the first tone which introduced the variation in the sequence (i.e. tone 2 for novel T1 [NT1], tone 3 for novel T2 [NT2], tone 4 for novel T3 [NT3], tone 5 for novel T4 [NT4]). Our hypothesised model (model one) had the highest evidence for tones two, three and four, while model six reported the highest evidence for tone five. Source data are provided as a Source Data file.

procedure are extensively reported in Supplementary Data 11 and depicted in Supplementary Fig. S12.

Finally, to strengthen the reliability of our results in MEG source space, we computed the same analysis for eight fronto-temporal and occipital MEG channels. Supplementary Fig. S13 and Supplementary Data 12 show that the outcome of this procedure aligned with the results in MEG source space.

## Discussion

This study is rooted in the foundation of previous investigations and provides a deeper understanding of the neural mechanisms and brain's functional hierarchies underlying the long-term recognition of auditory sequences. Here, in contrast to prior studies, we deliberately reduced the speed of the musical stimuli[28–32]. This allowed us to reveal that the recognition of auditory sequences relied on distinct brain responses to each tone forming the sequence, as opposed to the slow response lasting for the entire duration of the sequence detected by previous studies[28–32]. This evidence suggests that such slow signal might have been the summation of the neural responses to each sound of the sequence occurring at a faster tempo (i.e. each sound lasting 250 ms[28–32] as opposed to the 350 ms employed in the current study).

Furthermore, here we introduced systematic variations to the original memorised sequences to examine the brain mechanisms associated with conscious prediction error. The declarative nature of the prediction error, where participants were asked to identify differences in the sequences and provide a conscious response, extends our understanding beyond the existing body of research focused on automatic prediction error indexed by well-known ERP/Fs components such as N100, MMN, and ERAN[16,17,19–21]. Those studies revealed that such components were primarily generated by the auditory cortex[41,42], reporting a complementary yet much reduced recruitment of the medial cingulate gyrus, inferior frontal gyrus, and frontal and hippocampal regions[14,42]. Conversely, in our study we revealed an initial recruitment of the auditory cortex (100–150 ms) and then of the anterior and medial cingulate gyrus, ventromedial prefrontal cortex, and bilateral hippocampus (250–500 ms). Notably, our results showed that Heschl's gyrus discriminated memorised melodies versus systematic variations but did not distinguish the strength of the errors (i.e., the response was not significantly different across varied tones). Conversely, the prediction error observed in the bilateral hippocampus, anterior and medial cingulate gyrus was significantly stronger in response to the tone that introduced the variations compared to all other tones. Given the different brain responses observed in our study in comparison to previous literature on automatic prediction error, we argue that the brain signature underlying the awareness of the variation might be represented by the responses recorded in the anterior cingulate gyrus, ventromedial prefrontal cortex and hippocampus and their specific temporal dynamics.

Our results fit the well-known PCT[6,7,26], which states that the brain is constantly updating internal models to predict external information and stimuli. Recently, this theory has been proposed to explain the brain mechanisms underlying complex cognitive processes[43], finding a remarkable example in the neuroscience of music[7,26]. However, while PCT holds promise for understanding complex cognitive processes, the quantitative evidence in its favour has been limited. Thus, our study filled this gap by providing quantitative evidence of the brain functional hierarchies during long-term recognition of previously memorised and novel sequences. The results showed feedforward connections from the auditory cortices, representing sensory regions[44,45], to the hippocampus, anterior cingulate gyrus, and medial cingulate gyrus, brain areas which were previously associated with memory and predictive processes[46,47], and evaluation[48]. Simultaneously, we observed backward connections operating in the opposite direction. This hierarchical alignment persisted consistently throughout the entire sequence, except for the final tone. Here, the cingulate gyrus, which holds a pivotal role in decision-making and evaluation[48,49], assumed the top position within the hierarchy. For the memorised sequences, the hierarchy consisted of the auditory cortex sending feedforward signals to the hippocampus and anterior cingulate gyrus, and subsequently to the medial cingulate gyrus (and receiving feedback signals in the opposite direction). Similarly, for the varied sequences, the hierarchy showed feedforward connections from the auditory cortex to the hippocampus and medial cingulate and then to the anterior cingulate (and feedback in the opposite direction). This result suggests that, upon hearing the last tone, the brain might prepare the evaluation of the sequence (i.e. categorising it as 'memorised' or 'novel') and exactly for this reason either the anterior or medial cingulate gyri, both relevant regions for decision-making and evaluative processes[48,49], occupy the highest position in the hierarchy. In addition, our results present a significant dissociation between anterior and medial cingulate gyri, suggesting a prominent role of the medial part of the cingulate for recognition of memorised sequences and of the anterior part for novelty detection.

These findings align with previous research demonstrating the flow of information from sensory cortices to the medial temporal lobe, associative areas, and prefrontal regions of the brain[23,50]. Moreover, our results are coherent with the several studies which used PCT to investigate perceptual[51] and automatic memory mechanisms in the brain[22], as reviewed by Spratling[43]. These studies described how information processing, orientation tuning, contour integration, and binocular rivalry in the visual system relied on the hierarchical transition of the psychophysical and neurophysiological signals from the retina to the lateral geniculate nucleus and V1[52–55]. In the context of automatic sensory memory, PCT has also lent support to the interpretation of the MMN as an error signal which aids the brain in adjusting its internal predictive model, returning quantitative evidence of information flow from the primary auditory cortex to the superior temporal gyrus and inferior frontal gyrus[22]. Our findings expand on these studies by providing quantitative evidence of the hierarchical organisation of the brain during a complex cognitive task, involving episodic memory recognition for musical sequences. In addition, while the hierarchical architecture involved in recognising both memorised

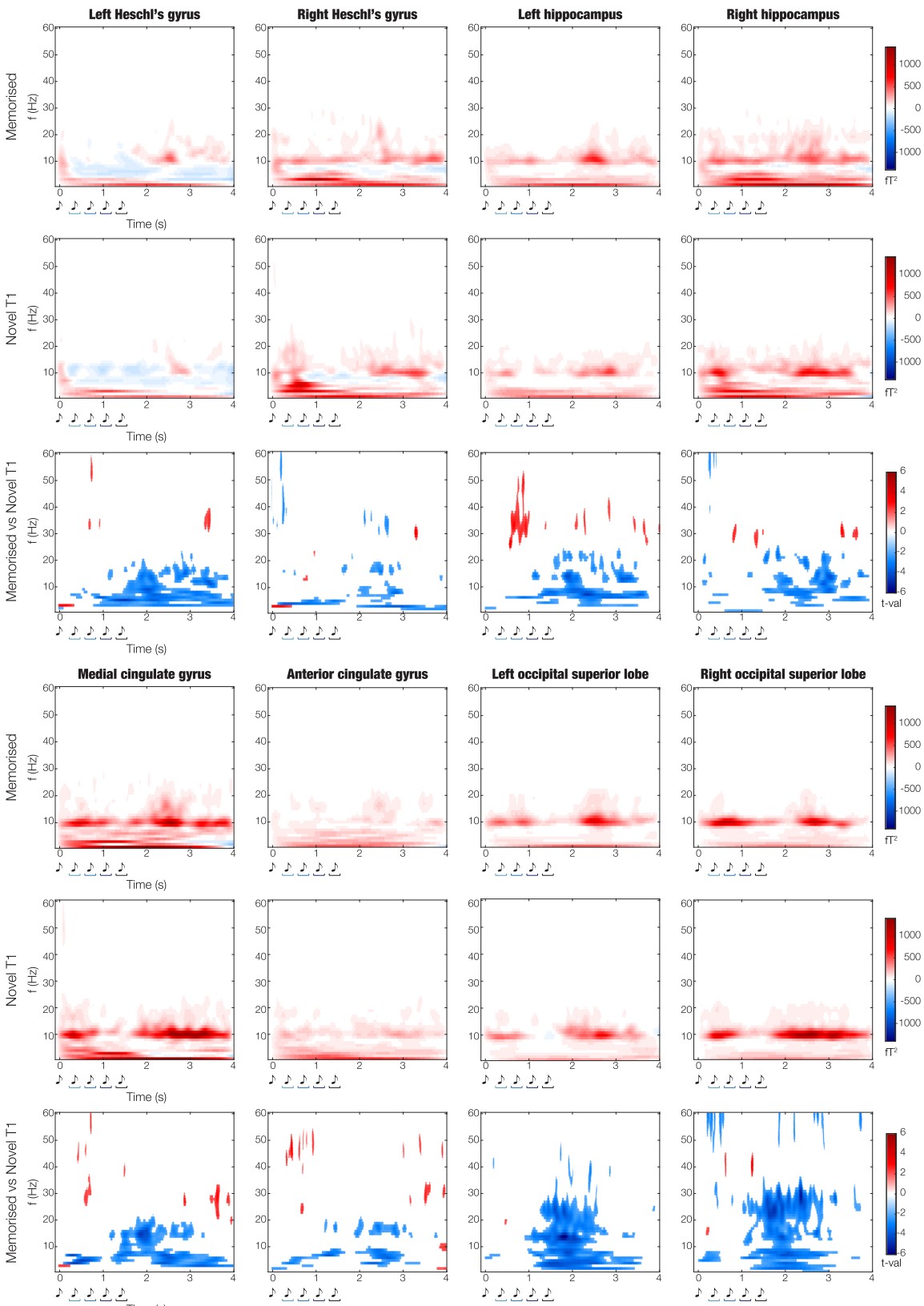

and varied sequences remained largely unchanged, the temporal dynamics, strength, and polarity of the brain signal sharply differed. Here, our results suggest that when the upcoming sound of the sequence matched the predicted sound based on the previously stored memory trace, first the auditory cortex and then hippocampus, anterior and medial cingulate gyri responded with positive components. Conversely, when the upcoming sound was incoherent with the prediction made by the brain, a pathway of primarily negative and faster components emerged within the same brain network. It is worth noting that also the inferior frontal gyrus, typically linked to short-term and working memory for music[56,57], displayed significantly different activity across experimental conditions. However, this region

**Fig. 7 | Induced responses during the recognition of memorised (M) and novel T1 (NT1) sequences.** For each of the selected automated anatomical labelling (AAL) regions of interest (ROIs; left Heschl's gyrus [LHG], right Heschl's gyrus [RHG], left hippocampus [LHP], right hippocampus [RHP], anterior cingulate gyrus [ACC], medial cingulate gyrus [MC], plus left and right occipital superior lobe for comparison purposes), three plots are provided. First, we illustrate the power spectrum computed using complex Morlet wavelet transform for the M and NT1 conditions. The power spectra were baseline corrected by subtracting, for each frequency, the average power in the baseline interval from the power at all time points. The colorbar indicates squared femtotesla (fT²). Second, we illustrate the significant results emerging from the contrasts between the power spectra of M versus NT1 (two-sided t-tests [$n$ = 83 participants] and cluster-based Monte-Carlo simulations [MCS; MCS, α = 0.05, MCS $p$-value = 0.001] correction for multiple comparisons). In this case, the colorbar indicates the t-values obtained by contrasting M versus NT1. Source data are provided as a Source Data file.

exhibited overall reduced responses compared to the other brain regions within the network. This observation further underscores the differentiation between long-term memory recognition for auditory sequences and other memory subsystems.

An additional finding pertains to our utilisation of multivariate pattern analysis and temporal generalisation techniques, as described by King and Dehaene[58]. Here, we revealed that when decoding memorised from novel sequences, the algorithm could differentiate the conditions starting from the first tone that was varied in the novel sequences. This learning could then be generalised by the algorithm with significant (yet reduced) accuracy for all the subsequent tones. This pattern was consistently observed across all categories of variations, and it implies that the differential processes between the two conditions remained consistent over time, indicating that the brain not only identifies the initial varied tone but also monitors the entire auditory sequence. This monitoring might have served the purpose of assessing whether the varied tone represented a transient mistake or the beginning of a proper varied sequence.

Our findings revealed a positive overall correlation between musical training and neural activity associated with the recognition of memorised and varied musical sequences. This correlation was particularly evident with regards to the neural responses to the sounds that introduced variations. This aligns with previous research indicating a connection between musical expertise and altered brain characteristics, including enhanced neural responses (e.g. enhanced auditory automatic prediction error)[14,26,59] and structural changes[60]. In this study, what distinguishes our findings was the extension of this relationship to the domain of long-term recognition of musical sequences and the underlying brain mechanisms.

On a behavioural level, both reaction times and accuracy remained consistent across memorised and three categories of novel sequences: NT1, NT2, and NT3. However, accuracy was significantly reduced, and reaction times increased for NT4. This shift might be attributed to the altered final tone in the sequence, eliciting a slower behavioural prediction error. Alternatively, the reduced accuracy and prolonged reaction times in NT4 could be linked to chunking; the mental grouping process that occurs when sequences are presented with a beat every four tones. In this view, after listening to four tones of the memorised sequence (corresponding to a complete beat), the perception that the sequence belonged to the group of previously learned sequences was very strong, and especially much stronger than after only three tones. For this reason, we did not observe a linear increase in reaction times and accuracy but only a difference between T4 and all the other conditions. Currently, we do not have enough data to make definitive claims and future studies are needed to systematically vary the length of the sequences.

Finally, the induced-response analysis showed that, after the end of the sequence, alpha and beta bands were stronger for the varied compared to the memorised sequences. Considering the modulation of alpha and beta activity in encoding and working memory tasks described in the scientific literature[61], this finding may suggest that post-stimulus increase of alpha-beta power might be relevant to process novel information. In addition, this analysis revealed a weak yet significant increase in gamma power for the memorised compared to the varied sequences, especially in the left hippocampus and Heschl's gyrus and in the cingulate gyrus. This is coherent with previous studies which reported increased gamma

power during recognition of target stimuli[62,63] and, more generally, a modulation of the brain oscillations associated with memory load and complex cognitive functions[64]. Although this study provided initial insights on the time-frequency modulation occurring in the brain during long-term recognition of auditory sequences, further research employing MEG and additional tools such as stereo-electroencephalography (SEEG) is needed to expand our experimental design and conduct cross-frequency coupling analysis, testing whether gamma-theta coupling is connected to long-term recognition of auditory sequences. This would also provide supporting evidence of the quality of the source reconstruction performed in this study since SEEG data is less sensitive to source leakage artifacts than MEG source reconstruction algorithms[65].

In conclusion, this study reveals the brain mechanisms and hierarchical dynamics of long-term memory recognition of previously memorised and varied auditory sequences, effectively integrating the PCT in the domain of complex cognitive tasks.

## Methods
### Participants
The participant sample consisted of 83 volunteers [33 males and 50 females (sex, biological attribute, self-reported)] aged 19–63 years old (mean age: 28.76 ± 8.06 years). The sample was recruited in Denmark and participants came from Western countries. We have not collected information about participants' gender since this was beyond the scope of our research. All participants were healthy and reported normal hearing. Their educational background was overall homogeneous. Specifically, 73 participants had either a university degree (bachelor's or master's degree, $n$ = 54) or were university students ($n$ = 19). The remaining 10 were divided as follows: five had a professional degree obtained after finishing high-school, while the remaining five had a high-school diploma. The project was approved by the Institutional Review Board (IRB) of Aarhus University (case number: DNC-IRB-2020-006). The experimental procedures were carried out in compliance with the Declaration of Helsinki – Ethical Principles for Medical Research. All participants gave the informed consent before starting the experimental procedure and received compensation for their participation in the study.

### Experimental stimuli and design
In this study, we used an old/new paradigm auditory recognition task[28,29,31,32] during magnetoencephalography (MEG) recordings. First, participants listened to a short musical piece twice and were asked to memorise it as much as possible. The musical piece consisted of the first four bars of the right-hand part of Johann Sebastian Bach's Prelude No. 2 in C Minor, BWV 847. In this piece, each bar included 16 tones. Thus, the total number of tones was 16*4 = 64. Each tone lasted approximately 350 ms for a total of 22,400 ms. In addition, to provide a sense of musical closure, we included a final tone after the four bars which lasted 1000 ms. Thus, the total duration was 23,400 ms (23.4 s). The stimulus is illustrated in musical notation in Supplementary Fig. S1. Second, participants were presented with 135 five-tone musical excerpts that lasted 1750 ms each. Participants were requested to state whether each excerpt belonged to the original music ('memorised' sequence [M], old) or was a varied musical sequence ('novel' sequence [N], new) (Fig. 1a). Twenty-seven excerpts were drawn from the original musical piece and 108 were variations of the original melodies

(Supplementary Fig. S2 shows all the sequences used in the study). The two categories of stimuli (M and N) were created as follows. The M sequences were comprised by the first five tones of the first three measures of the musical piece. These sequences were presented nine times each, for a total of 27 trials. The N sequences were created through systematic variations of the three M sequences (Fig. 1b). This procedure consisted of changing every musical tone of the sequence after the first (NT1), second (NT2), third (NT3) or fourth (NT4) tone. We created nine variations for each of the original M sequences and each of the four categories of N. This resulted in 27 N sequences for each category, and 108 N in total. To be noted, the variations were created according to the following rules:

- Inverted melodic contours (used twice): the melodic contour of the variation was inverted with respect to the original M sequence (i.e., if the M sequence had the following melodic contour: down-down-up-down, the N sequence would be: up-up-down-up).
- Same tone scrambled (used three times): the remaining tones of the M sequence were scrambled (e.g., M sequence: C-E-D-E-C, was converted into NT1 sequence: C-C-E-E-D). When this was not possible (e.g., in the case of NT4, where only the last tone is different from the M sequence), we substituted the last tone of the M sequence with a random tone.
- Same tone (used three times): the same tone was repeatedly used, in some cases varying only the octave (e.g., M sequence: C-E-D-E-C, was transformed into NT1 sequence: C-E8- E8- E8- E8).
- Scrambling intervals (used once): the intervals between the tones were scrambled (e.g., M sequence: 6th m – 2nd m – 2nd m – 3rd m, was adapted to NT1 sequence: 2nd m, 6th m, 3rd m, 2nd m).

To be noted, the harmonic structure of the N sequences with regards to the original M sequences was preserved in most of the cases. The only exceptions occurred for a few instances of the 'inverted melodic contours' and 'scrambling intervals' strategies, which were however important to control for other musical features such as melodic contour and intervals. This strategy was implemented to avoid potential confounding variables stemming from changes in harmony. However, investigating harmonic alterations concerning conscious brain prediction errors could prove relevant for future studies centred on diverse types of prediction errors.

The current procedure allowed us to investigate (i) the brain dynamics underlying the recognition of previously memorised auditory sequences and (ii) the conscious detection of the sequence variation.

MIDI versions of the musical piece and of the musical sequences described above were created using Finale (version 26, MakeMusic, Boulder, CO) and presented using Psychopy v3.0.

## Data acquisition
The MEG recordings were acquired in a magnetically shielded room at Aarhus University Hospital (AUH), Aarhus, Denmark, using an Elekta Neuromag TRIUX MEG scanner with 306 channels (Elekta Neuromag, Helsinki, Finland). The data was recorded at a sampling rate of 1000 Hz with an analogue filtering of 0.1–330 Hz. Before the recordings, the head shape of the participants and the position of four Head Position Indicator (HPI) coils were registered with respect to three anatomical landmarks using a 3D digitiser (Polhemus Fastrak, Colchester, VT, USA). This recording was later used to co-register the MEG data with the MRI anatomical scans. For the entire duration of the MEG recordings, the HPI coils registered the continuous localisation of the head, which was subsequently employed for movement correction. In addition, two sets of bipolar electrodes were used to record cardiac rhythm and eye movements. This allowed us to remove the electrocardiography (ECG) and electrooculography (EOG) artifacts in a later stage of the analysis pipeline.

The MRI scans were recorded on a CE-approved 3T Siemens MRI-scanner at AUH. The recorded data consisted of structural T1 (mprage with fat saturation) with a spatial resolution of $1.0 \times 1.0 \times 1.0$ mm and the following sequence parameters: echo time (TE) = 2.61 ms, repetition time (TR) = 2300 ms, reconstructed matrix size = $256 \times 256$, echo spacing = 7.6 ms, bandwidth = 290 Hz/Px.

The MEG and MRI recordings were acquired in two separate days.

## Behavioural data
We obtained behavioural data (number of correctly recognised trials and correspondent reaction times) from the experimental task carried out during the MEG recording.

Since the data was not normally distributed, we computed two independent Kruskal-Wallis H tests (non-parametric one-way analysis of variance) to assess whether the five categories of temporal sequences (M, NT1, NT2, NT3, NT4, NT5) differed in terms of correct responses and reaction times. Multiple comparisons were corrected using the Tukey-Kramer correction[66].

## MEG data pre-processing
The raw MEG sensor data (204 planar gradiometers and 102 magnetometers) was first pre-processed by MaxFilter[67] (version 2.2.15) to attenuate external interferences. We applied signal space separation (MaxFilter parameters: spatiotemporal signal space separation [SSS], down-sample from 1000 Hz to 250 Hz, movement compensation using cHPI coils [default step size: 10 ms], correlation limit between inner and outer subspaces used to reject overlapping intersecting inner/outer signals during spatiotemporal SSS: 0.98). We opted to down-sample the data by a factor of four because MEG is not particularly sensitive to high-gamma frequency (e.g. 100 Hz)[25,68], while it provides much better outcomes for lower frequency bands (e.g. 0.1–60 Hz, as done in the current study)[25,68]. Moreover, our research primarily focused on the event-related broadband brain data during memory recognition, which are often expressed in slower bands[27,28]. For these reasons, as commonly done in the literature[25,68], we downsampled the data, a procedure which significantly accelerated the computations by reducing data volume without substantial data loss[25,68].

The data was then converted into Statistical Parametric Mapping (SPM) format and further pre-processed and analysed in MATLAB (MathWorks, Natick, MA, USA) using a combination of in-house-built codes (LBPD, https://github.com/leonardob92/LBPD-1.0.git) and the Oxford Centre for Human Brain Activity (OHBA) Software Library (OSL)[69] (https://ohba-analysis.github.io/osl-docs/), a freely available software that builds upon Fieldtrip[70], FSL[71], and SPM[72] toolboxes.

The continuous MEG data was visually inspected to identify and remove large artifacts using the OSLview tool. The data that was removed was less than 0.1% of the amount of collected data. Independent component analyses (ICA) were used (OSL implementation) to discard the interference of eyeblinks and heartbeat artefacts from the brain data[73]. First, we decomposed the original signal into independent components. Second, we correlated all the components with the activity recorded by the EOG and ECG channels. This procedure identified the ICA component X that strongly correlated with either the EOG or the ECG (strongly correlated means that the correlation coefficient for component X was at least three times higher than the correlation coefficients for the remaining components). To further validate the accuracy of this procedure, we also visually inspected the highly correlated components. We assessed whether their topographic distribution across MEG channels matched the typical distribution associated with eyeblink and heartbeat activity. When both the correlations and visual inspections converged, indicating that the ICA component strongly reflected eyeblink or heartbeat activity, those components were discarded. Third, the signal was rebuilt using the remaining components. Finally, the signal was epoched in 135 trials (27M, 27 NT1, 27 NT2, 27 NT3, 27 NT4) and baseline-corrected by

removing the mean signal recorded in the baseline from the post-stimulus brain signal. Each trial lasted 4500 ms (4400 ms plus 100 ms of baseline time).

## Multivariate pattern analysis (decoding) – MEG sensors

We performed multivariate pattern analysis to decode different neural activity associated with the recognition of M versus N (Fig. 1c). Here, we computed four independent analyses, decoding M from each of the four categories of the novel sequences (i.e., M versus NT1, M versus NT2, M versus NT3, M versus NT4). This was our first analysis since decoding relies on all the available data and makes no assumptions about which MEG channels or time points should be selected. We used support vector machines (SVMs)[74] and calculated independent analyses for each participant. The MEG data was rearranged in a 3D matrix (channels x time points x trials) and submitted to the SVM algorithm. To avoid overfitting, a leave-one-out cross-validation approach was adopted to train the SVM classifier to decode the two experimental conditions. This procedure divided the trials into N different groups (here $N = 5$). Then, for each time point, it assigned N − 1 groups to the training set and the remaining $N_{th}$ group to the testing set. After that, the classifier ability to separate the two conditions was evaluated. This process was performed 100 times with random reassignment of the data to training and testing sets. To summarise, the decoding accuracy time series were averaged to obtain a final time series showing the performance of the classifier for each participant. In addition to the decoding accuracy time series (Fig. 2a), this analysis returned weights and activation patterns which can be used to learn which MEG channels provided the strongest contribution to the decoding analysis[75] (Fig. 2b).

To test the significance of the decoding results (chance level set at 50%), we employed a signed-rank permutation test against the chance level for each time point and then corrected for multiple comparisons using false-discovery rate (FDR) correction ($\alpha = 0.05$; FDR-adjusted $q < 0.012$).

To assess whether each pair of conditions were differentiated by neural patterns which were stable over time, we computed four temporal generalisation multivariate analyses (Fig. 2c). In this case, the algorithm used each time point of the training set to predict not only the same time point in the testing set, but all time points[76]. Here, the significance was tested using a signed-rank permutation test against the chance level (50%) for each time point, as the previous analyses. Then, we corrected for multiple comparisons using two-dimensional (2D) cluster-based Monte-Carlo simulations (MCS, $\alpha = 0.05$, MCS $p$-value = 0.001). First, we computed the clusters size of the continuous, binarized, significant values over time. The MCS $\alpha = 0.05$ means that to binarize the values outputted by the signed permutation test against the chance level that we described above, we used a $p$-value threshold of .05 (i.e. values below 0.05 were considered significant and inputted to the MCS as '1s' while values above 0.05 as '0s'). Thus, the first step of the MCS algorithm consisted of detecting the clusters of '1s'. Second, we made 1000 permutations of these binarized values. For each permutation, we computed the size of the maximum emerging cluster and built a reference distribution using those values. Finally, we considered significant the original clusters that were bigger than the 99.9% of the permuted data maximum cluster sizes (in other words, the MCS $p$-value = 0.001 refers to the percentage of the cluster sizes in the reference distribution that the original clusters should exceed to be considered significant). This procedure was observed for all the analogous MCS described in the study. Moreover, additional details on this MCS algorithm can be found in[28,31,32,77].

## Univariate analysis – MEG sensors

Pairwise decoding is a powerful technique, but it can only tell if two conditions are characterised by significantly different brain activity. It does not provide information about which condition presented the strongest brain activity. To answer this question, we computed an additional analysis using the brain activity recorded by the MEG channels which mainly contributed to the decoding algorithm. This was done following two steps: (i) averaging the activation patterns of the MEG channels in all the significant time windows emerged from the decoding algorithm (i.e. FDR-corrected decoding accuracy time series for M versus NT1, M versus NT2, M versus NT3, M versus NT4); (ii) selecting the MEG channels whose activation pattern was higher than the average (over the MEG channels) plus one standard deviation. This was done independently for magnetometers and gradiometers and returned the following MEG channels; magnetometers: 1321, 0211, 1441, 0131, 1411, 1221, 1331, 0121, 1311, 0341, 2611, 0241, 1511, 1341, 0111, 0221; gradiometers: 1442, 0242, 0222, 1332, 2612, 0232, 1513, 1312, 0312, 1222, 1342, 0342, 0123, 1423, 2422, 2613, 0213, 1212, 1323, 0413, 0132, 0142, 0113, 1233, 2033, 1413, 1123, 0143, 0323, 1343, 1612, 1412.

These MEG channels were then averaged into four distinct groups, based on the channel type and the polarity of the signal: (i) magnetometers negative N100 (Fig. 2d, left), (ii) magnetometers positive N100 (Fig. 2d, right), (iii) gradiometers negative N100 (Fig. 2e, left), and (iv) gradiometers positive N100 (Fig. 2e, right). The reasons why these four groups were considered are as follows: (i) magnetometers and gradiometers are different types of sensors and have a difference scale so they are usually analysed independently[25,68]; (ii) because of the relationship between electrical activity and the correspondent magnetic field, when the brain generates a truly negative response (e.g. N100 to the first sound of the sequence), the MEG channels record a magnetic field characterised by both negative and positive values (i.e. the same electric activity generates both a positive and negative polarity of the magnetic field that are measured independently by different MEG channels). The result of this well-known effect is that some channels show the 'true' polarity of the brain signal (e.g. the N100 to the first sound is negative) and other show the inverted polarity (e.g. the N100 to the first sound is positive). When averaging MEG channels together, this must be considered to prevent averaging out the signal. Thus, as depicted in Fig. 2, we averaged the MEG channels into the above-described four groups. As follows, we report the selected MEG channels divided into the four groups. (i) Magnetometers negative N100: 0131, 0211, 0121, 0341, 0241, 1511, 0111; (ii) magnetometers positive N100: 1321, 1441, 1411, 1221, 1331, 1311, 2611, 1341; (iii) gradiometers negative N100: 0213, 0232, 0312, 0342, 0132, 0222, 1423, 1212, 1123, 0413, 1413, 1233, 0323, 2613; (iv) gradiometers positive N100: 1442, 0242, 1332, 2612, 1513, 1312, 1222, 1342, 0123, 2422, 1323, 0142, 0113, 2033, 0143, 1343, 1612, 1412.

Then, independently for each group, we computed one two-sided t-test for each time point and each combination of M versus Ns (i.e., M versus NT1, M versus NT2, M versus NT3, M versus NT4). We corrected for multiple comparisons using a one-dimensional (1D) cluster-based MCS (MCS, $\alpha = 0.05$, MCS $p$-value = 0.001) (Supplementary Data 3). First, we identified the clusters of significant continuous values in time. Second, we computed 1000 permutations, randomising the significant values obtained from the t-tests. For each permutation, we extracted the maximum cluster size and built their reference distribution. Finally, we considered significant the original clusters that were larger than 99.9% of the permuted ones.

To provide full disclosure of our data, we have complemented these computations with a further analysis. In this case, we computed the same procedure described above, contrasting M versus each category of N (i.e. M versus NT1, M versus NT2, M versus NT3, M versus NT4) and correcting for multiple comparisons with cluster-based MCS (MCS, $\alpha = 0.05$, MCS $p$-value = 0.001). However, here we computed the contrasts independently for each MEG channel. The results are depicted in Supplementary Fig. S5a–z2 and reported in detail in Supplementary Data 4.

## Source reconstruction

MEG is a powerful tool to detect the whole-brain activity with excellent temporal resolution. However, to obtain a complete picture of the whole-brain activity underlying complex cognitive tasks the spatial component of the brain activity must be also identified. Here, we employed the established beamforming method[78–80] (Fig. 1e), built upon a combination of in-house-built codes and codes available in OSL, SPM, and FieldTrip.

To reconstruct the brain sources that generated the MEG signal, an inverse problem must be solved. The MEG recording shows the activity of the neural signals outside the head but cannot provide information on the specific brain sources which generated it. Thus, we used beamforming algorithms to solve this problem, implementing the two following steps: (i) designing a forward model and (ii) computing the inverse solution.

The forward model is a theoretical model which considers each brain source as an active dipole (brain voxel). It describes how the unitary strength of each dipole would be reflected over all MEG sensors. Here, we employed only the magnetometer channels and an 8-mm grid, which returned 3559 dipole locations (voxels) within the whole brain. This was done because the magnetometers are the MEG sensors that better capture the activity coming from deep sources of the brain, while the gradiometers are more sensitive to the activity originated in the portion of the cortex nearest to each gradiometer[25,68]. Since our hypothesis was that regions in the medial temporal lobe (e.g. hippocampus) played a role in long-term auditory recognition, we opted for the magnetometers to compute the source reconstruction. After co-registering the individual structural T1 data with the fiducial points (i.e., information about head landmarks), we computed the forward model by adopting the widely used method called Single Shell, which is presented in detail in Nolte[81]. The output of this computation, referred to as leadfield model, was stored in the matrix L (sources × MEG channels). In the three cases where the structural T1 was not available we performed the leadfield computation using a template (MNI152-T1 with 8-mm spatial resolution).

Then, we computed the inverse solution. As mentioned above, we chose the beamforming, which is one of the most popular and effective algorithms available in the field. This procedure employs a different set of weights which are sequentially applied to the source locations for isolating the contribution of each source to the activity recorded by the MEG channels. This is done for each time point of the recorded brain data. The beamforming inverse solution can be summarised by the following main steps.

The data recorded by MEG sensors (**B**) at time t, can be described by the following Eq. (1):

$$\mathbf{B}_{(t)} = \mathbf{L} * \mathbf{Q}_{(t)} + \varepsilon \qquad (1)$$

where **L** is the leadfield model, **Q** is the dipole matrix carrying the activity of each active dipole (q) at time t and ε is noise (see Huang and colleagues[80] for details). To solve the inverse problem, **Q** must be computed. In the beamforming algorithm, weights are computed and then applied to the MEG sensors at each time point, as shown for the single dipole q in Eq. (2):

$$\mathbf{q}_{(t)} = \mathbf{W}^T * \mathbf{B}_{(t)} \qquad (2)$$

To obtain q, the weights **W** should be computed (the superscript T refers to transpose matrix). To this goal, the beamforming relies on the matrix multiplication between **L** and the covariance matrix between MEG sensors **C**, which is calculated on the concatenated experimental trials. Specifically, for each brain source (dipole) q, the weights **W_q** are

computed as shown in Eq. (3):

$$\mathbf{W}_{(q)} = \left(\mathbf{L}_{(q)}^T * \mathbf{C}^{-1} * \mathbf{L}_{(q)}\right)^{-1} * \mathbf{L}_{(q)}^T * \mathbf{C}^{-1} \qquad (3)$$

To be noted, the computation of the leadfield model was performed for the three main orientations of each brain source (dipole), according to Nolte[81]. Before computing the weights, the orientations were reduced to one using the singular value decomposition algorithm on the matrix multiplication reported in Eq. (4). This procedure is widely adopted to simplify the beamforming output[64,65].

$$\mathbf{L} = svd(\mathbf{l}^T * \mathbf{C}^{-1} * \mathbf{l})^{-1} \qquad (4)$$

Here, **l** represents the leadfield model with the three orientations, while **L** is the resolved one-orientation model that was utilised in (3). Finally, the weights were applied to each brain source (dipole q) and time point. To be noted, the covariance matrix **C** was computed on the continuous signal, which was estimated by concatenating the trials of all experimental conditions. The weights were applied to the brain data associated with each condition and normalised according to Luckhoo and colleagues[78,82–84] for counterbalancing the reconstruction bias towards the centre of the head. The weights were applied to the neural activity averaged over trials for the evoked responses and to the neural activity of each independent trial for the induced responses. This procedure returned a time series for each of the 3559 brain sources (and each trial in the case of induced responses), commonly referred to as neural activity index[78,82]. The sign ambiguity of the evoked responses time series was adjusted for each brain source using its sign in correspondence with the N100 response to the first tone of the auditory sequences[28,31,32].

## Neural sources of the peak activity (MEG magnetometers) and musical training

After showing evidence of the differential brain activity in relation to the experimental conditions, we wanted to characterise the neural sources of the main peaks of brain activity recorded by the magnetometers in response to each tone of the musical sequences, as described in the section on the univariate analysis of MEG channels.

To detect the peaks of the brain activity recorded by the magnetometers for each tone (except the 1st one) of the musical sequence, we extracted the time indices of the minimum and maximum value recorded by the magnetometers shown in Figs. 2d and 3 in a 400 ms time window following the onset of each tone. The duration of this time window reflects the period during which independent responses to each sound occurred. This procedure returned the time index of two neural peaks for each experimental condition and each tone of the sequence (with exclusion of the 1st one which did not differ across conditions). This revealed that one peak of the neural activity occurred approximately at 350 ms after the onset of each tone (highlighted in yellow in Fig. 3). With regards to this peak, condition M showed the strongest response among all conditions. The other peak occurred in response to the tone which introduced the variation in the musical sequence (highlighted in purple in Fig. 3). In this case, the N condition which introduced the variation reported the strongest response among all other conditions. This occurred approximately at 250 ms after the varied tone (i.e. NT1 showed the strongest response to tone two among all conditions, NT2 to tone three, NT3 to tone four and NT4 to tone five). Extensive details on these neural activity peaks are reported in Table 2.

Then, for each of the defined peaks, we computed contrasts in MEG source space between M and N. More specifically, with regards to the peaks occurring approximately at 350 ms after each tone, we contrasted M versus NT1 (illustrated in the yellow boxes in Fig. 3). This was done because the peaks occurring at 350 ms after each tone were always greater for M, and NT1 was the only condition comprising

sounds that were always different from M. Similarly, with regards to the peaks occurring approximately at 250 ms after the onset of the varied tones, we contrasted the N condition which showed the strongest peak against M (illustrated in the purple boxes in Fig. 3). Here, for each condition in the pair, we averaged the absolute value of the activity of each of the 3559 reconstructed brain voxels in the ±20 ms time window around the correspondent time index. Then, we computed two-sided t-tests for each of the 3559 brain voxels by contrasting the pairs of conditions and FDR-corrected for multiple comparisons. The results of these analyses are shown in Fig. 3 and reported in detail in Table 2 and Supplementary Data 5. In addition, we reported the neural activity recorded by the MEG channels in topographic maps in Supplementary Fig. S6.

Furthermore, the brain activity recorded by the MEG magnetometers for each neural peak and hemisphere was correlated with the measure of participants' musical training provided by the Goldsmith Musical Sophistication Index (GOLD-MSI)[34] and corrected for multiple comparisons using FDR. The GOLD-MSI is a widely used standardised test which measures the ability of engaging with music in five subscales: (i) Active engagement with music, (ii) Perceptual abilities, (iii) Musical training, (iv) Emotions, and (v) Singing abilities. Here, we focused on the Musical training subscale, which encompasses data related to music education, formal training, average time spent playing musical instruments at different life stages, etc[34]. The Results are reported in Table 2.

## Statistical analysis – automated anatomical labelling (AAL) time series

After evaluating the neural sources of the different brain activity associated with the experimental conditions using the fine-grained 8-mm parcellation of the brain (Fig. 1e, 8-mm parcellation), we focused on a set of anatomically defined regions of interest (ROIs) (Fig. 1e, AAL parcellation). Here, we used a standard anatomical parcellation method known as automated anatomical labelling (AAL)[35] and calculated a time series for each of the 90 non-cerebellar ROIs of AAL (Fig. 1f, Evoked responses).

As specified above, our source reconstruction was conducted in an 8-mm space, resulting in 3559 brain voxels, each with its own time series. Thus, for each of the 90 ROIs, we identified the corresponding brain voxels and then averaged the time series of these associated voxels. This allowed us to obtain a final time series for each AAL ROI.

Then, we computed one two-sided t-test for each AAL ROI, each time point and each combination of M versus Ns. Finally, we corrected for multiple comparisons using one-dimensional (1D) cluster-based MCS (MCS, α = 0.05, MCS p-value = 0.001). Supplementary Fig. S7a–g and Supplementary Data 6 show all AAL ROIs and the significant differences between conditions. Figure 4 and Supplementary Fig. S8 show instead a selected array of ROIs which were particularly relevant for this study. Here, we identified the two ROIs that showed the strongest activity in absolute terms among auditory regions (i), medial temporal lobe (ii) and cingulate and prefrontal cortices (iii). We selected these regions based on the cognitive processes involved in the experimental task used in this study: audition (i), memory (ii), evaluation and decision-making (iii). As shown by Fig. 4 and Supplementary Fig. S8, these ROIs were left and right Heschl's gyrus (LHG, RHG), left and right hippocampus (LHP, RHP), anterior cingulate gyrus (ACC) and medial cingulate gyrus (MC), respectively. These six regions were also used for the Dynamic Causal Modelling (DCM) analysis described later.

In addition, since it has been shown that the source reconstruction can produce leakage of the brain signals between the ROIs[36], we took additional measures to ensure the robustness of our results. Specifically, we applied the multivariate source leakage correction method proposed by Colclough and colleagues[36] to the ROIs time series. This method consists of removing the zero-lag correlations

between the ROIs time series through a symmetric, multivariate orthogonalization process. After computing this correction, we performed the exact same statistical tests described above. The outcomes of this analysis are illustrated in Supplementary Fig. S9 and described in Supplementary Data 8.

Finally, as a further measure to enhance the reliability of our findings, in supplementary materials we reported the description of an alternative, functional parcellation derived from the data (Supplementary Figs. S7a–g, S10, S11, S15–S18 and Supplementary Data 9, 13, 14).

## Prediction error across brain regions of interest

After differentiating the brain activity of M versus N, we conducted an additional analysis to investigate the responses elicited by the varied tones within the N conditions. Regarding Heschl's gyrus (left and right), we focused on the late N100 peaking approximately at 150 ms after the onset of each varied tone. Regarding the hippocampus (left and right), anterior and medial cingulate gyrus, we focused on the responses peaking approximately at 250 ms after the onset of each varied tone (these responses were negative for hippocampus and anterior cingulate gyrus and positive for the medial cingulate gyrus). Our goal was to compare the responses to each individual tone independently within each ROI, for N conditions containing at least two varied tones. To achieve this, we computed the average neural activity within a ±20 ms time window centred around the designated time index (150 ms for Heschl's gyri and 250 ms for the hippocampus and cingulate gyrus), as depicted in Fig. 5 using red circles. Subsequently, we performed separate one-sided analyses of variance (ANOVAs) for each ROI and N condition containing at least two varied tones to assess whether the responses to subsequent varied tones differed, independently for each ROI. Here, we used the sequential neural peaks as independent variable and the neural activity as the dependent variable. Since this approach led to a total of 18 ANOVAs, we applied an FDR correction to account for multiple comparisons. Moreover, the posthoc analyses computed for each of the ANOVAs were corrected for multiple comparisons using Tukey–Kramer correction[66].

## Dynamic causal modelling (DCM)

DCM aims to infer the causal structure of interconnected or dispersed dynamical systems such as a network of brain regions (Fig. 1g). It employs a Bayesian model comparison approach that involves evaluating competing models which explain the generation of time series data associated with alternative network architectures. DCM has been widely used in the neuroscientific literature[37,38,85–88], especially but not limited to the framework of predictive coding, and it is particularly useful for estimating hierarchical relationships between brain regions in relation to experimental stimuli. In brief, a dynamic causal model for MEG responses starts with a reasonably plausible neuronal model which describes how neuronal populations generate the activity of the large brain regions that are source reconstructed using MEG. Here, we employed the widely used neural-mass model developed by David and colleagues[86]. This model characterises large brain regions by averaging the post-synaptic membrane depolarisation of different neuronal populations, which are interconnected in a vertical pattern, mimicking the canonical cortical microcircuit (CMC) structure outlined by Bastos and colleagues[89]. Then, the neural-mass model is integrated with a network architecture indicating the hypothesised feedforward and feedback connections between the brain regions that were identified from the MEG data. Finally, the DCM is fitted to the time series of those brain regions and the marginal likelihood of the model (model evidence), as well as the posterior and protected exceedance probabilities, are estimated. Typically, a first model representing the hypothesised network architecture/brain hierarchy is formulated, and its evidence is compared to the evidence of a few alternative, competing models. This is done to establish which is the network

architecture which shows the highest model evidence given the experimental data.

DCM is a popular and established method and extensive details on its functioning can be found in multiple experimental, theoretical and review papers[37,38,85–88]. Additional details on the Bayesian model comparison are available in the next section.

In this study, DCM was used to test our hypothesised model of brain hierarchies during recognition of memorised and varied musical sequences against five competing models (Fig. 6a).

These six models comprised the six AAL ROIs showed in Supplementary Fig. S8 and illustrated in the previous sections of the paper (LHG, RHG, LHP, RHP, ACC, MC). The models are described as follows:

1. Model 1: this is the model that we hypothesised would have the highest evidence and consisted of feedforward connections from LHG, RHG to LHP, RHP, ACC, MC, and feedback connections from LHP, RHP, ACC, MC to LHG, RHG.

2. Model 2: this alternative model consisted of feedforward connections from LHG, RHG to LHP, RHP, ACC, MC. No feedback connections were indicated. This model served to evaluate if there was evidence for a loop of feedforward and feedback connectivity between the auditory cortices and the other brain regions or if only feedforward connectivity was the most likely explanation of the data.

3. Model 3: this alternative model consisted of feedforward connections from LHG, RHG to LHP, RHP and from LHP, RHP to ACC, MC. The model also hypothesised feedback connections from ACC, MC to LHP, RHP and from LHP, RHP to LHG, RHG. This alternative model distinguished HP from ACC and MC. This model was formulated based on the anatomical contiguity of ACC and MC[90], as opposed to HP and wished to establish if there was an additional hierarchical layer between LHP, RHP and ACC, MC.

4. Model 4: this alternative model consisted of feedforward connections from LHG, RHG to ACC, MC and from ACC, MC to LHP, RHP. The model also hypothesised feedback connections from LHP, RHP to ACC, MC and from ACC, MC to LHG, RHG. This model was formulated with the same rationale behind model 3, but it tested the opposite direction of the connections between LHP, RHP and ACC, MC.

5. Model 5: this alternative model consisted of feedforward connections from LHG, RHG to MC and from MC to ACC, LHP, RHP. The model also hypothesised feedback connections from ACC, LHP, RHP to MC and from MC to LHG, RHG. This alternative model distinguished HP and ACC from MC. This model was formulated based on the higher correlations between the time series of ACC, LHP, RHP ($r_{ACC\text{-}LHP} = 0.542$; $r_{ACC\text{-}RHP} = 0.731$; $r_{LHP\text{-}RHP} = 0.486$) compared to the correlations between MC and ACC ($r = -0.565$), LHP ($r = 0.256$), and RHP ($-0.163$). It aimed to establish if there was an additional hierarchical layer between MC and LHP, RHP, ACC.

6. Model 6: this alternative model consisted of feedforward connections from LHG, RHG to ACC, LHP, RHP and from ACC, LHP, RHP to MC. The model also hypothesised feedback connections from MC to ACC, LHP, RHP and from ACC, LHP, RHP to LHG, RHG. This model was formulated with the same rationale behind model 5, but it tested the opposite direction of the connections between LHP, RHP, ACC and MC.

Based on the extensive evidence that the auditory cortices are the first brain regions to respond to sounds in the context of ERP/Fs[44,45], all the models hypothesised that the external auditory inputs reached LHG and RHG.

We aimed to test the network architecture of this array of brain regions for both recognition of previously memorised and novel musical sequences [i.e. for conditions M and N(s)]. Since DCM for MEEG is usually computed on relatively short time windows (e.g. 350–400 ms) and for one experimental perturbation at a time (e.g. one sound)[37,38,85–88], we computed a series of DCM analyses independently for each tone of the sequences, in a 350 ms time window from each tone onset. Specifically, we computed an independent DCM for tones two, three, four and five for condition M. Tone one was excluded because it was the same across all conditions.

In relation to the N conditions, we were interested only in the tone that introduced the variation in the sequences. Thus, we computed four DCMs, organised as follows: NT1 for tone two; NT2 for tone three; NT3 for tone four; NT4 for tone five. In conclusion, we computed a total of eight DCM analyses, always testing the same alternative models since we hypothesised the same hierarchy in the network architecture of M and Ns.

As a final note, following standard DCM procedures, the DCMs were estimated independently for each participant. The consistency of the results across participants was tested using random-effects Bayesian model selection (RFX BMS), which is described in detail in the following section.

## Bayesian model comparison across participants

Dynamic causal models are estimated with variational Laplace[91]. The variational Laplace is used for deriving both the posterior distribution and the marginal likelihood of the model itself, which is also known as the model evidence. Employing a Laplace approximation to the multivariate posterior distribution of model parameters, denoted as $q(\theta|y,m)$, we iteratively computed conditional means and covariances by maximising a lower bound on the logarithm of the model evidence, referred to as the log-evidence.

This optimisation utilises the Newton's method and a Fisher scoring scheme to maximise the (negative) variational free-energy F of the model (5):

$$F = E_q[\ln p(y|\theta,m)] - D_{KL}[q(\theta|y,m)||p(\theta|m)] \tag{5}$$

Where $E_q[\cdot]$ indicates the expectation under the variational posterior density $q(\theta|y,m)$, while $D_{KL}[\cdot]$ refers to the relative entropy [or *Kullback–Leibner (KL)* divergence]. This makes the free energy a lower-bound approximation to the log-evidence.

To gain a deeper understanding of the utility of free energy in model comparison, we can break it down into two components. The initial component comprises the anticipated log-likelihood of the data, denoting the model's precision. The subsequent component is the relative entropy, or KL divergence, between the multivariate posterior and prior probability distributions, signifying the model's complexity. This formulation of complexity as a KL divergence is based on the premise that the posterior need not deviate significantly from the prior to accommodate new data. In essence, a proficient model should yield posterior beliefs consistent with its prior beliefs when exposed to new data. Consequently, free energy can be dissected into accuracy minus complexity, jointly encapsulating a model's ability to explain the data. Our set of alternative models were then compared using their free energy, which indexes each model evidence[87]. To study the consistency of the free energy across the participants, we used a RFX BMS model selection procedure, which provides the posterior probability and protected exceedance probability of each model for model comparison within the entire sample of participants[88].

The posterior probability is determined using normalised Bayesian factors (BF) and reflects the likelihood of each model being independently a good fit based on the observed data. The protected exceedance probability describes the likelihood that a particular model is the best of explaining the data when compared to all the alternative models. In addition, by accounting for variations in model frequencies that could arise purely by chance[92], the protected exceedance probability keeps into account potential issues associated with multiple comparisons, enhancing the robustness of the results. Finally,

in addition to posterior probability and protected exceedance probability, we also calculated a statistical measure called the Bayesian omnibus risk (BOR). The BOR is a metric that quantifies the statistical risk associated with performing Bayesian model comparisons at a group level, taking into account potential variations in the likelihood of the models among different participants. In other words, the BOR indicates the likelihood that the protected exceedance probability observed in each model comparison is due to random chance within the participant sample. This parameter is a recent addition to the DCM framework and serves as a further tool to assess the reliability of the DCM results[37]. Taken together, the posterior probability, protected exceedance probability and BOR provide a state-of-the-art, conservative approach for determining which of the alternative DCM models has the strongest evidence within the entire population. Additionally, they offer a robust measure of the probability that this evidence is not simply a result of random chance[37,85,88,92].

This procedure was observed independently for the eight DCM analyses described in the previous section.

### Time-frequency analysis for induced responses

We computed a time-frequency analysis using complex Morlet wavelet transform (from 1 to 60 Hz with 1-Hz intervals)[93]. This analysis was conducted for induced responses, independently for the six AAL ROIs and for the four contrasts considered in this study (i.e., M versus NT1, M versus NT2, M versus NT3, M versus NT4). Specifically, the time-frequency decomposition was done independently for each trial, brain voxel, and participant. Moreover, baseline correction was applied by subtracting, for each frequency, the average power in the baseline interval from the power at all time points[39,94,95].

Then, as shown in Fig. 7 and Supplementary Fig. S12, the power spectrum of each trial and each brain voxel was averaged within each of the six ROIs. In addition, to provide results which can be more easily compared with previous literature, we have also computed the power spectrum for left and right occipital superior lobe. Finally, we performed a two-sided t-test for each frequency and time point, making four contrasts: M versus NT1, M versus NT2, M versus NT3, M versus NT4. The emerging $p$-values were binarized ($\alpha = 0.05$) and then submitted to a 2D MCS (MCS $p$-value = 0.001). Here, we calculated the clusters size of continuous significant values in time and frequency. Then, we made 1000 permutations of the binarized $p$-values. For each permutation, we measured the size of the maximum emerging cluster and built a reference distribution with one value for each permutation. Finally, the original clusters were considered significant when they were bigger than the 99.9% of the permuted data maximum cluster sizes.

To strengthen the reliability of our results in MEG source space, we computed the same analysis for eight fronto-temporal and occipital MEG channels. In this case, the power spectra were computed independently for each trial and then averaged (Supplementary Fig. S13 and Supplementary Data 12).

### Reporting summary

Further information on research design is available in the Nature Portfolio Reporting Summary linked to this article.

## Data availability

The pre-processed neuroimaging data generated in this study have been deposited in the Zenodo database under accession code [https://doi.org/10.5281/zenodo.10715160][96] and are publicly available. The raw neuroimaging data are protected and are not available due to data privacy laws. Source data exceeding 30MB as well as Supplementary Figs. in high resolution are provided in the same Zenodo database reported above. Source data are provided with this paper.

## Code availability

The MEG data was first pre-processed using MaxFilter 2.2.15. Then, the data was further pre-processed using Matlab R2016a or later versions (MathWorks, Natick, Massachusetts, United States of America). Specifically, we used codes from the Oxford Centre for Human Brain Activity Software Library (OSL), FMRIB Software Library (FSL) 6.0, SPM12 and Fieldtrip. To compute multivariate pattern analysis (decoding), we employed support vector machines (SVMs) (libsvm:http://www.csie.ntu.edu.tw/~cjlin/libsvm/). In-house-built code and functions used in this study are part of the LBPD repository which is available at the following link: https://doi.org/10.5281/zenodo.10701724[97]. The full analysis pipeline used in this study and the code employed for delivering the stimuli are available at the following link: https://doi.org/10.5281/zenodo.11072410[98].

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

## Acknowledgements

The Centre for Music in the Brain (MIB) is funded by the Danish National Research Foundation (project number DNRF117). L.B. is supported by Lundbeck Foundation (Talent Prize 2022), Carlsberg Foundation (CF20-0239), Centre for Music in the Brain, Linacre College of the University of Oxford (Lucy Halsall fund), Society for Education and Music Psychology (SEMPRE's 50th Anniversary Awards Scheme), and Nordic Mensa Fund. M.L.K. is supported by Centre for Music in the Brain and Centre for Eudaimonia and Human Flourishing, which is funded by the Pettit and Carlsberg Foundations. Additionally, we thank Fundación Mutua Madrileña (Mutua Madrileña Foundation, Madrid, Spain) for the support provided to G.F.R. and Mensa: The International High IQ Society (Italian section) for the support provided to F.C.

## Author contributions

L.B., G.F.R. and M.L.K. conceived the hypotheses. L.B. designed the study. L.B., M.L.K. and P.V. recruited the resources for the experiment. L.B., G.F.R. and F.C. collected the data. L.B. and G.F.R. performed pre-processing, statistical analysis and modelling. D.P. provided MATLAB codes for decoding analysis. M.D. provided initial MATLAB codes and assistance for the DCM analysis. D.P., M.D., M.L.K., G.F.R. and P.V. provided essential help to interpret and frame the results within the neuroscientific literature. L.B. and G.F.R. wrote the first draft of the manuscript. L.B., G.F.R., F.C. and M.L.K. prepared the figures. All the authors contributed to and approved the final version of the manuscript.

## Competing interests

The authors declare no competing interests.
