## [Peer Review File · Nature Communications]

Spatiotemporal brain hierarchies of auditory memory recognition and predictive codingReviewer #1 (Remarks to the Author):

I attached a PDF with the same comments but better formatting.

The current manuscript "Revealing the spacetime hierarchical whole-brain dynamics of auditory predictive coding" reports an experiment that compares recognizing and classifying short musical sequences (5 notes, 1750 ms) as known (i.e., encountered during the familiarization phase of the experiment) or not. The manuscript is very well written (well-structured, clear and concise), includes very clear and illustrative figures, and uses a wealth of different (often advanced) methods. The main achievement is that a model of a brain network underlying the recognition of musical sequences and a time course in which different processing steps take place within that model is derived from the results of the experiment.

- What are the noteworthy results?

The reported data are behavioural (recognition performance, correct classification as known or unknown, as well as reaction time), and neurophysiological responses that are evaluated in several ways (ERFs, source location, capacity to use the neurophysiological response for decoding, frequency signature). The succession of time windows and the neural generators are used to propose a model of a brain network underlying the recognition of musical sequences (Figure 4).

- Will the work be of significance to the field and related fields? How does it compare to the established literature? If the work is not original, please provide relevant references.

It definitely will. It provides an overview over the different processing steps during the recognition of musical sequences. It thereby sits halfway between the more general models of music perception (e.g., Koelsch, 2011), and those focussing on the free energy principle and mainly on predicting the next element in a sequence (e.g., Koelsch et al., 2019; Friston & Friston, 2013). It also characterizes the different steps quite clearly in terms of time window / underlying ERPs / ERFs, frequency characteristics and likely neural generators. It thereby can become a stepping stone into deriving hypotheses and exploring further aspects of music recognition (the authors, e.g., discuss how the beat might have affected the results for NT4).

- Does the work support the conclusions and claims, or is additional evidence needed?

Generally, hypotheses and results align very well. There is, in my view, no need for additional evidence. I named two limitations below; however, they rather need some re-analysis or discussion rather than additional evidence.

- Are there any flaws in the data analysis, interpretation, and conclusions?

Generally yes. However, there are two concerns:

(1) The changes that are made to the sequences are described in the article as purely local. However, Western tonal music on which the sequences are based also relies on harmonic regularities (applying to a musical sequence as a whole). If the creating of the stimulus preserved the harmonic structure, it should be named in the methods. If not, it has to be discussed as a limitation. Given that the results align quite well with the hypotheses, it's perhaps enough to mention that there is another level of hierarchy that likely also plays a role in recognizing musical sequences (or sth. alike).

(2) The choice of ROIs (see comment below) seems to be a bit arbitrary. Having the ROIs based on an established atlas / parcellation would make the claims regarding the brain network underlying the recognition of musical sequences stronger.

- Is the methodology sound? Does the work meet the expected standards in your field?

Altogether, the methodology appears sound. It includes a couple of advanced methods (e.g., the SVM). However, there are some aspects of the methodology that were either not clearly enough described or where the choice of settings (thresholds, etc.) was not clear:

(1) The Monte-Carlo-Simulations could be described in more detail. It is not perfectly clear what the values for α and p exactly represent (and, if p represents one in a thousand, the formulation "bigger than the 99.99%" is wrong; one in a thousand is 99.9%). α appears to be the value used for dichotomization, make that more explicit (or write what it is if I got it wrong). Another thing that escaped me was why different α were chosen? This should be more clearly justified.

(2) Which voxels were chosen to be included in the ROIs and why remains quite vague. For example, the ROIs for the auditory cortex extend over all gyri of the temporal lobe and seem partly to overlap (or at least be adjacent to) hippocampal regions and inferior temporal cortex.

Wouldn't it be better to rely on an established parcellation (e.g., Desikan-Killiany)? This also would make the argument regarding which brain network underlies the recognition of musical sequences stronger.

- Is there enough detail provided in the methods for the work to be reproduced?

Generally yes. However, some parts of the methods-description remain unclear:

- (1) The description says that "participants listened to a short (approximately 25 seconds long) musical piece twice" and afterwards describes that the "musical piece consisted of the first four measures of the right-hand part of Johann Sebastian Bach's Prelude No. 2 in C Minor, BWV 847". The description is ambiguous: four measures should be much shorter than 25 sec?
- (2) There seems to be a mix-up in Figure S1 containing all the sequences used in the study: (a) part 2 and 3 are switched (part 2 should be on p. 39, part 3 on p. 40); (b) the occasionally appearing 2/4 (time signature in some bars) is also confusing and should be removed.

Minor comments:

- Title: Consider adding "musical sequences" in the title.
- Table 2: there are two occurrences that deviate from the typical "pattern" of results: ACL for M vs. NT3, 1.45 – 1.55 s, and ACL for M vs. NT4, 1.75 – 1.85 s. They should perhaps be discussed.
- Table 3: Please provide both the time relative to the onset of the first tone of the sequence as well as the time relative to the onset of the deviation. This would make comparing between the deviations (NT1, ..., NT4) much easier.
- Figure 3: Please describe more exactly how the neural activity index was calculated. I am also wondering whether applying a low-pass filter would make the figure more easy to read?
- Figure 4c: I was wondering whether some form of significance threshold could be shown (e.g., a grey box from where a certain value of rho would become significant). I also think that now the direction of the cross-correlation is VLPFC → AC (since the peak is before the 0 on the time axis). Having it the other way round is perhaps easier to understand?
- l. 335: Define which frequency range was regarded as gamma. I am also wondering whether putting lines separating the different frequency bands in Figure 5 would help or rather make the figure more difficult to read.
- l. 357: the results ... expand (no s)
- l. 363: These peaks – It is unclear whether both the negative and the positive component are meant or the positive component in the different regions.
- l. 529: Their educational background was homogeneous. Be more explicit: What exactly was their education level?
- l. 607: Justify a bit why there was a downsampling by factor 4. Why were 1000 Hz recorded in the first place? And why wasn't that sampling rate kept?
- l. 621: Describe whether the selection of components for rejection was manual or automatized. If the former, describe the criteria for rejection, if the latter describe which package was used with which parameters.
- l. 724: was associated (not association)
- l. 728 - 731: The description is quite unclear: I guess the first part aims to describe a difference in neural activity between the conditions being either positive or negative? But isn't that passage about defining time windows: i.e., it should be mentioned that while the description uses x vs. $x + 1$, it is actually about how long a difference with a particular sign extends that is most relevant here? Also: Try to describe more clearly how looking at the results from decoding and looking at the corresponding brain activity was implemented.

Reviewer #2 (Remarks to the Author):

The authors used MEG to measure oscillatory activity associated with the explicit recognition of changes in a memorised 5-tone sequence. Multivariate pattern analysis is used on MEG data to define region in which the time series differ between the memorised stimulus and altered sequence at positions beyond the first. Analyses within regions were based on evoked response and time-frequency analysis. A hierarchy of areas is suggested based on higher order areas that send predictions to auditory cortex.

1. The study is a very carefully conducted description of the brain activity corresponding to the encoding of melodies into episodic memory (although the authors' term acme might be a little

hyperbolic).

2. I would like to have seen a more explicit description of a hypothesis. The study is couched in terms of hierarchical organisation of sequence processing but the predictive coding model is not explicitly tested, and if you do generate predictions I am not sure that they fit. For example, a fundamental aspect of predictive coding is the generation of predictions in higher areas and prediction error in sensory cortex when these are compared with incoming signal (note the prediction and prediction error arrows in figure 4 are the wrong way around). Classical predictive coding is based on a canonical coding scheme in which prediction errors are encoded as a gamma signal but the data here show increased gamma in response to the memorised melody. Does that fit with a predictive coding model?

3. I accept classical predictive coding has been applied to perception (and sometimes working memory) as opposed to episodic memory as here, but the authors might elaborate on how well their finding fits with predictive coding which readers will expect when they read the title.

4. The network in figure 4 raises questions (apart from the typo mentioned above) that I have listed.

a. I accept that inferior temporal cortex and hippocampal cortex are bundled together because of similar time series but they are different places with different types of connections to auditory cortex from the two (more indirect from hippocampus)

b. There is no analysis to examine effective connectivity between areas which I think is needed in a study that examines predictive coding. There are so many ways to do this including spectral Granger which would allow examination of key oscillatory correlates in different bands. I am not expecting such analyses between all the nodes but between key nodes like hippocampus and AC

c. In terms of the higher areas implicated here hippocampus is expected. Where is the inferior frontal gyrus? This area is implicated in large numbers of studies of melody analysis going back to fMRI studies of working memory for melody by Zatorre in 1990s. I guess the analysis here seeks differences between encoded and new which may be less critical in IFG compared to HC but this does merit comment.

Reviewer #3 (Remarks to the Author):

Revealing the spacetime hierarchical whole-brain dynamics of auditory predictive coding

Summary:

Bonetti and colleagues capitalize on their previous work and reuse a paradigm where participants have to memorize a melody (5 notes long) from a real piece (Bach's prelude in C minor, BWV847), and then distinguish them from variants of that melody in an old/new task. This paradigm allows them to investigate the predictive coding framework, by comparing the memorized (predicted) and novel (unpredicted) melodies. The novelty of the paper is to be found in the refinement of the paradigm, with systematic variations of the melody (starting from the 2nd, 3rd, 4th or 5th note) and increased time between notes. The authors analyze the brain activity of 83 participants recorded with MEG and claim to observe a network of brain regions involved in the task, as well as a hierarchy of processing through this network.

Overall Comments:

The paradigm used in this study seems efficient at eliciting differentiated brain responses to memorized vs novel melodies at the level of individual notes, and the decoding analyses displayed at the beginning of the result section are convincing.

However, missing information in subsequent analyses (in text, figures and/or method, particularly when defining regions of interest), combined with errors in graph legends and labels, make evaluation and reproduction impossible.

The two main claims of the paper are:

1. A broad network of regions involved in the musical task, from auditory to hippocampal and frontal regions. However, this network is not supported by connectivity analyses (Figure 4 is misleading in this respect), and relies solely on a functional ROI definition that is not properly documented in the manuscript.

2. A hierarchy of processing within this network. However, this is only supported by descriptive statistics showing a fixed delay in response between the left auditory cortex and the ventro-medial prefrontal cortex (2 of the 6 ROIs defined in the paper). The observation itself is not especially novel and gives very little support to the claim of a processing hierarchy.

Finally, the introduction falls short at explaining the rationale behind the paradigm used, and the discussion is filled with interpretations that are based either on descriptive statistics or on analyses that are not even mentioned in the results or methods sections.

This paper could be drastically improved by:

- reworking the figures, legend and method section, so that no information is missing for potential readers to assess the quality of the analyses or to perform a replication.
- making sure that inferential statistical testing is performed for all the claims and interpretations the authors are willing to make, as these are what allows us to generalize our findings to future observations.
- targeting more precisely what are the novel findings in light of the existing literature.

Major Comments:

- The introduction clearly states that the upgrades of this paradigm are the systematic variations of the melody and the 350 ms delay between notes. While the 350 ms delay is said to improve the results by avoiding a superposition of auditory responses, it is not clear why the systematic variations of melody were done this way, what the authors expected to see from that modification of the paradigm, and what it did bring to the MEG results compared to previous studies.
- In the introduction, I would suggest avoiding to justify the present work with sentences like "little is known about " (p.3 l.50) or "much remains to be learned" (p3. l.57). Stating that little is known about something is not sufficient to explain why you decided to study a phenomenon ("there are so many phenomena that we know little about, why did you choose to focus on this one?"). More generally, the introduction falls short at explaining the stakes of this work.
- In the section "Neural sources of the differential brain activity between M and N", it reads "we considered [...] the inspection of the MEG data after preprocessing", but this inspection is not described in the main text nor in the method. Even if some steps of analysis rely on visual inspection, it has to be stated clearly for transparency and replicability.

In the same paragraph, it is unclear where the temporal windows come from. For instance, the window "1.75-1.85 sec" does not correspond to a significant temporal cluster, in particular in Fig. 2.b.

Because of this lack of information, it is impossible to assess whether the analysis is valid or not.

- In the method section "Functional regions of interests (ROIs)", there is not enough information to understand how the ROIs are defined. First, the procedure here builds upon the spatial clusters from the previous section. However, the 14 clusters reported in table 2 are now grouped into 4 "broad regions" without any explanations. Second, it reads that "t-values [were computed] for each brain voxel and each timepoint contrasting M versus N" (p.31 l.752) without specifying which N condition (from NT1 to NT4) was used or how they were grouped. Finally, assuming that the peaks of t-values of each of the 4 broad regions were defined temporally (and not spatially), it would have been useful to report the timing of these peaks (do they correspond to the initial auditory burst? or to the response to a specific note onset? or an activity at a timing unrelated to auditory responses?).
- The results in Table 3 show a stronger activity for M vs N (i.e. $M > N$) for most ROIs, except in the cingulate cortex. This result seems contradictory with the predictive coding hypothesis (the responses to surprising notes in N are expected to show a greater amplitude than in M), and with the interpretation of the results shown in Fig. 3 (for instance: "late N100 responses localized in AC were stronger for N versus M", p.12 l.266). This should be addressed in the discussion.

• In Fig. 3:

- The y-axis label "Neural activity index" is not described (in legend, main text or method). This is concerning, since it is one of the main results of the paper.
- In the legend, the M and NT1 curves are described as "M vs NT1" and "NT1 vs M" contrasts. Shouldn't these two curves be symmetrical, or similar in some ways? Or is it simply a wrong

legend?

- The timing of the "late N100" mentioned in the text is not clearly shown on the figure. Plotting vertical lines at the exact timing of not onsets may help.
- All regions of interest display an initial auditory burst after the first note onset. The fact that it is observed in all ROIs (even the cingulate cortex) raises suspicions about the source reconstruction procedure.

• In Fig. 4:

• While I appreciate the visual clarity of Fig. 4.a, the "brain network" shown is not supported by any connectivity measures and is purely descriptive. Moreover, the "high-order regions" and "low-order regions" labels are interpretative, and not derived from data nor analyses. Finally, since the ROIs may not be defined properly, the only part of this network that is based on data may be compromised. This is especially important, as one of the main claims of the paper is based on that figure.

• Similarly, Fig. 4.b and 4.c are only descriptive, and do not show any inferential statistics to support the finding of a temporal hierarchy of processing. Moreover, the fixed delay in response to surprising notes is shown only between left AC and VMPFC (only 2 of the 6 ROIs displayed in the network in Fig. 4.a). It would have been relevant to replicate the analysis along the connections hypothesized above (and at the very least with right AC and VMPFC). Finally, the authors say that the "strength of the signal increases over time for VMPFC" in the legend and in the discussion (p.18, l.368), but this not visible from the figure and not supported by any analysis (descriptive or inferential). Again, this is especially important, as the second main claims of the paper is based on that figure.

Finally, a delay in response between "low" and "high" order regions of the auditory system (i.e. from auditory to frontal) is not a novel finding (for instance, see Fig. 7 from Rocchi et al., 2021).

• In Fig. 5, the time-frequency analysis shows similar response patterns in all ROIs, which again casts some doubts on the validity of the source reconstruction procedure. Furthermore, the figures do not show which data points are significant or not (or all the points are significant?), which is not acceptable.

• In the discussion, many interpretations are based on descriptive results, or on results that are not shown in the result section:

• the claim of "a rapid hierarchical pathway progressing from the auditory cortex to the ventromedial prefrontal cortex" (p.18 l.364) and that there is "information flowing from auditory cortex to ventromedial prefrontal cortex and hippocampal regions" (p.21, l.455) should be based on directed connectivity measures.

• the claim that "the later responses in the ventromedial prefrontal cortex and hippocampus occurred only after the tone that disrupted the original sequences" (p.18 l.374) could be explicitly tested by grouping the responses to notes in M, grouping the first novel note in N, and looking at the contrast between them.

• the claim that "The strength of the responses was progressively greater depending on the tone used to introduce the variation" (p.18, l.378) also has to be empirically tested.

• the claim that "... we observed a distinct, faster response to each tone of the sequence. The frequency of these responses was of approximately 1 – 1.5 Hz" (p.19, l.403)" is also not supported by any analysis, even descriptive.

Minor Comments:

• There is no information about the musical background of the participants. If this data is not available for some reason, it would be preferable to state it clearly rather than omitting it.

• For the sake of clarity, I would suggest reducing the size of the figure's legends by removing all the interpretation of the results that they contain. There are also some mistakes to correct (for instance: wrong colors described in Fig. 3).

• Before showing the MVPA results in Fig. 2, and to help the reader getting a better idea of what the data look like, I would suggest to show some simple univariate analysis, or at least in the supplementary figures (e.g. a topographic map of the general auditory response to all conditions, or simple contrasts between conditions at the sensor level).

• The temporal generalization of the decoding analysis is interesting but not mentioned in the discussion. For an example of interpretation, see King & Dehaene (2014).

• In the section "Neural sources of the differential brain activity between M and N", the table 2 is

hard to read. Plotting the clusters on a 3D brain (grouped by time window or by contrast) may help.

- Although it is good to be enthusiastic and positive about one's results, I would suggest avoiding wording like "This study represents the acme of our recent investigations", and any sentence written to highlight your work without reporting facts or interpretations.

Bibliography:

- King, J. R., & Dehaene, S. (2014). Characterizing the dynamics of mental representations: the temporal generalization method. *Trends in cognitive sciences*, 18(4), 203-210.
- Rocchi, F., Oya, H., Balezeau, F., Billig, A. J., Kocsis, Z., Jenison, R. L., ... & Petkov, C. I. (2021). Common fronto-temporal effective connectivity in humans and monkeys. *Neuron*, 109(5), 852-868.

REVIEWER COMMENTS

Reviewer #1 (Remarks to the Author):

The current manuscript “Revealing the spacetime hierarchical whole-brain dynamics of auditory predictive coding” reports an experiment that compares recognizing and classifying short musical sequences (5 notes, 1750 ms) as known (i.e., encountered during the familiarization phase of the experiment) or not. The manuscript is very well written (well-structured, clear and concise), includes very clear and illustrative figures, and uses a wealth of different (often advanced) methods. The main achievement is that a model of a brain network underlying the recognition of musical sequences and a time course in which different processing steps take place within that model is derived from the results of the experiment.

We sincerely appreciate your kind words and the invaluable insights provided in your thorough review. Your constructive feedback has significantly contributed to enhancing the quality and clarity of our manuscript. In response to your thoughtful comments, the following key revisions have been implemented:

1. *Preservation of harmonic structure in the musical stimuli:*
 - a. We have provided detailed clarification on the careful preservation of harmonic structure when creating the stimuli.
 - b. We have suggested that future studies could explore additional types of variations, including changes in harmonic structure.
2. *Definition of ROIs:*
 - a. We have conducted additional analyses using the automated anatomical labelling (AAL) parcellation, aligning with your suggestion.
 - b. We have presented the results based on AAL parcellation in the revised manuscript, reinforcing the foundation of our claims. Moreover, we have reported in supplementary materials the results related to the functional parcellation presented in the original manuscript. This allows full transparency and provides the readers with additional, relevant information.
3. *Monte-Carlo-Simulations (MCS) clarification:*
 - a. We have provided a detailed description of the MCS procedure, addressing concerns about α and p values.
 - b. We have emphasised the use of a consistent α value and corrected a typo in percentage.
4. *Methodology soundness:*
 - a. We have clarified the rationale for default sampling rate and downsampling.
 - b. We have provided additional details on ICA for artifact rejection, ensuring transparency.
5. *Brain hierarchies and Dynamic Causal Modelling (DCM):*
 - a. We have conducted DCM analyses to support claims of hierarchical processing in the brain.
 - b. To this aim, we have included a new author (Ass. Prof. Martin Dietz) who has extensive DCM expertise. With his help, we were able to provide quantitative evidence of hierarchical organisation of the brain during long-term recognition of musical sequences.
6. *Presentation and Clarity:*
 - a. We have revised figures, tables, and descriptions based on your suggestions for improved clarity.

Our extensive additional clarifications and analyses were complemented by 50 novel panels distributed across 18 supplementary figures and by 15 supplementary tables, providing a better understanding of every detail of our procedures.

Please find detailed responses to each of your comments below in this letter. To facilitate your review, we have reported in this letter the revised sections/paragraphs of the manuscript in relation to your main comments. For minor comments, we have indicated only the line of the revised manuscript where changes were made to avoid unnecessary length in this letter.

The fully revised figures are provided in the revised manuscript and in the supplementary materials. Moreover, they are also attached to the submission in high definition.

We are confident that these revisions have strengthened the manuscript, addressing all your valuable recommendations. Thank you once again for your time and expertise in reviewing our work.

Detailed answers

- What are the noteworthy results?

The reported data are behavioural (recognition performance, correct classification as known or unknown, as well as reaction time), and neurophysiological responses that are evaluated in several ways (ERFs, source location, capacity to use the neurophysiological response for decoding, frequency signature). The succession of time windows and the neural generators are used to propose a model of a brain network underlying the recognition of musical sequences (Figure 4).

- Will the work be of significance to the field and related fields? How does it compare to the established literature? If the work is not original, please provide relevant references. It definitely will. It provides an overview over the different processing steps during the recognition of musical sequences. It thereby sits halfway between the more general models of music perception (e.g., Koelsch, 2011), and those focussing on the free energy principle and mainly on predicting the next element in a sequence (e.g., Koelsch et al., 2019; Friston & Friston, 2013). It also characterizes the different steps quite clearly in terms of time window / underlying ERPs / ERFs, frequency characteristics and likely neural generators. It thereby can become a stepping stone into deriving hypotheses and exploring further aspects of music recognition (the authors, e.g., discuss how the beat might have affected the results for NT4).

We are very grateful for your evaluation of our work and for providing these excellent references. We have made sure to properly cite and discuss them in light of our results (e.g. line 54):

“Aiming to bridge this gap, prior investigations centred on automatic predictive processes framed within PCT. These studies assessed auditory automatic prediction error, often relying on well-established event-related potential/field (ERP/F) components such as N100, mismatch negativity (MMN), and error-related negativity (ERAN) ¹⁻⁹. They demonstrated that such components were automatically evoked in response to auditory stimuli, deviations from expected sound features, likelihood of occurrence of musical tones (N100, MMN), and varied harmonic properties (ERAN)”.

Line 579:

“Furthermore, here we introduced systematic variations to the original memorised sequences to examine the brain mechanisms associated with conscious prediction error. The declarative nature of the prediction error, where participants were asked to identify differences in the sequences and provide a conscious response, extends our understanding beyond the existing body of research focused on automatic prediction error indexed by well-known ERP/Fs components such as N100, MMN and ERAN ^{4, 5, 7-9}. This research revealed that such components were primarily generated by the auditory cortex ^{10, 11}, reporting a complementary yet much reduced recruitment of the medial cingulate gyrus, inferior frontal gyrus, and frontal and hippocampal regions ^{2, 11}”

Line 599:

“Our results fit the well-known PCT ¹²⁻¹⁴, which states that the brain is constantly updating internal models to predict external information and stimuli. Recently, this theory has been proposed to explain the brain mechanisms underlying complex cognitive processes ¹⁵, finding a remarkable example in the neuroscience of music ^{13, 14}. However, while PCT holds promise for understanding complex cognitive processes, the quantitative evidence in its favour has been limited. Thus, our study filled this gap by providing quantitative evidence of the brain functional hierarchies during long-term recognition of previously memorised and novel sequences”.

- Does the work support the conclusions and claims, or is additional evidence needed? Generally, hypotheses and results align very well. There is, in my view, no need for additional evidence. I named two limitations below; however, they rather need some re-analysis or discussion rather than additional evidence.

As you can see below, following your suggestions, we have improved the analyses as well as the discussion of our findings.

- Are there any flaws in the data analysis, interpretation, and conclusions?

Generally yes.

However, there are two concerns:

(1) The changes that are made to the sequences are described in the article as purely local. However, Western tonal music on which the sequences are based also relies on harmonic regularities (applying to a musical sequence as a whole). If the creating of the stimulus preserved the harmonic structure, it should be named in the methods. If not, it has to be discussed as a limitation. Given that the results align quite well with the hypotheses, it's perhaps enough to mention that there is another level of hierarchy that likely also plays a role in recognizing musical sequences (or sth. alike).

We appreciate your insightful point, and we share your view on the significance of preserving the harmonic structure.

When creating the stimuli for both the 'memorised' and 'novel' conditions, we took great care in controlling various factors, including the harmonic structure, melodic contour, intervals, number of tones, dynamics, and pitch range of the sequences. Indeed, in most cases, when creating the N sequences, we maintained the same harmonic structure of the original M sequence. The only exceptions occurred in a few instances of 'inverted melodic contours' and 'scrambled intervals', where, to control for other musical features, such as melodic contour and intervals, the harmonic structure could not be always preserved. However, these were only a very few cases, as shown in **Figure S2**.

Thus, we can confidently state that the harmonic structure should not pose a confounding factor in our results. Moreover, inspired by your comment, we have suggested that future studies may use different categories of variations including changes in the harmonic structure of the melodies.

We have clarified this in the revised manuscript (line 757):

"To be noted, the harmonic structure of the N sequences with regards to the original M sequences was preserved in most of the cases. The only exceptions occurred for a few instances of the 'inverted melodic contours' and 'scrambling intervals' strategies, which were however important to control for other musical features such as melodic contour and intervals. This strategy was implemented to avoid potential confounding variables stemming from changes in harmony. However, investigating harmonic alterations concerning conscious brain prediction errors could prove relevant for future studies centred on diverse types of prediction errors".

(2) The choice of ROIs (see comment below) seems to be a bit arbitrary. Having the ROIs based on an established atlas / parcellation would make the claims regarding the brain network underlying the recognition of musical sequences stronger.

This is an excellent point. We agree, providing results using an established parcellation is a very good idea. Please, find our detailed answer in response to your comment below (after the MCS).

• Is the methodology sound? Does the work meet the expected standards in your field? Altogether, the methodology appears sound. It includes a couple of advanced methods (e.g., the SVM). However, there are some aspects of the methodology that were either not clearly enough described or were the choice of settings (thresholds, etc.) was not clear:

(1) The Monte-Carlo-Simulations could be described in more detail. It is not perfectly clear what the values for α and p exactly represent (and, if p represents one in a thousand, the formulation "bigger than the 99.99%" is wrong; one in a thousand is 99.9%). α appears to be the value used for dichotomization, make that more explicit (or write what it is if I got it wrong). Another thing that escaped me was why different α were chosen? This should be more clearly justified.

You are right, our description did not clearly explain what α and p were. In the following, we provide a better description of the MCS procedure.

After computing a series of statistical tests (e.g. a t-test for each time-point of two time series which were contrasted), we obtained a vector of values showing, for each time-point, whether the statistical test was significant or not. Then, we computed the clusters size of the continuous, binarized, significant values over time. The MCS $\alpha = .05$ means that, to binarize the values outputted by the t-tests that we described above, we used a *p-value* threshold of .05 (i.e. values below .05 were considered significant and inputted to the MCS as '1s' while values above .05 as '0s'). Thus, the first step of the MCS algorithm consisted of detecting the clusters of '1s'. Second, we made 1000 permutations of these binarized values (both '1s' and '0s'). For each permutation, we computed the size of the maximum emerging cluster of '1s' and built a reference distribution using these sizes. Finally, we considered significant the original clusters that were bigger than the 99.9% of the permuted data maximum cluster sizes (in other words, the MCS *p-value* = .001 refers to the percentage of the cluster sizes in the reference distribution that the original clusters should exceed to be considered significant). This procedure was observed for all the MCS described in the study. Moreover, additional details on this MCS algorithm can be found in previous papers ¹⁶⁻¹⁹.

As you rightfully pointed out, we agree that using different values of α is not a good practice. Thus, following your comment, we have always used the same value of $\alpha = .05$.

Thank you also for noticing our typo (99.99% instead of 99.9%). We have corrected it.

We have now better explained our procedure for the first MCS presented in the revised manuscript.

Line 861:

"Here, the significance was tested using a signed permutation test against the chance level (50%) for each time-point, as the previous analyses. Then, we corrected for multiple comparisons using two-dimensional (2D) cluster-based Monte-Carlo simulations (MCS, $\alpha = .05$, MCS *p-value* = .001). First, we computed the clusters size of the continuous, binarized, significant values over time. The MCS $\alpha = .05$ means that to binarize the values outputted by the signed permutation test against the chance level that we described above, we used a *p-value* threshold of .05 (i.e. values below .05 were considered significant and inputted to the MCS as '1s' while values above .05 as '0s'). Thus, the first step of the MCS algorithm consisted of detecting the clusters of '1s'. Second, we made 1000 permutations of these binarized values. For each permutation, we computed the size of the maximum emerging cluster and built a reference distribution using those values. Finally, we considered significant the original clusters that were bigger than the 99.9% of the permuted data maximum cluster sizes (in other words, the MCS *p-value* = .001 refers to the percentage of the cluster sizes in the reference distribution that the original clusters should exceed to be considered significant). This procedure was observed for all the analogous MCS described in the study. Moreover, additional details on this MCS algorithm can be found in ¹⁶⁻¹⁹".

(2) Which voxels were chosen to be included in the ROIs and why remains quite vague. For example, the ROIs for the auditory cortex extend over all gyri of the temporal lobe and seem partly to overlap (or at least be adjacent to) hippocampal regions and inferior temporal cortex. Wouldn't it be better to rely on an established parcellation (e.g., Desikan-Killiany)? This also would make the argument regarding which brain network underlies the recognition of musical sequences stronger.

Thank you for your insightful comment. We appreciate your suggestion and understand the concern regarding the vagueness in defining the regions of interest (ROIs) in our study. In our initial approach, we chose a functional parcellation based on the specific brain data obtained during our experimental task. This approach was aimed at avoiding any a priori assumptions about the brain regions involved. However, as you pointed out, our functional parcellation did indeed result in ROIs covering extensive areas, including some overlap or adjacency with hippocampal and inferior temporal cortex regions, particularly in the case of the auditory cortex ROIs.

In response to your suggestion, we conducted additional analyses using the Automated Anatomical Labelling (AAL) parcellation ²⁰, which is widely recognised and has been

employed in some of our previous MEG studies (e.g. ²¹⁻²⁴). This alternative approach allowed us to define ROIs with a more specific spatial extent.

Using the AAL ROIs, we repeated the same analyses described in the original version of the manuscript, which revealed significantly different brain activity between the experimental conditions across several AAL ROIs. Furthermore, we complemented these results with Dynamic Causal Modelling (DCM) analyses, providing quantitative evidence of the hierarchical structure among six key AAL ROIs that were particularly relevant to the task. These ROIs included left and right Heschl's gyri, left and right hippocampus, anterior cingulate gyrus, and medial cingulate gyrus.

These adjustments have not only enhanced the differentiation among the regions but also reinforced the robustness and reliability of our original findings. Your suggestion to employ an established parcellation has significantly improved the foundation of our claims and facilitated direct comparisons with existing literature.

In light of these outcomes, the revised version of the manuscript primarily reports the results based on the AAL parcellation. To ensure complete transparency, we have also provided comprehensive and additional information to elucidate how we derived the functional parcellation. However, this is now only reported in supplementary material.

As shown, for example, in **Figures 4** and **S11**, these two procedures showed convergent results, highlighting that the core significance of our findings does not depend on the chosen parcellation.

For your convenience, we report as follows the revised Results section (line 349):

“Automated anatomical labelling (AAL) time series

After evaluating the neural sources of the different brain activity associated with the experimental conditions using the fine-grained 8-mm parcellation of the brain (**Figure 1e**, “8-mm parcellation”), we focused on a set of anatomically defined regions of interest (ROIs) (**Figure 1e**, “AAL parcellation”). Here, we used a standard anatomical parcellation method known as automated anatomical labelling (AAL) ²⁰ and calculated a time series for each of the 90 non-cerebellar ROIs of AAL (**Figure 1f**, left, “Evoked responses”).

As specified above, our source reconstruction was conducted in an 8-mm space, resulting in 3559 brain voxels, each with its own time series. Thus, for each of the 90 ROIs, we identified the corresponding brain voxels and then averaged the time series of these associated voxels. This allowed us to obtain a final time series for each AAL ROI.

Then, we computed one t-test for each AAL ROI, each time-point and each combination of M versus Ns. Finally, we corrected for multiple comparisons using 1D cluster-based MCS (MCS, $\alpha = .05$, MCS *p-value* = .001). Confirming the results from the 8-mm brain parcellation shown in **Figure 3**, this analysis returned several significant clusters of differential brain activity over time between M and Ns. These were primarily localised in auditory cortices (e.g. Heschl's gyrus, superior temporal gyrus) hippocampus and para-hippocampal gyrus, inferior temporal cortex, anterior, medial and posterior cingulate gyrus, ventromedial prefrontal cortex. **Figure S7_{a-g}** and **Tables S6** and **S7** show all AAL ROIs and the significant differences between conditions. **Figures 4** and **S8** shows instead a selected array of ROIs which were particularly relevant for this study. Here, we identified the two ROIs that showed the maximum activity in absolute value among (i) auditory regions, (ii) medial temporal lobe, and (iii) cingulate and prefrontal cortices. We selected these broad regions based on the cognitive processes involved in the experimental task used in this study: (i) audition, (ii) memory, and (iii) evaluation and decision-making. As shown by **Figures 4** and **S8**, these ROIs were left and right Heschl's gyrus (LHG, RHG), left and right hippocampus (LHP, RHP), anterior cingulate gyrus (ACC, left and right averaged together) and medial cingulate gyrus (MC, ACC, left and right averaged together), respectively.

In essence, the contrasts showed that M versus N was characterized by stronger activity in RHG, LHP, RHP and ACC at 350 – 450 ms after the onset of each tone. Similarly, M presented stronger negative activity than N in the MC at 400 – 500 ms after the onset of each tone (**Figure 4**). Conversely, late N100 responses localised in LHG were stronger for N versus M. Moreover, LHP, RHP and ACC showed a stronger response for N versus M occurring at 250 – 300 ms after altering the original sequences (**Figure 4**). **Table 3** reports detailed statistics of the largest significant clusters, while complete statistical results of these six ROIs are described in **Table S7**.

Contrast	ROI	Temporal extent of the largest clusters from the 1st tone of the sequence	Peak t-value	P-value
Positive activity				
M versus NT1 (onset deviation NT1: 350 ms)	ACC	580 – 830	7.45	< .001
	LHG	520 – 630	7.17	< .001
	RHP	580 – 780	7.66	< .001
M versus NT2 (onset deviation NT2: 700 ms)	ACC	940 – 1160	7.48	< .001
	LHG	890 – 970	6.26	< .001
	RHP	950 – 1160	6.24	< .001
M versus NT3 (onset deviation NT3: 1005 ms)	ACC	1310 – 1540	6.13	< .001
	RHG	1660 – 1760	5.37	< .001
	RHP	1320 – 1520	6.72	< .001
M versus NT4 (onset deviation NT4: 1400 ms)	ACC	1700 – 1890	6.54	< .001
	LHG	1640 – 1720	6.58	< .001
	RHP	1680 – 1890	6.56	< .001
Negative activity				
M versus NT1	MC	680 – 860	-6.06	< .001
M versus NT2	MC	980 – 1180	-6.17	< .001
M versus NT3	MC	1360 – 1550	-5.96	< .001
M versus NT4	MC	1720 – 1920	-6.17	< .001

Table 3. Largest clusters of stronger activity of memorised (M) versus novel sequences (Ns).

Largest clusters of significantly stronger activity of M versus Ns computed for the six selected automated anatomical (AAL) regions of interest (ROIs): left Heschl's gyrus (LHG), right Heschl's gyrus (RHG), left hippocampus (LHP), right hippocampus (RHP), anterior cingulate gyrus (ACC), medial cingulate gyrus (MC). The table shows the contrast (including the onset of the first varied note in the N sequence), the correspondent ROI, the temporal extent (in ms) of the largest cluster, the peak t-value of the cluster and the associated Monte-Carlo simulations (MCS) p-value. The MC shows stronger negativity since the polarity of the MC signal was primarily negative, thus stronger activity in the MC for M versus N was indicated by a more pronounced negativity for M compared to N. All the other significant clusters, for both the selected six and the remaining AAL ROIs, are reported in detail in **Tables S6 and S7**.

In addition, we applied the multivariate source leakage correction method proposed by Colclough and colleagues²⁵ (see Methods for details) to the ROIs time series and computed the same statistical analysis described above. As shown in **Figure S9** and **Table S8**, the results aligned with our initial findings, thereby corroborating their robustness.

Finally, as a further measure to enhance the reliability of our findings, we reported in supplementary material the description of an alternative, functional parcellation derived from the data. As shown by supplementary Figures S7_{a-g}, S10 and S11 and Tables S9, the contrasts across experimental conditions

when using the functional parcellation returned results which were highly comparable with AAL, demonstrating that the significance of our findings did not depend on the chosen parcellation. All the subsequent analyses reported in the manuscript were computed on the AAL parcellation. Moreover, we complemented them by performing the same analyses using the functional parcellation, as reported in the supplementary material (**Figures S15, S16, S17, S18** and **Tables S14, S15**)”.

Additional changes were made in the Methods section, starting at line 1043:

- Is there enough detail provided in the methods for the work to be reproduced? Generally yes. However, some parts of the methods-description remain unclear: (1) The description says that “participants listened to a short (approximately 25 seconds long) musical piece twice” and afterwards describes that the “musical piece consisted of the first four measures of the right-hand part of Johann Sebastian Bach’s Prelude No. 2 in C Minor, BWV 847”. The description is ambiguous: four measures should be much shorter than 25 sec?

The musical piece consisted of the first four bars of the right-hand part of Johann Sebastian Bach’s Prelude No. 2 in C Minor, BWV 847. In this piece, each bar included 16 tones. Thus, the total number of tones was $16 \times 4 = 64$. Each tone lasted approximately 350 ms for a total of 22400 ms. In addition, to provide a sense of musical closure, we included a final tone after the four bars which lasted 1000 ms. Thus, the total was 23400 ms which correspond to 23.4 seconds. This figure provides the piece in musical notation. Inspired by your comment, we have provided a more complete and accurate description of the short musical piece in the methods section (line 721):

“The musical piece consisted of the first four bars of the right-hand part of Johann Sebastian Bach’s Prelude No. 2 in C Minor, BWV 847. In this piece, each bar included 16 tones. Thus, the total number of tones was $16 \times 4 = 64$. Each tone lasted approximately 350 ms for a total of 22400 ms. In addition, to provide a sense of musical closure, we included a final tone after the four bars which lasted 1000 ms. Thus, the total was 23400 ms which correspond to 23.4 seconds. The piece is illustrated in musical notation in **Figure S1**. A MIDI version of the piece was created using Finale (MakeMusic, Boulder, CO)”.

Moreover, we have provided a new supplementary figure (**Figure S1**) which illustrates the short musical piece in musical notation.

- (2) There seems to be a mix-up in Figure S1 containing all the sequences used in the study: (a) part 2 and 3 are switched (part 2 should be on p. 39, part 3 on p. 40); (b) the occasionally appearing 2/4 (time signature in some bars) is also confusing and should be removed.

Thank you very much for noticing this oversight from our side. It is indeed true, (i) two pages of **Figure S1** were switched; we have fixed that. Moreover, we have removed the 2/4 time signature wherever it was present. Please, note that the **Figure S1** of the original manuscript is now coded as **Figure S2**.

Minor comments:

- Title: Consider adding “musical sequences” in the title.

Thank you for this suggestion.

We have indeed slightly modified the title, but we would prefer not to refer to “musical sequences” in the title to maintain brevity.

We have instead made sure to clearly specify in the abstract that our stimuli were “musical sequences”.

- Table 2: there are two occurrences that deviate from the typical “pattern” of results: ACL

for M vs. NT3, 1.45 – 1.55 s, and ACL for M vs. NT4, 1.75 – 1.85 s. They should perhaps be discussed.

Following further analyses, we have significantly revised **Table 2**, and it no longer includes that specific result. Nevertheless, inspired by your comment, we have made sure to carefully discuss every result reported in the manuscript.

- Table 3: Please provide both the time relative to the onset of the first tone of the sequence as well as the time relative to the onset of the deviation. This would make comparing between the deviations (NT1, ..., NT4) much easier.

Following your comment, we have incorporated the deviation onset (NT1, NT2, NT3, NT4) in **Table 3**. As you rightfully pointed out, this enhancement improves the table's readability and helps highlight the significant differences in brain activity following the introduction of deviations.

- Figure 3: Please describe more exactly how the neural activity index was calculated. I am also wondering whether applying a low-pass filter would make the figure more easy to read?

Thank you for highlighting this issue.

The “Neural activity index” is simply the output of the source reconstruction, corrected for the spatially varying bias which leads to an overestimation of the variance in the centre of the brain^{26, 27}. The correction consists of normalising the weights computed by the beamforming algorithm by using a noise covariance matrix between the MEG sensors. This matrix can either be measured or estimated. We used the weights normalization technique described by Luckhoo and colleagues²⁷, which is an effective and standard solution to correct for the bias towards the centre of the brain during source reconstruction.

Please, note that “Neural activity index” is a pure number (i.e. it does not have any unit of measurement).

Following your suggestion, we have specified in the source reconstruction sections that the time series outputted by the source reconstruction procedure were named “Neural activity index”.

Line 986:

“Finally, the weights were applied to each brain source and time-point. To be noted, the covariance matrix C was computed on the continuous signal, which was estimated by concatenating the trials of all experimental conditions. The weights were applied to the brain data associated with each condition and normalised according to Luckhoo and colleagues²⁶⁻²⁹. For counterbalancing the reconstruction bias towards the centre of the head. The weights were applied to the neural activity averaged over trials for the evoked responses and to the neural activity of each independent trial for the induced responses. This procedure returned a time series for each of the 3559 brain sources (and each trial in the case of induced responses), commonly referred to as “neural activity index”^{26, 27”}.

And line 277:

“Neural sources of the MEG channels peak activity

First, we computed source reconstruction by employing a local-spheres forward model and a beamforming approach as inverse solution in an 8-mm grid (corresponding to 3559 brain voxels), using magnetometers only (**Figure 1e**). This procedure returned a time series for each of the reconstructed brain voxel, which is commonly referred to as neural activity index (see Methods for extensive details on this standard procedure”).

- Figure 4c: I was wondering whether some form of significance threshold could be shown (e.g., a grey box from where a certain value of rho would become significant). I also think that now the direction of the cross-correlation is VLPFC → AC (since the peak is before the 0 on the time axis). Having it the other way round is perhaps easier to understand?

Inspired by your comment and by the observations of the other Reviewers, we computed additional analyses to support our claim of hierarchical processing in the brain. In fact, instead of simply computing cross-correlations, we performed Dynamic Causal Modelling

(DCM). DCM, developed by Karl Friston and colleagues within the framework of predictive coding theory³⁰⁻³⁵, infers causal relationships in dynamical systems like brain networks. It employs Bayesian model comparison to evaluate competing models that explain the generation of time series data and their hierarchical relationships³⁰⁻³⁵.

Our DCM analysis confirmed the principles of predictive coding. It revealed feedforward connections from sensory regions (auditory cortices) to higher-order areas (hippocampi, anterior cingulate and medial cingulate gyri), as well as backward connections from higher-order regions to sensory regions. This aligns with the idea that predictions originate in higher-order areas and are transmitted to sensory regions. Additionally, our analysis demonstrated that sensory regions send inputs to higher-order areas after initial stimulus processing.

Importantly, these findings not only aligned with existing predictive coding literature but also significantly expanded on it. In fact, they provided quantitative evidence of predictive coding and fast-scale brain hierarchies during a complex cognitive task, involving episodic memory recognition for musical sequences.

Here, we showed that hippocampi, medial and anterior cingulate gyri operated at the same hierarchical level, receiving feedforward signals from the auditory cortex and simultaneously sending back backward signals. Interestingly, this hierarchical alignment persisted consistently throughout the entire sequence, except for the final tone. Here, the cingulate gyrus, which holds a pivotal role in decision-making and evaluation^{36, 37}, assumed the top position within the hierarchy. For the memorised sequences, the hierarchy consisted of the auditory cortex sending feedforward signals to the hippocampus and anterior cingulate gyrus, and subsequently to the medial cingulate gyrus (and receiving feedback signals in the opposite direction). Similarly, for the varied sequences, the hierarchy showed feedforward connections from the auditory cortex to the hippocampus and medial cingulate and then to the anterior cingulate (and feedback in the opposite direction). This result suggests that when listening to the last tone the brain might prepare the evaluation of the sequence (i.e. categorising it as 'memorised' or 'novel') and for this reason either the anterior or medial cingulate, crucial regions for decision-making and evaluative processes^{36, 37}, may need to occupy the highest position in the hierarchy. In addition, our results present a significant dissociation between anterior and medial cingulate, suggesting a privileged role of the medial part of the cingulate for recognition of memorised sequences and of the anterior part for novelty detection.

Extensive details on the DCM procedure, results and discussions are provided in the revised manuscript.

For your convenience, we report as follows the revised Results section (line 473):

"Dynamic causal modelling

In this study, DCM was used to test our hypothesised model of brain hierarchies during recognition of memorised and varied musical sequences against five competing models, which are described in detail in the Methods section and illustrated in **Figure 1g**. These models comprised the six AAL ROIs described in the previous section and depicted in **Figure S8**. Our hypothesised model consisted of feedforward connections from LHG, RHG to LHP, RHP, ACC, MC and feedback connections from LHP, RHP, ACC, MC to LHG, RHG (**Figure 6a**, red box).

Since DCM for MEEG is usually computed on relatively short time-windows (e.g. 350 – 400 ms) and for one experimental perturbation (i.e. sound) at a time^{30, 31}, we computed a series of DCM analyses independently for each tone of the sequences, in a 350 ms time-window from each tone onset. Specifically, we computed an independent DCM for tones two, three, four and five for condition M. In relation to the N conditions, we were interested only in the tone that introduced the variation in the sequences. Thus, we computed four DCMs, organised as follows: tone two for NT1; tone three for NT2; tone four for NT3; tone five for NT4.

To compare alternative models, we estimated the free energy F associated with the computation of each model (model evidence). To study the consistency of F across the population, we used a random-effects Bayesian model selection (RFX BMS) procedure, which provides the posterior probability and protected exceedance probability of each model, indicating which of the alternative models has the

strongest model evidence within the whole population. In addition, we computed the Bayesian omnibus risk (BOR), which represents the probability that the protected exceedance probability associated with each model comparison is attributable to chance within the sample of participants (**Figure 6b** and **6c**)³⁰.

This procedure was observed independently for the eight DCM analyses described above (four for M and four for Ns).

Results indicated that our hypothesised model was the one with the highest evidence for all the DCM analyses of tones two, three and four (**Figure 6**). This result showed a high probability of non-being driven by random chance, as demonstrated by the BOR parameter: $BOR_{M-tone2} = 2.91e-07$; $BOR_{M-tone3} = 4.33e-04$; $BOR_{M-tone4} = .008$; $BOR_{NT1-tone2} = 4.49e-12$; $BOR_{NT2-tone3} = .010$; $BOR_{NT3-tone4} = .003$.

Regarding tone five, the model with the highest evidence for M showed a hierarchical architecture which consisted of the auditory cortex leading to the hippocampus and anterior cingulate gyrus and subsequently to the medial cingulate gyrus ($BOR_{M-tone5} = .006$). For NT4, the model with the highest evidence presented a hierarchy which started in the auditory cortex and led to the hippocampus and medial cingulate gyrus and then to the anterior cingulate gyrus ($BOR_{NT4-tone5} = .036$)”.

Discussion (line 599):

“Our results fit the well-known PCT¹²⁻¹⁴, which states that the brain is constantly updating internal models to predict external information and stimuli. Recently, this theory has been proposed to explain the brain mechanisms underlying complex cognitive processes¹⁵, finding a remarkable example in the neuroscience of music^{13, 14}. However, while PCT holds promise for understanding complex cognitive processes, the quantitative evidence in its favour has been limited. Thus, our study filled this gap by providing quantitative evidence of the brain functional hierarchies during long-term recognition of previously memorised and novel sequences. The results showed feedforward connections from the auditory cortices, representing sensory regions^{38, 39}, to the hippocampus, anterior cingulate gyrus, and medial cingulate gyrus, brain areas which were previously associated with memory and predictive processes^{40, 41}, and evaluation³⁶. Simultaneously, we observed backward connections operating in the opposite direction. This hierarchical alignment persisted consistently throughout the entire sequence, except for the final tone. Here, the cingulate gyrus, which holds a pivotal role in decision-making and evaluation^{36, 37}, assumed the top position within the hierarchy. For the memorised sequences, the hierarchy consisted of the auditory cortex sending feedforward signals to the hippocampus and anterior cingulate gyrus, and subsequently to the medial cingulate gyrus (and receiving feedback signals in the opposite direction). Similarly, for the varied sequences, the hierarchy showed feedforward connections from the auditory cortex to the hippocampus and medial cingulate and then to the anterior cingulate (and feedback in the opposite direction). This result may suggest that when listening to the last tone the brain is preparing the evaluation of the sequence (i.e. categorising it as ‘memorised’ or ‘novel’) and for this reason either the anterior or medial cingulate, relevant regions for decision-making and evaluative processes^{36, 37}, may need to occupy the highest position in the hierarchy.

In addition, our results present a significant dissociation between anterior and medial cingulate, suggesting a privileged role of the medial part of the cingulate for recognition of memorised sequences and of the anterior part for novelty detection.

These findings align with previous research that has demonstrated the flow of information from sensory cortices to the medial temporal lobe, associative areas, and prefrontal regions of the brain^{42, 43}. Moreover, our results are coherent with the several studies which used PCT to investigate perceptual⁴⁴ and automatic memory mechanisms in the brain⁴⁵, as reviewed by Spratling¹⁵. Indeed, they described how information processing, orientation tuning, contour integration, binocular rivalry in the visual system relied on the hierarchical transition of the psychophysical and neurophysiological signals from the retina to the lateral geniculate nucleus and V1⁴⁶⁻⁴⁹. In the context of automatic sensory memory, PCT has also lent support to the interpretation of the MMN as an error signal which aids the brain in adjusting its internal predictive model, returning quantitative evidence of information flow from the primary auditory cortex to the superior temporal gyrus and inferior frontal gyrus⁴⁵. Our findings expand on these studies by providing quantitative evidence of the hierarchical organisation of the brain during a complex cognitive task, involving episodic memory recognition for musical sequences. In addition, while the hierarchical architecture involved in recognising both memorised and varied sequences remained largely unchanged, the temporal dynamics, strength, and polarity of the brain signal sharply differed”.

Additional changes have been made to the Methods section (line 1098).

Moreover, a graphical depiction of the DCM outcome is provided in **Figure 6**.

Finally, we are glad to report that, in relation to our new extensive DCM analyses, we have invited Ass. Professor Martin Dietz to join the paper as an author. In his career, he has primarily been working with DCM applied to EEG, MEG and fMRI data and has a wide expertise in analysing this data and interpreting the outcomes of DCM.

• I. 335: Define which frequency range was regarded as gamma. I am also wondering whether putting lines separating the different frequency bands in Figure 5 would help or rather make the figure more difficult to read.

This is a very good point. We have now better specified that we referred to gamma band for the frequencies higher than 30 Hz⁵⁰. Moreover, we have provided additional time-frequency analyses and plots. The new plots show the main effect of conditions M and NT1 as well as their thresholded contrast, which highlights only the significant differences (**Figure 7**).

Furthermore, additional supplementary figures show the results for the other contrasts (i.e. M vs NT2, M vs NT3, M vs NT4, **Figure S12**) as well as the results for M, NT1 and their contrast for eight fronto-temporal and occipital MEG channels (**Figure S13**).

Altogether, the results aligned to each other and highlight two key findings:

1. A strongly increased theta-alpha-beta power for N conditions vs M (especially in the frequencies 4-30 Hz) after the end of the musical sequences.
2. An overall increase of gamma power (higher than 30 Hz) for M vs N conditions, evidenced by a few small yet significant clusters.

Accordingly, in the revised discussion, we have enhanced our focus on the increased theta-alpha-beta power for N conditions vs M (especially in the frequencies 4-30 Hz) after the end of the musical sequences.

Results (line 533):

“Time-frequency analysis for induced responses

We computed time-frequency analysis using complex Morlet wavelet transform (from 1 to 60 Hz with 1-Hz intervals). This analysis was conducted for induced responses. First, we estimated independent time-frequency decompositions for each trial and each voxel of the six ROIs described above (LHG, RHG, LHP, RHP, ACC, MC) plus left and right occipital superior lobe. Then, the computed power spectra were averaged over voxels and over trials within each ROI. Finally, in line with the previous analyses, we calculated four contrasts (M versus NT1, M versus NT2, M versus NT3, and M versus NT4). Specifically, we computed a t-test for each frequency and time-point and corrected for multiple comparisons using 2D cluster-based MCS (MCS, $\alpha = .05$, MCS p-value = .001). As shown in **Figure 7**, results were similar across ROIs and displayed a generalised stronger power for N versus M between 2 and 20 Hz (corresponding to, approximately, theta, alpha, and beta bands), in the time window 1.0 – 3.0 seconds ($p < .001$). In addition, a few significant clusters showed a moderate yet significant increase of power for M versus N ($p < .001$) for frequencies in the gamma band (i.e., higher than 30 Hz)⁵⁰. Detailed statistical results about this procedure are extensively reported in **Table S11** and depicted in **Figure S12**.

Finally, to strengthen the reliability of our results in MEG source space, we computed the same analysis for eight fronto-temporal and occipital MEG channels. **Figure S13** and **Table S12** shows that the outcome of this procedure aligned with the results in MEG source space”.

Discussion (line 682):

“Finally, the induced-response analysis showed that, after the end of the sequence, alpha and beta bands were stronger for the varied compared to the memorised sequences. Considering the modulation of alpha and beta activity in encoding and working memory tasks described in the scientific literature⁵¹, this finding may suggest that post-stimulus increase of alpha-beta power might be relevant to process novel information. In addition, this analysis revealed a weak yet significant increase in gamma power for the memorised compared to the varied sequences. This is coherent with previous studies which reported increased gamma power during recognition of target stimuli^{52, 53} and, more generally, a modulation of the brain oscillations associated with memory load and complex cognitive functions⁵⁴. Although this study provided initial insights on the time-frequency modulation occurring in the brain during long-term recognition of auditory sequences, further research employing MEG and additional tools such as stereo-electroencephalography (SEEG) is needed to expand our experimental design and

conduct cross-frequency coupling analysis, testing whether gamma-theta coupling is connected to long-term recognition of auditory sequences”.

Additional changes were made in the Methods section (line 1232).

We did not add the lines separating the frequency bands because, as you pointed out, it would make the figures slightly more difficult to read.

• I. 357: the results ... expand (no s)

Thank you, we have corrected it.

• I. 363: These peaks – It is unclear whether both the negative and the positive component are meant or the positive component in the different regions.

You are right, we meant to refer to the positive component in the auditory cortex which followed the well-known N100. However, we have now removed that particular sentence from the revised Discussion. Nevertheless, we have made sure to clearly describe our results throughout the revised Discussion.

• I. 529: Their educational background was homogeneous. Be more explicit: What exactly was their education level?

The total number of participants was 83. Seventy-three had either a university degree (bachelor's or master's degree, $n = 54$) or were university students ($n = 19$). The remaining 10 were divided as follows: five had a professional degree obtained after finishing high-school, while the remaining five had only the high-school diploma.

We have provided this information in the revised manuscript (line 705):

“The participant sample consisted of 83 volunteers [33 males and 50 females (sex, biological attribute)] aged 19 to 63 years old (mean age: 28.76 ± 8.06 years). The sample was recruited in Denmark and participants came from Western countries. All participants were healthy and reported normal hearing. Their educational background was overall homogeneous. Specifically, 73 participants had either a university degree (bachelor's or master's degree, $n = 54$) or were university students ($n = 19$). The remaining 10 were divided as follows: five had a professional degree obtained after finishing high-school, while the remaining five had only the high-school diploma”.

• I. 607: Justify a bit why there was a downsampling by factor 4. Why were 1000 Hz recorded in the first place? And why wasn't that sampling rate kept?

The data was recorded at 1000 Hz because this is the default recording setting for the Elekta Neuromag TRIUX MEG scanner with 306 channels (Elekta Neuromag, Helsinki, Finland).

We downsampled the data by factor 4 for two reasons:

1. MEG is not particularly sensitive to high-gamma frequency (e.g. 100 Hz)^{55,56}, while it provides much better outcomes for lower frequency bands (e.g. 0.1 – 60 Hz, as done in the current study)^{55, 56}. Moreover, our research focused on the event-related broadband brain data during memory recognition, which are often expressed in slower bands^{16,57}. In particular, we wished to observe the broadband brain mechanisms underlying recognition of memorised and novel musical sequences and how these mechanisms were hierarchically organised. For these reasons, we did not need a sampling rate which was more fine-grained than 250 Hz.
2. Downsampling the data after MEG recording (e.g. to 250 Hz) is a rather common procedure^{55, 56}. It significantly accelerates computations by reducing data volume without substantial data loss, particularly for investigations like ours.

We have provided additional information in the Methods section of the revised manuscript (line 802):

“We opted to downsample the data by a factor of four because MEG is not particularly sensitive to high-gamma frequency (e.g. 100 Hz) ^{55, 56}, while it provides much better outcomes for lower frequency bands (e.g. 0.1 – 60 Hz, as done in the current study) ^{55, 56}. Moreover, our research focused on the event-related broadband brain data during memory recognition, which are often expressed in slower bands ^{16, 57}. For these reasons, as commonly done in the literature ^{55, 56}, we downsampled the data, a procedure which significantly accelerated the computations by reducing data volume without substantial data loss ^{55, 56}”.

- I. 621: Describe whether the selection of components for rejection was manual or automatized. If the former, describe the criteria for rejection, if the latter describe which package was used with which parameters.

This is a very good point, and we agree that we did not provide sufficient details in the original version of the manuscript. As follows we provide a detailed description of the standard procedure that we followed.

Independent component analyses (ICA) were used (OSL implementation) to discard the interference of eyeblinks and heart-beat artefacts from the brain data. First, we decomposed the original signal into independent components. Second, we correlated all the components with the activity recorded by the EOG and ECG channels. This procedure identified the ICA component X that strongly correlated with either the EOG or the ECG (strongly correlated means that the correlation coefficient for component X was at least three times higher than the correlation coefficients for the remaining components). To validate the accuracy of this procedure, we visually inspected the highly correlated components. We assessed whether their topographic distribution across MEG channels matched the typical distribution associated with eyeblink and heartbeat activity. When both the correlations and visual inspections converged, indicating that the ICA component strongly reflected eyeblink or heartbeat activity, those components were excluded. Third, the signal was rebuilt using the remaining components.

This information has been reported in the revised manuscript (line 818):

“Independent component analyses (ICA) were used (OSL implementation) to discard the interference of eyeblinks and heart-beat artefacts from the brain data ⁵⁸. First, we decomposed the original signal into independent components. Second, we correlated all the components with the activity recorded by the EOG and ECG channels. This procedure identified the ICA component X that strongly correlated with either the EOG or the ECG (strongly correlated means that the correlation coefficient for component X was at least three times higher than the correlation coefficients for the remaining components). To further validate the accuracy of this procedure, we also visually inspected the highly correlated components. We assessed whether their topographic distribution across MEG channels matched the typical distribution associated with eyeblink and heartbeat activity. When both the correlations and visual inspections converged, indicating that the ICA component strongly reflected eyeblink or heartbeat activity, those components were discarded. Third, the signal was rebuilt using the remaining components”.

- I. 724: was associated (not association)

Thanks, we fixed it.

- I. 728 - 731: The description is quite unclear: I guess the first part aims to describe a difference in neural activity between the conditions being either positive or negative? But isn't that passage about defining time windows: i.e., it should be mentioned that while the description uses x vs. $x + 1$, it is actually about how long a difference with a particular sign extends that is most relevant here? Also: Try to describe more clearly how looking at the results from decoding and looking at the corresponding brain activity was implemented.

We have significantly improved this segment of our analyses and modified its description accordingly. E.g. Results, from line 251, and Methods, from line 878.

Reviewer #2 (Remarks to the Author):

The authors used MEG to measure oscillatory activity associated with the explicit recognition of changes in a memorised 5-tone sequence. Multivariate pattern analysis is used on MEG data to define region in which the time series differ between the memorised stimulus and altered sequence at positions beyond the first. Analyses within regions were based on evoked response and time-frequency analysis. A hierarchy of areas is suggested based on higher order areas that send predictions to auditory cortex.

We would like to express our gratitude for your thoughtful and constructive review of our manuscript. Your insightful comments have significantly contributed to refining the clarity and scientific rigour of our work. In response to your detailed feedback, we have implemented the following key revisions:

1. *Clarification of the hypotheses, which are presented as follows:*
 - a. The conscious recognition of previously memorised auditory sequences relied on the recruitment of a widespread brain network responding to each sound of the sequence.
 - b. The detection of the varied auditory sequences elicited a conscious prediction error in the brain, originating in the auditory cortex and propagating to hippocampus, anterior cingulate and ventromedial prefrontal cortex.
 - c. In relation to the recognition of both previously memorised and novel sequences, we expected to observe a brain hierarchy characterised by feedforward connections from auditory cortices to hippocampus and cingulate gyrus, along with feedback connections in the opposite direction.
2. *Gamma activity and Predictive Coding:*
 - a. We have toned down our emphasis on gamma activity in the discussion, placing a stronger focus on the evidence of brain hierarchies and predictive coding supported by the Dynamic Causal Modelling (DCM) analysis.
3. *Alignment with Predictive Coding Theory (PCT):*
 - a. We have expanded on how our findings align with predictive coding theory in the Discussion section, emphasising the relevance of predictive coding in understanding complex cognitive tasks which involves episodic memory such as our task.
4. *Definition of brain regions of interest (ROIs):*
 - a. We have conducted additional analyses using the automated anatomical labelling (AAL) parcellation, aligning with Reviewer's suggestion.
 - b. We have presented the results based on AAL parcellation in the revised manuscript, reinforcing the foundation of our claims. Moreover, we have reported in supplementary materials the results related to the functional parcellation presented in the original manuscript. This allows full transparency and provides the readers with additional, relevant information.
5. *Effective connectivity analysis:*
 - a. In response to your suggestion, we applied DCM for effective connectivity analysis, offering insights into the hierarchical relationships and predictive coding principles between key brain areas.
6. *Role of Inferior Frontal Gyrus (IFG):*
 - a. Your comment regarding the limited role of IFG in our findings prompted further investigation. The revised manuscript now includes a dedicated paragraph discussing the relevance of IFG in the context of long-term recognition.

Our extensive additional clarifications and analyses were complemented by 50 novel panels distributed across 18 supplementary figures and by 15 supplementary tables, providing a better understanding of every detail of our procedures.

Please find detailed responses to each of your comments below in this letter. To facilitate your review, we have reported in this letter the revised sections/paragraphs of the manuscript in relation to your main comments. For minor comments, we have indicated only the line of the revised manuscript where changes were made to avoid unnecessary length in this letter.

The fully revised figures are provided in the revised manuscript and in the supplementary materials. Moreover, they are also attached to the submission in high definition.

We trust that these enhancements strengthen the scientific merit of our work, aligning it more closely with your expectations. We sincerely appreciate the time and expertise you dedicated to reviewing our manuscript.

Detailed answers

1. The study is a very carefully conducted description of the brain activity corresponding to the encoding of melodies into episodic memory (although the authors' term acme might be a little hyperbolic).

Thank you for your kind words. We agree that 'acme' is an exaggeration, and we have now removed it.

2. I would like to have seen a more explicit description of a hypothesis. The study is couched in terms of hierarchical organisation of sequence processing but the predictive coding model is not explicitly tested, and if you do generate predictions I am not sure that they fit. For example, a fundamental aspect of predictive coding is the generation of predictions in higher areas and prediction error in sensory cortex when these are compared with incoming signal (note the prediction and prediction error arrows in figure 4 are the wrong way around). Classical predictive coding is based on a canonical coding scheme in which prediction errors are encoded as a gamma signal but the data here show increased gamma in response to the memorised melody. Does that fit with a predictive coding model?

You are right, the description of our hypotheses can be improved.

The main hypotheses of the study were:

1. The conscious recognition of previously memorised auditory sequences relied on the recruitment of a widespread brain network including auditory cortex, hippocampus, cingulate gyrus. We expected to observe that these regions responded to each tone forming the sequence, with the auditory cortex's response preceding that of the hippocampus and cingulate gyrus.
2. The detection of the varied auditory sequences elicited a conscious prediction error in the brain, originating in the auditory cortex and propagating to hippocampus, anterior cingulate and ventromedial prefrontal cortex. While we anticipated a consistent response in the auditory cortex to the introduction of varied sounds at any point in the sequence, our expectation was that the anterior cingulate gyrus and ventromedial prefrontal cortex would exhibit their strongest response exclusively when the variation in the sequence was introduced.
3. We hypothesised to observe a brain hierarchy characterised by feedforward connections from auditory cortices to hippocampus and cingulate gyrus, along with feedback connections in the opposite direction. Moreover, while we anticipated differences in the temporal dynamics, strength and polarity of brain responses between recognising previously memorised and varied auditory sequences, we expected consistent brain hierarchies in both processes.

These three hypotheses are firmly grounded in previous literature, particularly within the predictive coding framework¹²⁻¹⁴, and they expand on it by investigating predictive coding using fast-scale neurophysiology during a complex episodic memory task centred on auditory (musical) information. We have clearly stated these hypotheses at the end of the Introduction. Moreover, since we have been asked to carefully follow the editorial guidelines of *Nature Communications* when performing the current revision, after stating the hypotheses, we had to also provide a succinct summary of the main findings at the end of Introduction.

For your convenience, we report the fully revised Introduction as follows.

"Introduction

The spatiotemporal features of hierarchical processing in the brain are essential to fully grasp the neural substrates of human perception and cognition, as suggested by the predictive coding theory (PCT)^{59-63, 12, 13}. To elucidate such brain mechanisms, much research has focused on the visual system^{64, 65}, which primarily relies on the recognition of patterns arranged in space. Conversely, the auditory system

extracts information from patterns and sequences that develop over time ⁶⁶, providing unique opportunities to understand the temporal hierarchies of the brain.

Extensive research spanning decades has established the hierarchical organisation of auditory perception, with a particular emphasis on the processing of elementary auditory stimuli. This hierarchical progression starts at the peripheral level within the cochlea and moves forward to the auditory pathway, which encompasses the brainstem, pons, trapezoid body, superior olivary complex, lateral lemniscus, inferior and medial geniculate nucleus of the thalamus, and ends in the primary auditory cortex ^{67, 68}. However, how auditory information is integrated from the auditory cortex to the whole brain has not been fully established yet.

Aiming to bridge this gap, prior investigations focused on automatic predictive processes within the framework of PCT. These studies assessed auditory automatic prediction error, often relying on well-established event-related potential/field (ERP/F) components such as N100, mismatch negativity (MMN), and error-related negativity (ERAN) ¹⁻⁹. They demonstrated that such components were automatically evoked in response to auditory stimuli, deviations from expected sound features, likelihood of occurrence of musical tones (N100, MMN), and varied harmonic properties (ERAN). Additional studies employed dynamic causal modelling (DCM) to investigate the brain hierarchical architecture during automatic predictive processes, as indexed by MMN. This research has yielded quantitative evidence demonstrating the flow of information from the primary auditory cortex to the superior temporal gyrus and the inferior frontal gyrus ⁴⁵.

Expanding upon these investigations, Rocchi and colleagues provided evidence of hierarchical organisation in the auditory system using direct electrical brain stimulation. They showed effective connectivity from the auditory cortex to regions in the medial temporal lobe and prefrontal cortex, including the hippocampus and ventro-lateral prefrontal cortex ⁴³. Yet, it remains uncertain how the integration of auditory information across the whole brain relates to complex cognitive functions, such as the conscious recognition and prediction of auditory sequences evolving over time.

To address this question, a unique perspective has arisen through the integration of musical memory paradigms with magnetoencephalography (MEG). Music is a complex artform acquiring meaning through the combination of its constituent elements extended over time ⁶⁹, while MEG, particularly when used in combination with MRI, allows to track the brain activity with excellent temporal resolution in the order of milliseconds and acceptable spatial accuracy ⁵⁶. It is precisely due to these attributes that this combination offers a unique framework for investigating the rapid memory and predictive processes of temporal information in the human brain ^{13, 14}.

Indeed, recent studies employing MEG and musical paradigms investigated the brain mechanisms associated with perception, manipulation and recognition of sound sequences. As an example, Albouy and colleagues ⁵⁷ explored the brain activity underlying memory retention, showing that theta oscillations in the dorsal stream of the participants' brain predicted their auditory WM performance. Along this line, we recently uncovered a vast network of brain areas responsible for encoding and recognising previously memorised and novel musical sounds. This network extended from the auditory cortex to the medial cingulate, inferior temporal cortex, insula, frontal operculum, and hippocampus ^{16, 22, 70}. We also observed that the complexity of music ¹⁷ and individual cognitive differences ¹⁸ influenced the activity recorded in this network.

In our previous studies, we utilised relatively rapid musical tones with a duration of 250 ms. However, this approach prevented us from closely tracking the neural dynamics associated with each sound within the sequences, leading to frequent overlap with the brain responses to subsequent tones. For this reason, we were also unable to study the rapid hierarchies between the brain regions that we revealed, which is necessary to understand the neurophysiology of conscious recognition of auditory sequences.

Moreover, while our previous works showed differences in brain activity when comparing previously memorised versus completely novel auditory sequences, we did not investigate the scenario in which an original sequence was systematically varied. This prevented us from exploring how the brain generates prediction errors within the framework of a conscious long-term memory recognition task, as opposed to the several instances of automatic prediction errors described in the literature ¹⁻⁹.

To address these questions, in the present study we made two essential modifications to our previously employed experimental paradigm: (i) we compared the original five-tone melodies with systematic variations occurring at the 2nd, 3rd, 4th, or 5th tone; (ii) we increased the duration of the tones from 250 ms to 350 ms. These seemingly minor modifications were of great importance, as they enabled us to directly address our three main hypotheses, stated as follows. First, the conscious recognition of previously memorised auditory sequences relied on the hierarchical recruitment of a widespread brain network including auditory cortex, hippocampus and cingulate gyrus. We expected to observe that these regions responded to each tone forming the sequence, with the auditory cortex preceding the hippocampus and cingulate gyrus. Second, the detection of the varied auditory sequences elicited a

prediction error, originating in the auditory cortex and then propagating to hippocampus, anterior cingulate and ventromedial prefrontal cortex. While we expected a consistent response in the auditory cortex to the introduction of varied sounds at any point in the sequence, we hypothesised that the anterior cingulate gyrus and ventromedial prefrontal cortex exhibited their strongest response exclusively when the variation in the sequence was introduced. Third, sequence recognition relied on a brain hierarchy characterised by feedforward connections from auditory cortices to hippocampus and cingulate gyrus, simultaneous with feedback connections in the opposite direction. Moreover, recognition of both previously memorised and varied sequences relied on the same hierarchy, while the temporal dynamics, strength and polarity of brain responses differed.

These hypotheses were confirmed by the results of the experiment, which showed that the brain distinctively responds to each tone forming the previously memorised sequence, with the auditory cortex preceding hippocampus and anterior and medial cingulate gyrus. In addition, the detection of the sequence variations elicits a conscious prediction error in the brain, originating in the auditory cortex and propagating to hippocampus, anterior and medial cingulate. While the auditory cortex has a constant response to the varied sounds introduced at any point in the sequence, the hippocampus and anterior and medial cingulate gyrus exhibit their strongest response exclusively to the sound which introduced the variation in the sequence. Finally, we provide quantitative evidence of the brain functional hierarchies during long-term auditory recognition. Here, as hypothesised, feedforward connections originate from the auditory cortices and extend to the hippocampus, anterior cingulate gyrus, and medial cingulate gyrus, along with feedback connections in the opposite direction. Notably, throughout the sequence, the hippocampus and cingulate gyrus maintain the same hierarchical level, except for the final tone, where the cingulate gyrus assumes the top position within the hierarchy”.

Moreover, as you rightfully pointed out, in the original version of the manuscript, we did not quantitatively test the hierarchical organisation and predictive coding of the brain during recognition of musical sequences. Following your comment, we have computed a novel extensive analysis using the Dynamic Causal Modelling (DCM).

As you will know, DCM was developed by Karl Friston and colleagues within the framework of predictive coding theory³⁰⁻³⁵ to infer causal relationships in dynamical systems like brain networks. It employs Bayesian model comparison to evaluate competing models that explain the generation of time series data and their hierarchical relationships³⁰⁻³⁵.

Our DCM analysis confirmed the principles of predictive coding. It revealed feedforward connections from sensory regions (auditory cortices) to higher-order areas (hippocampi, anterior cingulate and medial cingulate gyri), as well as backward connections from higher-order regions to sensory regions. This aligns with the idea that predictions originate in higher-order areas and are transmitted to sensory regions. Additionally, our analysis demonstrated that sensory regions send inputs to higher-order areas after initial stimulus processing.

Importantly, these findings not only aligned with existing predictive coding literature but also significantly expanded on it. In fact, they provided quantitative evidence of predictive coding and fast-scale brain hierarchies during a complex cognitive task, involving episodic memory recognition for musical sequences.

Here, we showed that hippocampi, medial and anterior cingulate gyri operated at the same hierarchical level, receiving feedforward signals from the auditory cortex and simultaneously sending back backward signals. Interestingly, this hierarchical alignment persisted consistently throughout the entire sequence, except for the final tone. Here, the cingulate gyrus, which holds a pivotal role in decision-making and evaluation^{36, 37}, assumed the top position within the hierarchy. For the memorised sequences, the hierarchy consisted of the auditory cortex sending feedforward signals to the hippocampus and anterior cingulate gyrus, and subsequently to the medial cingulate gyrus (and receiving feedback signals in the opposite direction). Similarly, for the varied sequences, the hierarchy showed feedforward connections from the auditory cortex to the hippocampus and medial cingulate and then to the anterior cingulate (and feedback in the opposite direction). This result suggests that when listening to the last tone the brain might

prepare the evaluation of the sequence (i.e. categorising it as ‘memorised’ or ‘novel’) and for this reason either the anterior or medial cingulate, crucial regions for decision-making and evaluative processes^{36, 37}, may need to occupy the highest position in the hierarchy. In addition, our results present a significant dissociation between anterior and medial cingulate, suggesting a prominent role of the medial part of the cingulate for recognition of memorised sequences and of the anterior part for novelty detection.

Extensive details on the DCM procedure, results and discussions are provided in the revised manuscript.

For your convenience, we report as follows the revised Results section (line 473):

“Dynamic causal modelling

In this study, DCM was used to test our hypothesised model of brain hierarchies during recognition of memorised and varied musical sequences against five competing models, which are described in detail in the Methods section and illustrated in **Figure 1g**. These models comprised the six AAL ROIs described in the previous section and depicted in **Figure S8**. Our hypothesised model consisted of feedforward connections from LHG, RHG to LHP, RHP, ACC, MC and feedback connections from LHP, RHP, ACC, MC to LHG, RHG (**Figure 6a**, red box).

Since DCM for MEEG is usually computed on relatively short time-windows (e.g. 350 – 400 ms) and for one experimental perturbation (i.e. sound) at a time^{30, 31}, we computed a series of DCM analyses independently for each tone of the sequences, in a 350 ms time-window from each tone onset. Specifically, we computed an independent DCM for tones two, three, four and five for condition M. In relation to the N conditions, we were interested only in the tone that introduced the variation in the sequences. Thus, we computed four DCMs, organised as follows: tone two for NT1; tone three for NT2; tone four for NT3; tone five for NT4.

To compare alternative models, we estimated the free energy F associated with the computation of each model (model evidence). To study the consistency of F across the population, we used a random-effects Bayesian model selection (RFX BMS) procedure, which provides the posterior probability and protected exceedance probability of each model, indicating which of the alternative models has the strongest model evidence within the whole population. In addition, we computed the Bayesian omnibus risk (BOR), which represents the probability that the protected exceedance probability associated with each model comparison is attributable to chance within the sample of participants (**Figure 6b** and **6c**)³⁰.

This procedure was observed independently for the eight DCM analyses described above (four for M and four for Ns).

Results indicated that our hypothesised model was the one with the highest evidence for all the DCM analyses of tones two, three and four (**Figure 6**). This result showed a high probability of non-being driven by random chance, as demonstrated by the BOR parameter: $BOR_{M-tone2} = 2.91e-07$; $BOR_{M-tone3} = 4.33e-04$; $BOR_{M-tone4} = .008$; $BOR_{NT1-tone2} = 4.49e-12$; $BOR_{NT2-tone3} = .010$; $BOR_{NT3-tone4} = .003$.

Regarding tone five, the model with the highest evidence for M showed a hierarchical architecture which consisted of the auditory cortex leading to the hippocampus and anterior cingulate gyrus and subsequently to the medial cingulate gyrus ($BOR_{M-tone5} = .006$). For NT4, the model with the highest evidence presented a hierarchy which started in the auditory cortex and led to the hippocampus and medial cingulate gyrus and then to the anterior cingulate gyrus ($BOR_{NT4-tone5} = .036$)”.

And the revised Discussion (line 599):

“Our results fit the well-known PCT¹²⁻¹⁴, which states that the brain is constantly updating internal models to predict external information and stimuli. Recently, this theory has been proposed to explain the brain mechanisms underlying complex cognitive processes¹⁵, finding a remarkable example in the neuroscience of music^{13, 14}. However, while PCT holds promise for understanding complex cognitive processes, the quantitative evidence in its favour has been limited. Thus, our study filled this gap by providing quantitative evidence of the brain functional hierarchies during long-term recognition of previously memorised and novel sequences. The results showed feedforward connections from the auditory cortices, representing sensory regions^{38, 39}, to the hippocampus, anterior cingulate gyrus, and medial cingulate gyrus, brain areas which were previously associated with memory and predictive processes^{40, 41}, and evaluation³⁶. Simultaneously, we observed backward connections operating in the opposite direction. This hierarchical alignment persisted consistently throughout the entire sequence, except for the final tone. Here, the cingulate gyrus, which holds a pivotal role in decision-making and evaluation^{36, 37}, assumed the top position within the hierarchy. For the memorised sequences, the hierarchy consisted of the auditory cortex sending feedforward signals to the hippocampus and anterior cingulate gyrus, and subsequently to the medial cingulate gyrus (and receiving feedback signals in the

opposite direction). Similarly, for the varied sequences, the hierarchy showed feedforward connections from the auditory cortex to the hippocampus and medial cingulate and then to the anterior cingulate (and feedback in the opposite direction). This result may suggest that when listening to the last tone the brain is preparing the evaluation of the sequence (i.e. categorising it as 'memorised' or 'novel') and for this reason either the anterior or medial cingulate, relevant regions for decision-making and evaluative processes ^{36,37}, may need to occupy the highest position in the hierarchy.

In addition, our results present a significant dissociation between anterior and medial cingulate, suggesting a privileged role of the medial part of the cingulate for recognition of memorised sequences and of the anterior part for novelty detection.

These findings align with previous research that has demonstrated the flow of information from sensory cortices to the medial temporal lobe, associative areas, and prefrontal regions of the brain ^{42,43}. Moreover, our results are coherent with the several studies which used PCT to investigate perceptual ⁴⁴ and automatic memory mechanisms in the brain ⁴⁵, as reviewed by Spratling ¹⁵. Indeed, they described how information processing, orientation tuning, contour integration, binocular rivalry in the visual system relied on the hierarchical transition of the psychophysical and neurophysiological signals from the retina to the lateral geniculate nucleus and V1 ⁴⁶⁻⁴⁹. In the context of automatic sensory memory, PCT has also lent support to the interpretation of the MMN as an error signal which aids the brain in adjusting its internal predictive model, returning quantitative evidence of information flow from the primary auditory cortex to the superior temporal gyrus and inferior frontal gyrus ⁴⁵. Our findings expand on these studies by providing quantitative evidence of the hierarchical organisation of the brain during a complex cognitive task, involving episodic memory recognition for musical sequences. In addition, while the hierarchical architecture involved in recognising both memorised and varied sequences remained largely unchanged, the temporal dynamics, strength, and polarity of the brain signal sharply differed”.

Additional changes were made to the Methods section (line 1098).

Moreover, a graphical depiction of the DCM outcome is provided in **Figure 6**.

Finally, we are glad to report that, in relation to our new extensive DCM analyses, we have invited Ass. Professor Martin Dietz to join the paper as an author. In his career, he has primarily been working with DCM applied to EEG, MEG and fMRI data and has a wide expertise in analysing this data and interpreting the outcomes of DCM.

Lastly, regarding gamma activity, we clarified in the revised manuscript that its effect, even if significant, was not particularly strong. As a result, we have toned down our emphasis on its interpretation in the discussion section, which now focuses more on the evidence of brain hierarchies and predictive coding, supported by the DCM analysis. Nevertheless, as follows, we report the discussion on the induced responses provided in the revised manuscript (line 682):

“Finally, the induced-response analysis showed that, after the end of the sequence, alpha and beta bands were stronger for the varied compared to the memorised sequences. Considering the modulation of alpha and beta activity in encoding and working memory tasks described in the scientific literature ⁵¹, this finding may suggest that post-stimulus increase of alpha-beta power might be relevant to process novel information. In addition, this analysis revealed a weak yet significant increase in gamma power for the memorised compared to the varied sequences. This is coherent with previous studies which reported increased gamma power during recognition of target stimuli ^{52, 53} and, more generally, a modulation of the brain oscillations associated with memory load and complex cognitive functions ⁵⁴. Although this study provided initial insights on the time-frequency modulation occurring in the brain during long-term recognition of auditory sequences, further research employing MEG and additional tools such as stereo-electroencephalography (SEEG) is needed to expand our experimental design and conduct cross-frequency coupling analysis, testing whether gamma-theta coupling is connected to long-term recognition of auditory sequences”.

3. I accept classical predictive coding has been applied to perception (and sometimes working memory) as opposed to episodic memory as here, but the authors might elaborate

on how well their finding fits with predictive coding which readers will expect when they read the title.

Thank you for this insightful comment. You raise a valid concern about how well our findings align with predictive coding theory (PCT), especially given the title of our study. While PCT has traditionally been applied to perceptual and automatic memory processes, we acknowledge that its direct application to episodic memory is less common. In the broader field of cognitive neuroscience, PCT has been suggested to have relevance in understanding complex cognitive tasks, including music cognition¹³⁻¹⁵. However, precise quantitative evidence of PCT in the context of complex cognitive processes has been limited.

Thus, our study filled this gap by providing quantitative evidence of the brain functional hierarchies during long-term recognition of previously memorised and novel sequences. As described in more details above in response to one of your previous comments, the results showed feedforward connections from the auditory cortices, representing sensory regions^{38, 39}, to the hippocampus, anterior cingulate gyrus, and medial cingulate gyrus, brain areas which were previously associated with memory and predictive processes^{40, 41}, and evaluation³⁶. Simultaneously, we observed backward connections operating in the opposite direction. This hierarchical alignment persisted consistently throughout the entire sequence, except for the final tone. Here, the cingulate gyrus, which holds a pivotal role in decision-making and evaluation^{36, 37}, assumed the top position within the hierarchy.

These findings align with previous research that has demonstrated the flow of information from sensory cortices to the medial temporal lobe, associative areas, and prefrontal regions of the brain^{42, 43}. Moreover, our results are coherent with the several studies which used PCT to investigate perceptual⁴⁴ and automatic memory mechanisms in the brain⁴⁵, as reviewed by Spratling¹⁵. Indeed, they described how information processing, orientation tuning, contour integration, binocular rivalry in the visual system relied on the hierarchical transition of the psychophysical and neurophysiological signals from the retina to the lateral geniculate nucleus and V1⁴⁶⁻⁴⁹. In the context of automatic sensory memory, PCT has also lent support to the interpretation of the MMN as an error signal which aids the brain in adjusting its internal predictive model, returning quantitative evidence of information flow from the primary auditory cortex to the superior temporal gyrus and inferior frontal gyrus⁴⁵. Importantly, our findings expand on these studies by providing quantitative evidence of the hierarchical organisation of the brain during a complex cognitive task, involving episodic memory recognition for musical sequences.

We have reported this in the Discussion section of the revised manuscript (line 599) as well as in response to your previous comments on the necessity of testing with quantitative evidence the predictive coding processes involved in our experimental task.

4. The network in figure 4 raises questions (apart from the typo mentioned above) that I have listed.

a. I accept that inferior temporal cortex and hippocampal cortex are bundled together because of similar time series but they are different places with different types of connections to auditory cortex from the two (more indirect from hippocampus)

You are right, our functional parcellation bundled together regions like the inferior temporal cortex and hippocampus, which have distinct neuroanatomy and structural connectivity.

In response to your suggestion, as well as to the inputs from other Reviewers, we conducted an additional analysis using the Automated Anatomical Labelling (AAL) parcellation²⁰, a widely recognised anatomical parcellation that have been extensively employed by previous MEG studies (e.g.²¹⁻²⁴). This approach allowed us to define more spatially specific ROIs, addressing the issue of grouping together regions with different neuroanatomical features.

With the AAL ROIs, we repeated the same analyses as in the original manuscript, revealing significant differences in brain activity between experimental conditions across several ROIs. Furthermore, as mentioned above, we complemented these results with Dynamic Causal Modelling (DCM) analyses, providing quantitative evidence of the hierarchical structure among six key AAL ROIs that were particularly relevant to the task.

These ROIs closely mirrored those in the original manuscript but were more precisely defined on a spatial level, indexing only the core brain regions that we were interested in and that highly discriminated the experimental conditions in the whole brain contrast analysis. Indeed, the ROIs included left and right Heschl's gyri, left and right hippocampus, bilateral anterior cingulate gyrus, and bilateral medial cingulate gyrus.

Thanks to your comment, we made these adjustments which have not only enhanced the spatial focus of the ROIs but also reinforced the robustness and reliability of our original findings, improving the foundation of our claims. Moreover, using an established parcellation facilitates direct comparisons with existing literature.

In light of these outcomes, the revised version of the manuscript primarily reports the results based on the AAL parcellation. To ensure complete transparency, we have also provided comprehensive and additional information to elucidate how we derived the functional parcellation. However, this is now only reported in supplementary material.

As shown, for example, in **Figures 4** and **S11**, these two procedures showed convergent results, highlighting that the core significance of our findings does not depend on the chosen parcellation.

We report the revised Results section (line 349):

“Automated anatomical labelling (AAL) time series

After evaluating the neural sources of the different brain activity associated with the experimental conditions using the fine-grained 8-mm parcellation of the brain (**Figure 1e**, “8-mm parcellation”), we focused on a set of anatomically defined regions of interest (ROIs) (**Figure 1e**, “AAL parcellation”). Here, we used a standard anatomical parcellation method known as automated anatomical labelling (AAL) ²⁰ and calculated a time series for each of the 90 non-cerebellar ROIs of AAL (**Figure 1f**, left, “Evoked responses”).

As specified above, our source reconstruction was conducted in an 8-mm space, resulting in 3559 brain voxels, each with its own time series. Thus, for each of the 90 ROIs, we identified the corresponding brain voxels and then averaged the time series of these associated voxels. This allowed us to obtain a final time series for each AAL ROI.

Then, we computed one t-test for each AAL ROI, each time-point and each combination of M versus Ns. Finally, we corrected for multiple comparisons using 1D cluster-based MCS (MCS, $\alpha = .05$, MCS *p-value* = .001). Confirming the results from the 8-mm brain parcellation shown in **Figure 3**, this analysis returned several significant clusters of differential brain activity over time between M and Ns. These were primarily localised in auditory cortices (e.g. Heschl's gyrus, superior temporal gyrus) hippocampus and para-hippocampal gyrus, inferior temporal cortex, anterior, medial and posterior cingulate gyrus, ventromedial prefrontal cortex. **Figure S7_{a-g}** and **Tables S6** and **S7** show all AAL ROIs and the significant differences between conditions. **Figures 4** and **S8** shows instead a selected array of ROIs which were particularly relevant for this study. Here, we identified the two ROIs that showed the maximum activity in absolute value among auditory regions (i), medial temporal lobe (ii) and cingulate and prefrontal cortices (iii). We selected these broad regions based on the cognitive processes involved in the experimental task used in this study: audition (i), memory (ii), evaluation and decision-making (iii). As shown by **Figures 4** and **S8**, these ROIs were left and right Heschl's gyrus (LHG, RHG), left and right hippocampus (LHP, RHP), anterior cingulate gyrus (ACC, left and right averaged together) and medial cingulate gyrus (MC, ACC, left and right averaged together), respectively.

In essence, the contrasts showed that M versus N was characterized by stronger activity in RHG, LHP, RHP and ACC at 350 – 450 ms after the onset of each tone. Similarly, M presented stronger negative activity than N in the MC at 400 – 500 ms after the onset of each tone (**Figure 4**). Conversely, late N100 responses localised in LHG were stronger for N versus M. Moreover, LHP, RHP and ACC showed a stronger response for N versus M occurring at 250 – 300 ms after altering the original sequences (**Figure 4**). **Table 3** reports detailed statistics of the largest significant clusters, while complete statistical results of these six ROIs are described in **Table S7**.

Contrast	ROI	Temporal extent of the largest clusters from the 1st tone of the sequence	Peak t-value	P-value
Positive activity				
M versus NT1 (onset deviation NT1: 350 ms)	ACC	580 – 830	7.45	< .001
	LHG	520 – 630	7.17	< .001
	RHP	580 – 780	7.66	< .001
M versus NT2 (onset deviation NT2: 700 ms)	ACC	940 – 1160	7.48	< .001
	LHG	890 – 970	6.26	< .001
	RHP	950 – 1160	6.24	< .001
M versus NT3 (onset deviation NT3: 1005 ms)	ACC	1310 – 1540	6.13	< .001
	RHG	1660 – 1760	5.37	< .001
	RHP	1320 – 1520	6.72	< .001
M versus NT4 (onset deviation NT4: 1400 ms)	ACC	1700 – 1890	6.54	< .001
	LHG	1640 – 1720	6.58	< .001
	RHP	1680 – 1890	6.56	< .001
Negative activity				
M versus NT1	MC	680 – 860	-6.06	< .001
M versus NT2	MC	980 – 1180	-6.17	< .001
M versus NT3	MC	1360 – 1550	-5.96	< .001
M versus NT4	MC	1720 – 1920	-6.17	< .001

Table 3. Largest clusters of stronger activity of memorised (M) versus novel sequences (Ns).

Largest clusters of significantly stronger activity of M versus Ns computed for the six selected automated anatomical (AAL) regions of interest (ROIs): left Heschl's gyrus (LHG), right Heschl's gyrus (RHG), left hippocampus (LHP), right hippocampus (RHP), anterior cingulate gyrus (ACC), medial cingulate gyrus (MC). The table shows the contrast (including the onset of the first varied note in the N sequence), the correspondent ROI, the temporal extent (in ms) of the largest cluster, the peak t-value of the cluster and the associated Monte-Carlo simulations (MCS) p-value. The MC shows stronger negativity since the polarity of the MC signal was primarily negative, thus stronger activity in the MC was reflected in a more pronounced negativity. All the other significant clusters, for both the selected six and the remaining AAL ROIs, are reported in detail in **Tables S6** and **S7**.

In addition, we applied the multivariate source leakage correction method proposed by Colclough and colleagues²⁵ (see Methods for details) to the ROIs time series and computed the same statistical analysis described above. As shown in **Figure S9** and **Table S8**, the results aligned with our initial findings, thereby corroborating their robustness.

Finally, as a further measure to enhance the reliability of our findings, we reported in supplementary material the description of an alternative, functional parcellation derived from the data. As shown by supplementary Figures S7_{a-g}, S10 and S11 and Tables S9, the contrasts across experimental conditions when using the functional parcellation returned results which were highly comparable with AAL,

demonstrating that the significance of our findings did not depend on the chosen parcellation. All the subsequent analyses reported in the manuscript were computed on the AAL parcellation. Moreover, we complemented them by performing the same analyses using the functional parcellation, as reported in the supplementary material (**Figures S15, S16, S17, S18** and **Tables S14, S15**)”.

Additional changes were made to the Methods section (line 1043).

b. There is no analysis to examine effective connectivity between areas which I think is needed in a study that examines predictive coding. There are so many ways to do this including spectral Granger which would allow examination of key oscillatory correlates in different bands. I am not expecting such analyses between all the nodes but between key nodes like hippocampus and AC.

Thank you for your feedback. You rightfully raised the need for an analysis of effective connectivity between key brain areas, especially in the context of a study examining brain hierarchies and predictive coding. We appreciate your suggestion and have carefully considered the available analytical methods for investigating hierarchical processing and effective functional connectivity. In response, we selected Dynamic Causal Modelling (DCM).

DCM is a well-established method in the neuroscientific literature that allows us to infer the causal structure of interconnected brain regions, addressing the hierarchical relationships and predictive coding principles. This approach involves evaluating competing models to explain the generation of time series data and their hierarchical connections.

Our choice of DCM precisely aligns with the focus of this study on predictive coding, as DCM has been primarily used to assess quantitative evidence of brain hierarchy in the framework of predictive coding. Please, find extensive information on this procedure in response to your previous comments on predictive coding and effective connectivity.

In addition, although we complemented our main broadband analyses with time-frequency investigations, the primary emphasis of this study is on the brain functional hierarchies of the broadband neural signal underlying long-term recognition of musical sequences. Thus, on this occasion, we decided not to specifically explore different frequency bands within the DCM analysis. However, we acknowledge that studying oscillations, and in particular cross-frequency coupling in the context of complex recognition processes may represent a valid future perspective. For this reason, in the revised manuscript, we have discussed the potential of using stereo-electroencephalography (SEEG), which is sensitive to a wide range of oscillations, including high gamma, to study high-gamma-theta coupling during music recognition (line 682):

“Finally, the induced-response analysis showed that, after the end of the sequence, alpha and beta bands were stronger for the varied compared to the memorised sequences. Considering the modulation of alpha and beta activity in encoding and working memory tasks described in the scientific literature⁵¹, this finding may suggest that post-stimulus increase of alpha-beta power might be relevant to process novel information. In addition, this analysis revealed a weak yet significant increase in gamma power for the memorised compared to the varied sequences. This is coherent with previous studies which reported increased gamma power during recognition of target stimuli^{52, 53} and, more generally, a modulation of the brain oscillations associated with memory load and complex cognitive functions⁵⁴. Although this study provided initial insights on the time-frequency modulation occurring in the brain during long-term recognition of auditory sequences, further research employing MEG and additional tools such as stereo-electroencephalography (SEEG) is needed to expand our experimental design and conduct cross-frequency coupling analysis, testing whether gamma-theta coupling is connected to long-term recognition of auditory sequences”.

c. In terms of the higher areas implicated here hippocampus is expected. Where is the inferior frontal gyrus? This area is implicated in large numbers of studies of melody

analysis going back to fMRI studies of working memory for melody by Zatorre in 1990s. I guess the analysis here seeks differences between encoded and new which may be less critical in IFG compared to HC but this does merit comment.

You are right, our whole-brain analyses did not show a predominant role for the inferior frontal gyrus (IFG) during long-term recognition of previously memorised and varied musical sequences. As you mentioned, IFG has featured prominently in studies related to pitch analysis, particularly in the context of working memory for melodies. Conversely, in our study, the primary focus was on episodic memory and long-term recognition, and this distinction may account for the relatively limited involvement of IFG in our findings. Indeed, our analyses suggested that the IFG's role in the context of auditory long-term recognition may be less critical compared to the hippocampus.

However, since now we focused on the neural activity associated with AAL ROIs, we were able to specifically investigate IFG. Interestingly, our results did reveal a role for IFG in the recognition of musical sequences, as its activity significantly discriminated between experimental conditions. However, it is worth noting that the activity originating in the IFG was not as pronounced as in other key brain regions highlighted in our manuscript, such as the hippocampus, inferior temporal cortex, anterior cingulate, and ventromedial prefrontal cortex.

In conclusion, your comment has been instrumental in improving the depth of our discussion. In response, we have included a new paragraph in the revised manuscript, which underscores the relevance of IFG and delves into why it may not be as pivotal in the context of long-term recognition as it is in tasks related to pitch evaluation, short-term memory, and working memory (line 640):

"Here, our results suggest that when the upcoming sound of the sequence matched the predicted sound based on the previously stored memory trace, first the auditory cortex and then hippocampus, anterior and medial cingulate gyri responded with positive components. Conversely, when the upcoming sound was incoherent with the prediction made by the brain, a pathway of primarily negative and faster components emerged within the same brain network. It is worth noting that also the inferior frontal gyrus, typically linked to short-term and working memory for music^{71, 72}, displayed significantly different activity across experimental conditions. However, this region exhibited overall reduced responses compared to the other brain regions within the network. This observation further underscores the differentiation between long-term memory recognition for auditory sequences and other memory subsystems".

Reviewer #3 (Remarks to the Author):

Revealing the spacetime hierarchical whole-brain dynamics of auditory predictive coding

Summary:

Bonetti and colleagues capitalize on their previous work and reuse a paradigm where participants have to memorize a melody (5 notes long) from a real piece (Bach's prelude in C minor, BWV847), and then distinguish them from variants of that melody in an old/new task. This paradigm allows them to investigate the predictive coding framework, by comparing the memorized (predicted) and novel (unpredicted) melodies. The novelty of the paper is to be found in the refinement of the paradigm, with systematic variations of the melody (starting from the 2nd, 3rd, 4th or 5th note) and increased time between notes. The authors analyse the brain activity of 83 participants recorded with MEG and claim to observe a network of brain regions involved in the task, as well as a hierarchy of processing through this network.

Overall Comments:

The paradigm used in this study seems efficient at eliciting differentiated brain responses to memorized vs novel melodies at the level of individual notes, and the decoding analyses displayed at the beginning of the result section are convincing.

However, missing information in subsequent analyses (in text, figures and/or method, particularly when defining regions of interest), combined with errors in graph legends and labels, make evaluation and reproduction impossible.

We highly appreciate your thorough report and detailed comments which have strongly contributed to enhancing the quality and clarity of our manuscript. In response to your comments, the following key revisions have been implemented:

1. *Definition of the brain regions of interest (ROIs):*

- a. We conducted an additional analysis using the Automated Anatomical Labelling (AAL) parcellation²⁰, a widely recognised anatomical parcellation that have been extensively employed by previous MEG studies (e.g.²¹⁻²⁴). This approach allowed us to define more spatially specific ROIs, facilitated direct comparisons with existing literature and did not rely on a few arbitrary steps which were followed to define the functional parcellation.
- b. We have provided more details and computed additional analyses which made our procedure for defining the functional parcellation clearer and more solid and reported them in the supplementary material.

2. *Quantitative testing of brain hierarchy and effective connectivity:*

- a. Dynamic Causal Modelling (DCM) has been used to provide quantitative evidence of hierarchical processing in the brain.
- b. To this aim, we have included a new author (Ass. Prof. Martin Dietz) who has extensive DCM expertise. With his help, we implemented the DCM analysis which now supports our claim of hierarchical processing, revealing feedforward and backward connections between auditory cortices, cingulate gyrus and hippocampus which align with predictive coding theory.

3. *Clarification of figures and methods:*

- a. We acknowledge your concern about missing information and a few errors in figures and captions. We have carefully reworked all figures, legends, captions, tables, methods, and results sections to address these issues.

4. *Introduction:*

- a. The hypotheses were clarified, and the Introduction improved according to your suggestions and *Nature Communications'* guidelines.

- b. The rationale behind the changes in the experimental design have been made clearer.
5. *Time-frequency analysis and source reconstruction:*
- a. Time-frequency analyses on MEG channels and the Occipital superior lobe have been added to support the validity of source reconstruction.
 - b. We acknowledge the limitations of source leakage during source reconstruction and have included an additional analysis with source leakage correction, demonstrating the robustness of our results.
 - c. We have clarified how the neural activity index was calculated during our standard source reconstruction procedure.
6. *Empirical testing of claims:*
- a. Claims have been revised to be more grounded in empirical evidence, and statistical tests have been applied to support any assertion.
 - b. Descriptive claims without empirical support have either been removed or revised to align with statistical evidence.
7. *Additional analyses and discussion:*
- a. Following your suggestions, we have computed additional minor analyses and inserted novel paragraphs to better discuss our findings (e.g. we have revealed the impact of musical expertise on the neural data; we have discussed the temporal generalisation results. Moreover, additional details are available below in response to your specific comments).

Our extensive additional clarifications and analyses were complemented by 50 novel panels distributed across 18 supplementary figures and by 15 supplementary tables, providing a better understanding of every detail of our procedures.

Please, find detailed responses to each of your comments below in this letter. To facilitate your review, we have reported in this letter the revised sections/paragraphs of the manuscript in relation to your main comments. For minor comments, we have indicated only the line of the revised manuscript where changes were made to avoid unnecessary length in this letter.

We are confident that these revisions have strengthened the manuscript, addressing all your valuable recommendations. Thank you once again for your time and expertise in reviewing our work and helping us to improve the manuscript.

Detailed answers

The two main claims of the paper are:

1. A broad network of regions involved in the musical task, from auditory to hippocampal and frontal regions. However, this network is not supported by connectivity analyses (Figure 4 is misleading in this respect), and relies solely on a functional ROI definition that is not properly documented in the manuscript.

We agree that in the original version of the manuscript we did not provide sufficiently clear evidence of the hierarchical processing between brain regions. After a careful evaluation of the available analytical methods for studying hierarchical processing and functional connectivity, we opted for the Dynamic Causal Modelling (DCM). DCM was developed by Karl Friston and colleagues within the framework of predictive coding theory³⁰⁻³⁵ to infer the causal structure of interconnected or dispersed dynamical systems such as a network of brain regions. It employs a Bayesian model comparison approach that involves evaluating competing models which explain the generation of time series data and their hierarchical relationship. DCM has been widely used in the neuroscientific literature³⁰⁻³⁵, especially but not limited to the framework of predictive coding, and it is particularly useful for estimating hierarchical relationship between brain regions with regards to experimental stimuli. In relation to our new extensive DCM analyses, we have invited Ass. Professor Martin Dietz to join the paper as an author. In his career, he has primarily been working with DCM applied to EEG, MEG and fMRI data and has a wide expertise in analysing this data and interpreting the outcomes of DCM.

We also agree that our description of the functional ROI parcellation that we used was not sufficiently explained. In response to your suggestion, we took two significant steps to enhance our approach:

1. We conducted an additional analysis using the Automated Anatomical Labelling (AAL) parcellation²⁰, a widely recognised anatomical parcellation that have been extensively employed by previous MEG studies (e.g.²¹⁻²⁴). This approach allowed us to define more spatially specific ROIs, facilitated direct comparisons with existing literature and did not rely on a few arbitrary steps which were followed to define the functional parcellation.
2. We have provided more details and computed additional analyses which made our procedure for defining the functional parcellation clearer and more solid.

As detailed below in response to your specific comment on the functional parcellation, we computed the same analyses for both the AAL parcellation and the functional parcellation, obtaining converging results.

In light of these outcomes, and also following the comments provided by the other Reviewers, in the revised version of the manuscript we have reported the results based on the AAL parcellation. In addition, to ensure complete transparency and additional information which may be relevant for the readers, we have also improved the description of our previous analysis using the functionally derived parcellation and moved it to the supplementary material.

Please, find our complete answers below, after your specific comments on the brain hierarchies (i) and the functional parcellation (ii).

2. A hierarchy of processing within this network. However, this is only supported by descriptive statistics showing a fixed delay in response between the left auditory cortex and the ventro-medial prefrontal cortex (2 of the 6 ROIs defined in the paper). The observation itself is not especially novel and gives very little support to the claim of a processing hierarchy.

You are right, this provides little support for our claim of a processing hierarchy.

For this reason, as described above, we have computed a novel analysis using DCM.

Our DCM analysis confirmed the principles of predictive coding. It revealed feedforward connections from sensory regions (auditory cortices) to higher-order areas (hippocampi, anterior cingulate and medial cingulate gyri), as well as backward connections from higher-order regions to sensory regions. This aligns with the idea that predictions originate in higher-order areas and are transmitted to sensory regions. Additionally, our analysis demonstrated that sensory regions send inputs to higher-order areas after initial stimulus processing, supporting previous evidence, such as Rocchi and colleagues⁴³ that you rightfully recommended.

In addition, our findings not only aligned with existing predictive coding literature but also significantly expanded on it. In fact, they provided quantitative evidence of predictive coding and fast-scale brain hierarchies during a complex cognitive task, involving episodic memory recognition for musical sequences.

Here, we showed that hippocampi, medial and anterior cingulate gyri operated at the same hierarchical level, receiving feedforward signals from the auditory cortex and simultaneously sending back backward signals. This hierarchical alignment persisted consistently throughout the entire sequence, except for the final tone. Here, the cingulate gyrus, which holds a pivotal role in decision-making and evaluation^{36, 37}, assumed the top position within the hierarchy. For the memorised sequences, the hierarchy consisted of the auditory cortex sending feedforward signals to the hippocampus and anterior cingulate gyrus, and subsequently to the medial cingulate gyrus (and receiving feedback signals in the opposite direction). Similarly, for the varied sequences, the hierarchy showed feedforward connections from the auditory cortex to the hippocampus and medial cingulate and then to the anterior cingulate (and feedback in the opposite direction). This result suggests that, when listening to the last tone, the brain might prepare the evaluation of the sequence (i.e. categorising it as 'memorised' or 'novel') and exactly for this reason either the anterior or medial cingulate, both relevant regions for decision-making and evaluative processes^{36, 37}, occupy the highest position in the hierarchy. In addition, our results present a significant dissociation between anterior and medial cingulate, suggesting a role of the medial part of the cingulate for recognition of memorised sequences and of the anterior part for novelty detection.

Extensive details on the DCM procedure, results and discussions are provided in the revised manuscript.

For your convenience, we report as follows the revised Results section (line 473):

“Dynamic causal modelling

In this study, DCM was used to test our hypothesised model of brain hierarchies during recognition of memorised and varied musical sequences against five competing models, which are described in detail in the Methods section and illustrated in **Figure 1g**. These models comprised the six AAL ROIs described in the previous section and depicted in **Figure S8**. Our hypothesised model consisted of feedforward connections from LHG, RHG to LHP, RHP, ACC, MC and feedback connections from LHP, RHP, ACC, MC to LHG, RHG (**Figure 6a**, red box).

Since DCM for MEEG is usually computed on relatively short time-windows (e.g. 350 – 400 ms) and for one experimental perturbation (i.e. sound) at a time^{30, 31}, we computed a series of DCM analyses independently for each tone of the sequences, in a 350 ms time-window from each tone onset. Specifically, we computed an independent DCM for tones two, three, four and five for condition M. In relation to the N conditions, we were interested only in the tone that introduced the variation in the sequences. Thus, we computed four DCMs, organised as follows: tone two for NT1; tone three for NT2; tone four for NT3; tone five for NT4.

To compare alternative models, we estimated the free energy F associated with the computation of each model (model evidence). To study the consistency of F across the population, we used a random-effects Bayesian model selection (RFX BMS) procedure, which provides the posterior probability and protected exceedance probability of each model, indicating which of the alternative models has the strongest model evidence within the whole population. In addition, we computed the Bayesian omnibus risk (BOR), which represents the probability that the protected exceedance probability associated with each model comparison is attributable to chance within the sample of participants (**Figure 6b** and **6c**)

³⁰.

This procedure was observed independently for the eight DCM analyses described above (four for M and four for Ns).

Results indicated that our hypothesised model was the one with the highest evidence for all the DCM analyses of tones two, three and four (**Figure 6**). This result showed a high probability of non-being driven by random chance, as demonstrated by the BOR parameter: $BOR_{M-tone2} = 2.91e-07$; $BOR_{M-tone3} = 4.33e-04$; $BOR_{M-tone4} = .008$; $BOR_{NT1-tone2} = 4.49e-12$; $BOR_{NT2-tone3} = .010$; $BOR_{NT3-tone4} = .003$.

Regarding tone five, the model with the highest evidence for M showed a hierarchical architecture which consisted of the auditory cortex leading to the hippocampus and anterior cingulate gyrus and subsequently to the medial cingulate gyrus ($BOR_{M-tone5} = .006$). For NT4, the model with the highest evidence presented a hierarchy which started in the auditory cortex and led to the hippocampus and medial cingulate gyrus and then to the anterior cingulate gyrus ($BOR_{NT4-tone5} = .036$)".

And in the Discussion section (line 599):

"Our results fit the well-known PCT¹²⁻¹⁴, which states that the brain is constantly updating internal models to predict external information and stimuli. Recently, this theory has been proposed to explain the brain mechanisms underlying complex cognitive processes¹⁵, finding a remarkable example in the neuroscience of music^{13, 14}. However, while PCT holds promise for understanding complex cognitive processes, the quantitative evidence in its favour has been limited. Thus, our study filled this gap by providing quantitative evidence of the brain functional hierarchies during long-term recognition of previously memorised and novel sequences. The results showed feedforward connections from the auditory cortices, representing sensory regions^{38, 39}, to the hippocampus, anterior cingulate gyrus, and medial cingulate gyrus, brain areas which were previously associated with memory and predictive processes^{40, 41}, and evaluation³⁶. Simultaneously, we observed backward connections operating in the opposite direction. This hierarchical alignment persisted consistently throughout the entire sequence, except for the final tone. Here, the cingulate gyrus, which holds a pivotal role in decision-making and evaluation^{36, 37}, assumed the top position within the hierarchy. For the memorised sequences, the hierarchy consisted of the auditory cortex sending feedforward signals to the hippocampus and anterior cingulate gyrus, and subsequently to the medial cingulate gyrus (and receiving feedback signals in the opposite direction). Similarly, for the varied sequences, the hierarchy showed feedforward connections from the auditory cortex to the hippocampus and medial cingulate and then to the anterior cingulate (and feedback in the opposite direction). This result suggests that when listening to the last tone the brain might prepare the evaluation of the sequence (i.e. categorising it as 'memorised' or 'novel') and exactly for this reason either the anterior or medial cingulate, both relevant regions for decision-making and evaluative processes^{36, 37}, occupy the highest position in the hierarchy.

In addition, our results present a significant dissociation between anterior and medial cingulate, suggesting a privileged role of the medial part of the cingulate for recognition of memorised sequences and of the anterior part for novelty detection.

These findings align with previous research that has demonstrated the flow of information from sensory cortices to the medial temporal lobe, associative areas, and prefrontal regions of the brain^{42, 43}. Moreover, our results are coherent with the several studies which used PCT to investigate perceptual⁴⁴ and automatic memory mechanisms in the brain⁴⁵, as reviewed by Spratling¹⁵. Indeed, they described how information processing, orientation tuning, contour integration, binocular rivalry in the visual system relied on the hierarchical transition of the psychophysical and neurophysiological signals from the retina to the lateral geniculate nucleus and V1⁴⁶⁻⁴⁹. In the context of automatic sensory memory, PCT has also lent support to the interpretation of the MMN as an error signal which aids the brain in adjusting its internal predictive model, returning quantitative evidence of information flow from the primary auditory cortex to the superior temporal gyrus and inferior frontal gyrus⁴⁵. Our findings expand on these studies by providing quantitative evidence of the hierarchical organisation of the brain during a complex cognitive task, involving episodic memory recognition for musical sequences. In addition, while the hierarchical architecture involved in recognising both memorised and varied sequences remained largely unchanged, the temporal dynamics, strength, and polarity of the brain signal sharply differed".

Additional changes were made in the Methods section (line 1098).

Moreover, a graphical depiction of the DCM outcome is provided in Figure 6.

Finally, the introduction falls short at explaining the rationale behind the paradigm used, and the discussion is filled with interpretations that are based either on descriptive statistics or on analyses that are not even mentioned in the results or methods sections.

We have better explained the rationale behind our study, the three main hypotheses and the paradigm that we used.

Moreover, since we have been asked to carefully follow the editorial guidelines of *Nature Communications*, we have also provided a succinct summary of the main findings at the end of Introduction, following the hypotheses. In addition, motivated by your comment, we have removed any claim from the discussion that was not supported by clearly described analysis.

For your convenience, we report the fully revised Introduction as follows.

“Introduction

The spatiotemporal features of hierarchical processing in the brain are essential to fully grasp the neural substrates of human perception and cognition, as suggested by the predictive coding theory (PCT) ^{59-63, 12, 13}. To elucidate such brain mechanisms, much research has focused on the visual system ^{64, 65}, which primarily relies on the recognition of patterns arranged in space. Conversely, the auditory system extracts information from patterns and sequences that develop over time ⁶⁶, providing unique opportunities to understand the temporal hierarchies of the brain.

Extensive research spanning decades has established the hierarchical organisation of auditory perception, with a particular emphasis on the processing of elementary auditory stimuli. This hierarchical progression starts at the peripheral level within the cochlea and moves forward to the auditory pathway, which encompasses the brainstem, pons, trapezoid body, superior olivary complex, lateral lemniscus, inferior and medial geniculate nucleus of the thalamus, and ends in the primary auditory cortex ^{67, 68}. However, how auditory information is integrated from the auditory cortex to the whole brain has not been fully established yet.

Aiming to bridge this gap, prior investigations focused on automatic predictive processes within the framework of PCT. These studies assessed auditory automatic prediction error, often relying on well-established event-related potential/field (ERP/F) components such as N100, mismatch negativity (MMN), and error-related negativity (ERAN) ¹⁻⁹. They demonstrated that such components were automatically evoked in response to auditory stimuli, deviations from expected sound features, likelihood of occurrence of musical tones (N100, MMN), and varied harmonic properties (ERAN). Additional studies employed dynamic causal modelling (DCM) to investigate the brain hierarchical architecture during automatic predictive processes, as indexed by MMN. This research has yielded quantitative evidence demonstrating the flow of information from the primary auditory cortex to the superior temporal gyrus and the inferior frontal gyrus ⁴⁵.

Expanding upon these investigations, Rocchi and colleagues provided evidence of hierarchical organisation in the auditory system using direct electrical brain stimulation. They showed effective connectivity from the auditory cortex to regions in the medial temporal lobe and prefrontal cortex, including the hippocampus and ventro-lateral prefrontal cortex ⁴³. Yet, it remains uncertain how the integration of auditory information across the whole brain relates to complex cognitive functions, such as the conscious recognition and prediction of auditory sequences evolving over time.

To address this question, a unique perspective has arisen through the integration of musical memory paradigms with magnetoencephalography (MEG). Music is a complex artform acquiring meaning through the combination of its constituent elements extended over time ⁶⁹, while MEG, particularly when used in combination with MRI, allows to track the brain activity with excellent temporal resolution in the order of milliseconds and acceptable spatial accuracy ⁵⁶. It is precisely due to these attributes that this combination offers a unique framework for investigating the rapid memory and predictive processes of temporal information in the human brain ^{13, 14}.

Indeed, recent studies employing MEG and musical paradigms investigated the brain mechanisms associated with perception, manipulation and recognition of sound sequences. As an example, Albouy and colleagues ⁵⁷ explored the brain activity underlying memory retention, showing that theta oscillations in the dorsal stream of the participants' brain predicted their auditory WM performance. Along this line, we recently uncovered a vast network of brain areas responsible for encoding and recognising previously memorised and novel musical sounds. This network extended from the auditory cortex to the medial cingulate, inferior temporal cortex, insula, frontal operculum, and hippocampus ^{16, 22, 70}. We also observed that the complexity of music ¹⁷ and individual cognitive differences ¹⁸ influenced the activity recorded in this network.

In our previous studies, we utilised relatively rapid musical tones with a duration of 250 ms. However, this approach prevented us from closely tracking the neural dynamics associated with each sound within the sequences, leading to frequent overlap with the brain responses to subsequent tones. For this reason, we were also unable to study the rapid hierarchies between the brain regions that we revealed, which is necessary to understand the neurophysiology of conscious recognition of auditory sequences.

Moreover, while our previous works showed differences in brain activity when comparing previously memorised versus completely novel auditory sequences, we did not investigate the scenario in which an original sequence was systematically varied. This prevented us from exploring how the brain generates prediction errors within the framework of a conscious long-term memory recognition task, as opposed to the several instances of automatic prediction errors described in the literature¹⁻⁹.

To address these questions, in the present study we made two essential modifications to our previously employed experimental paradigm: (i) we compared the original five-tone melodies with systematic variations occurring at the 2nd, 3rd, 4th, or 5th tone; (ii) we increased the duration of the tones from 250 ms to 350 ms. These seemingly minor modifications were of great importance, as they enabled us to directly address our three main hypotheses, stated as follows. First, the conscious recognition of previously memorised auditory sequences relied on the hierarchical recruitment of a widespread brain network including auditory cortex, hippocampus and cingulate gyrus. We expected to observe that these regions responded to each tone forming the sequence, with the auditory cortex preceding the hippocampus and cingulate gyrus. Second, the detection of the varied auditory sequences elicited a prediction error, originating in the auditory cortex and then propagating to hippocampus, anterior cingulate and ventromedial prefrontal cortex. While we expected a consistent response in the auditory cortex to the introduction of varied sounds at any point in the sequence, we hypothesised that the anterior cingulate gyrus and ventromedial prefrontal cortex exhibited their strongest response exclusively when the variation in the sequence was introduced. Third, sequence recognition relied on a brain hierarchy characterised by feedforward connections from auditory cortices to hippocampus and cingulate gyrus, simultaneous with feedback connections in the opposite direction. Moreover, recognition of both previously memorised and varied sequences relied on the same hierarchy, while the temporal dynamics, strength and polarity of brain responses differed.

These hypotheses were confirmed by the results of the experiment, which showed that the brain distinctively responds to each tone forming the previously memorised sequence, with the auditory cortex preceding hippocampus and anterior and medial cingulate gyrus. In addition, the detection of the sequence variations elicits a conscious prediction error in the brain, originating in the auditory cortex and propagating to hippocampus, anterior and medial cingulate. While the auditory cortex has a constant response to the varied sounds introduced at any point in the sequence, the hippocampus and anterior and medial cingulate gyrus exhibit their strongest response exclusively to the sound which introduced the variation in the sequence. Finally, we provide quantitative evidence of the brain functional hierarchies during long-term auditory recognition. Here, as hypothesised, feedforward connections originate from the auditory cortices and extend to the hippocampus, anterior cingulate gyrus, and medial cingulate gyrus, along with feedback connections in the opposite direction. Notably, throughout the sequence, the hippocampus and cingulate gyrus maintain the same hierarchical level, except for the final tone, where the cingulate gyrus assumes the top position within the hierarchy”.

This paper could be drastically improved by:

- reworking the figures, legend and method section, so that no information is missing for potential readers to assess the quality of the analyses or to perform a replication.

Thank you for this suggestion. We have made sure to describe in detail every procedure we followed. Moreover, we have heavily reworked all the figures, legends, captions, and tables, as well as the Methods and Results sections. Furthermore, we have computed additional, supporting analyses and provided 18 supplementary figures to ensure complete transparency in our procedures.

Full replication of the results in this manuscript can be done using the code provided here:

(<https://github.com/leonardob92/LBPD-1.0.git>;

https://github.com/leonardob92/HierarchicalPredictiveCoding_Music_MEG.git)

and using the data available at the following link:

<https://doi.org/10.5281/zenodo.8046524>

- making sure that inferential statistical testing is performed for all the claims and interpretations the authors are willing to make, as these are what allows us to generalize our findings to future observations.

We have made sure to make claims only for results supported by inferential statistics.

- targeting more precisely what are the novel findings in light of the existing literature. We have better clarified the novelty of our findings in light of the existing literature throughout the fully revised Discussion:

“Discussion

This study is rooted in the foundation of previous investigations and provides a deeper understanding of the neural mechanisms and brain’s functional hierarchies underlying the long-term recognition of auditory sequences. Here, in contrast to prior studies, we deliberately reduced the speed of the musical stimuli^{16-18, 22, 70}. This allowed us to reveal that the recognition of auditory sequences relied on distinct brain responses to each tone forming the sequence, as opposed to the slow response lasting for the entire duration of the sequence detected by previous studies^{16-18, 22, 70}. This evidence suggests that the previously described slow signal might have been the summation of the neural responses to each sound of the sequence occurring at a faster tempo (i.e. each sound lasting 250 ms^{16-18, 22, 70} as opposed to the 350 ms employed in the current study).

Furthermore, here we introduced systematic variations to the original memorised sequences to examine the brain mechanisms associated with conscious prediction error. The declarative nature of the prediction error, where participants were asked to identify differences in the sequences and provide a conscious response, extends our understanding beyond the existing body of research focused on automatic prediction error indexed by well-known ERP/Fs components such as N100, MMN and ERAN^{4, 5, 7-9}. This research revealed that such components were primarily generated by the auditory cortex^{10, 11}, reporting a complementary yet much reduced recruitment of the medial cingulate gyrus, inferior frontal gyrus, and frontal and hippocampal regions^{2, 11}. Conversely, in our study we revealed an initial recruitment of the auditory cortex (100 – 150 ms) and then of the anterior and medial cingulate gyrus, ventromedial prefrontal cortex and bilateral hippocampus (250 – 500 ms). Notably, our results showed that Heschl’s gyrus discriminated memorised melodies versus systematic variations but did not distinguish the strength of the errors (i.e., the response was not significantly different across varied tones). Conversely, the prediction error observed in the bilateral hippocampus, anterior and medial cingulate gyrus was significantly stronger in response to the tone that introduced the variations compared to all other tones. Given the different brain responses observed in our study in comparison to previous literature on automatic prediction error, we argue that the brain signature underlying the awareness of the variation might be represented by the responses recorded in the anterior cingulate gyrus, ventromedial prefrontal cortex and hippocampus and their specific temporal dynamics.

Our results fit the well-known PCT¹²⁻¹⁴, which states that the brain is constantly updating internal models to predict external information and stimuli. Recently, this theory has been proposed to explain the brain mechanisms underlying complex cognitive processes¹⁵, finding a remarkable example in the neuroscience of music^{13, 14}. However, while PCT holds promise for understanding complex cognitive processes, the quantitative evidence in its favour has been limited. Thus, our study filled this gap by providing quantitative evidence of the brain functional hierarchies during long-term recognition of previously memorised and novel sequences. The results showed feedforward connections from the auditory cortices, representing sensory regions^{38, 39}, to the hippocampus, anterior cingulate gyrus, and medial cingulate gyrus, brain areas which were previously associated with memory and predictive processes^{40, 41}, and evaluation³⁶. Simultaneously, we observed backward connections operating in the opposite direction. This hierarchical alignment persisted consistently throughout the entire sequence, except for the final tone. Here, the cingulate gyrus, which holds a pivotal role in decision-making and evaluation^{36, 37}, assumed the top position within the hierarchy. For the memorised sequences, the hierarchy consisted of the auditory cortex sending feedforward signals to the hippocampus and anterior cingulate gyrus, and subsequently to the medial cingulate gyrus (and receiving feedback signals in the opposite direction). Similarly, for the varied sequences, the hierarchy showed feedforward connections from the auditory cortex to the hippocampus and medial cingulate and then to the anterior cingulate (and feedback in the opposite direction). This result may suggest that when listening to the last tone the brain is preparing the evaluation of the sequence (i.e. categorising it as ‘memorised’ or ‘novel’) and for this reason either the anterior or medial cingulate, relevant regions for decision-making and evaluative processes^{36, 37}, may need to occupy the highest position in the hierarchy.

In addition, our results present a significant dissociation between anterior and medial cingulate, suggesting a privileged role of the medial part of the cingulate for recognition of memorised sequences and of the anterior part for novelty detection.

These findings align with previous research that has demonstrated the flow of information from sensory cortices to the medial temporal lobe, associative areas, and prefrontal regions of the brain^{42, 43}. Moreover, our results are coherent with the several studies which used PCT to investigate perceptual⁴⁴ and automatic memory mechanisms in the brain⁴⁵, as reviewed by Spratling¹⁵. Indeed, they

described how information processing, orientation tuning, contour integration, binocular rivalry in the visual system relied on the hierarchical transition of the psychophysical and neurophysiological signals from the retina to the lateral geniculate nucleus and V1 ⁴⁶⁻⁴⁹. In the context of automatic sensory memory, PCT has also lent support to the interpretation of the MMN as an error signal which aids the brain in adjusting its internal predictive model, returning quantitative evidence of information flow from the primary auditory cortex to the superior temporal gyrus and inferior frontal gyrus ⁴⁵. Our findings expand on these studies by providing quantitative evidence of the hierarchical organisation of the brain during a complex cognitive task, involving episodic memory recognition for musical sequences. In addition, while the hierarchical architecture involved in recognising both memorised and varied sequences remained largely unchanged, the temporal dynamics, strength, and polarity of the brain signal sharply differed. Here, when the upcoming sound of the sequence was matched with the predicted sound based on the previously stored memory trace, first the auditory cortex and then hippocampus, anterior and medial cingulate gyri responded with positive components. Conversely, when the upcoming sound was incoherent with the prediction made by the brain, a pathway of primarily negative and faster components emerged within the same brain network. It is worth noting that also the inferior frontal gyrus, typically linked to short-term and working memory for music ^{71, 72}, displayed significantly different activity across experimental conditions. However, this region exhibited overall reduced responses compared to the other brain regions within the network. This observation further underscores the differentiation between long-term memory recognition for auditory sequences and other memory subsystems.

An additional finding pertains to our utilisation of multivariate pattern analysis and temporal generalisation techniques, as described by King and Dehaene ⁷³. Here, we revealed that when decoding memorised from novel sequences, the algorithm could differentiate the conditions starting from the first tone that was varied in the novel sequences. This learning could then be generalised by the algorithm with significant (yet reduced) accuracy for all the subsequent tones. This pattern was consistently observed across all categories of variations, and it implies that the differential processes between the two conditions remained consistent over time, indicating that the brain not only identifies the initial varied tone but also monitors the entire auditory sequence. This monitoring might have served the purpose of assessing whether the varied tone represented a transient mistake or the beginning of a proper varied sequence.

Our findings revealed a positive overall correlation between musical training and neural activity associated with the recognition of memorised and varied musical sequences. This correlation was particularly evident with regards to the neural responses to the sounds that introduced variations. This aligns with previous research indicating a connection between musical expertise and altered brain characteristics, including enhanced neural responses (e.g. enhanced auditory automatic prediction error) ^{2, 14, 74} and structural changes ⁷⁵. In this study, what distinguishes our findings was the extension of this relationship to the domain of long-term recognition of musical sequences and the underlying brain mechanisms.

On a behavioural level, both reaction times and accuracy remained consistent across memorised and three categories of novel sequences: NT1, NT2, and NT3. However, accuracy was significantly reduced, and reaction times increased for NT4. This shift might be attributed to the altered final tone in the sequence, eliciting a slower behavioural prediction error. Alternatively, the reduced accuracy and prolonged reaction times in NT4 could be linked to chunking; the mental grouping process that occurs when sequences are presented with a beat every four tones. In this view, after listening to four tones of the memorised sequence (corresponding to a complete beat), the perception that the sequence belonged to the group of previously learned sequences was very strong and, especially much stronger than after only three tones. For this reason, we did not observe a linear increase in reaction times and accuracy but only a difference between T4 and all the other conditions. Currently, we do not have enough data to make definitive claims and future studies are needed to systematically vary the length of the sequences.

Finally, the induced-response analysis showed that, after the end of the sequence, alpha and beta bands were stronger for the varied compared to the memorised sequences. Considering the modulation of alpha and beta activity in encoding and working memory tasks described in the scientific literature ⁵¹, this finding may suggest that post-stimulus increase of alpha-beta power might be relevant to process novel information. In addition, this analysis revealed a weak yet significant increase in gamma power for the memorised compared to the varied sequences. This is coherent with previous studies which reported increased gamma power during recognition of target stimuli ^{52, 53} and, more generally, a modulation of the brain oscillations associated with memory load and complex cognitive functions ⁵⁴. Although this study provided initial insights on the time-frequency modulation occurring in the brain during long-term recognition of auditory sequences, further research employing MEG and additional

tools such as stereo-electroencephalography (SEEG) is needed to expand our experimental design and conduct cross-frequency coupling analysis, testing whether gamma-theta coupling is connected to long-term recognition of auditory sequences. This would also provide supporting evidence of the quality of the source reconstruction performed in this study since SEEG data is less sensitive to source leakage artifacts than MEG source reconstruction algorithms⁷⁶.

In conclusion, this study reveals the brain mechanisms and hierarchical dynamics of long-term memory recognition of previously memorised and varied auditory sequences, effectively integrating the PCT in the domain of complex cognitive tasks”.

Major Comments:

- The introduction clearly states that the upgrades of this paradigm are the systematic variations of the melody and the 350 ms delay between notes. While the 350 ms delay is said to improve the results by avoiding a superposition of auditory responses, it is not clear why the systematic variations of melody were done this way, what the authors expected to see from that modification of the paradigm, and what it did bring to the MEG results compared to previous studies.

The introduction of systematic variations starting from tone 2 of the sequence was a deliberate choice, motivated by the need to explore a specific scenario that our previous works did not cover. While our earlier studies demonstrated significant differences in brain activity between previously memorised and completely novel auditory sequences, they did not allow us to investigate how the brain generates prediction errors within the framework of a conscious long-term memory recognition task.

By adopting this modified paradigm, we could distinguish the brain mechanisms underlying conscious prediction error from the well-established evidence of automatic prediction errors described in the literature¹⁻⁹. Indeed, previous studies indexed automatic prediction error by using well-known ERF/Fs components such as N100, MMN and ERAN^{4, 5, 7-9}. They revealed that such components were primarily generated by the auditory cortex^{10, 11}, reporting a complementary yet much reduced recruitment of the medial cingulate gyrus, inferior frontal gyrus, and frontal and hippocampal regions^{2, 11}. Conversely, in our study we investigated the prediction error in the context of conscious long-term memory recognition, revealing an initial recruitment of the auditory cortex (100 – 150 ms) and then of the anterior and medial cingulate gyrus, ventromedial prefrontal cortex and hippocampus (250 – 500 ms). Moreover, our results showed that Heschl's gyrus discriminated memorised melodies versus systematic variations but did not distinguish the strength of the errors (i.e., the response was not significantly different across varied tones). Conversely, the prediction error observed in the bilateral hippocampus, anterior cingulate gyrus and medial cingulate gyrus was significantly stronger in response to the tone that introduced the variations compared to all other tones. Given the different brain responses observed in our study in comparison to previous literature on automatic prediction error, we argue that the brain signature underlying the awareness of the variation might be represented by the responses recorded in the anterior cingulate gyrus, ventromedial prefrontal cortex and hippocampus and their specific temporal dynamics.

We have clarified the rationale behind the development of the experimental paradigm in the revised Introduction. Moreover, we have improved the discussion of our findings, making sure to only discuss results supported by inferential statistics.

Please, refer to the revised Introduction and Discussion in response to your previous comments.

- In the introduction, I would suggest avoiding to justify the present work with sentences like “little is known about ” (p.3 l.50) or “much remains to be learned” (p3. l.57). Stating that little is known about something is not sufficient to explain why you decided to study a phenomenon (“there are so many phenomena that we know little about, why did you

choose to focus on this one?”). More generally, the introduction falls short at explaining the stakes of this work.

We acknowledge your point about avoiding general expressions like 'little is known about' and 'much remains to be learned.' In light of your feedback, we have made revisions to the introduction to provide a more concrete and explicit rationale for our study. We have replaced these expressions with specific statements that better elucidate the motivations and significance of our work. Additionally, we have taken steps to improve the connections between sentences in the Introduction to provide a clearer and more coherent flow of information. As you could read above in response to your previous comments, we believe these changes have strengthened the introduction.

- In the section “Neural sources of the differential brain activity between M and N”, it reads “we considered [...] the inspection of the MEG data after preprocessing”, but this inspection is not described in the main text nor in the method. Even if some steps of analysis rely on visual inspection, it has to be stated clearly for transparency and replicability.

In the same paragraph, it is unclear where the temporal windows come from. For instance, the window “1.75-1.85 sec” does not correspond to a significant temporal cluster, in particular in Fig. 2.b.

Because of this lack of information, it is impossible to assess whether the analysis is valid or not.

We agree and we have completely reshaped this section of the manuscript as you can see in response to your other comments.

In the method section “Functional regions of interests (ROIs)”, there is not enough information to understand how the ROIs are defined. First, the procedure here builds upon the spatial clusters from the previous section. However, the 14 clusters reported in table 2 are now grouped into 4 “broad regions” without any explanations. Second, it reads that “t-values [were computed] for each brain voxel and each timepoint contrasting M versus N” (p.31 l.752) without specifying which N condition (from NT1 to NT4) was used or how they were grouped. Finally, assuming that the peaks of t-values of each of the 4 broad regions were defined temporally (and not spatially), it would have been useful to report the timing of these peaks (do they correspond to the initial auditory burst? or to the response to a specific note onset? or an activity at a timing unrelated to auditory responses?).

You correctly pointed out that our original explanation of the functional ROI parcellation method was insufficient. After a careful evaluation, in response to your comment, as well as suggestions from other Reviewers, we decided to shift our focus to using an established anatomical parcellation instead of the functionally derived one.

Specifically, we adopted the widely recognised Automated Anatomical Labelling (AAL) parcellation²⁰, which was previously employed in several MEG studies (e.g.²¹⁻²⁴). This approach allowed us to define more spatially specific ROIs and provided the advantage of direct comparisons with existing literature. Furthermore, the AAL parcellation, unlike the functional parcellation, provides a more transparent and reliable approach by eliminating the need for a few arbitrary steps that were inherent in the functional parcellation. Thus, in the revised manuscript, we replaced the analyses based on the original functional parcellation with the AAL ROIs.

To achieve this, we conducted a series of steps, described as follows.

First, as specified in the manuscript, the source reconstruction was conducted in an 8-mm space, resulting in 3559 brain voxels, each with its own time series. Afterwards, for each of the 90 ROIs, we identified the corresponding brain voxels and then averaged the time series of these associated voxels. This allowed us to obtain a final time series for each AAL ROI. Then, we computed one t-test for each AAL ROI, each time-point and each

combination of M versus Ns. Finally, we corrected for multiple comparisons using one-dimensional (1D) cluster-based MCS (MCS, $\alpha = .05$, MCS p -value = .001).

Confirming the results from the 8-mm brain parcellation shown in **Figure 3**, this analysis returned several significant clusters of differential brain activity over time between M and Ns. These were primarily localised in auditory cortices (e.g. Heschl's gyrus, superior temporal gyrus) hippocampus and para-hippocampal gyrus, inferior temporal cortex, anterior, medial and posterior cingulate gyrus, ventromedial prefrontal cortex. **Figure S7a-g** and **Table S6** show all AAL ROIs and the significant differences between conditions. **Figures 4** and **S8** shows instead a selected array of ROIs which were particularly relevant for this study. Here, we identified the two ROIs that showed the strongest reconstructed activity among auditory regions (i), medial temporal lobe (ii) and cingulate and prefrontal cortices (iii). We selected these brain regions based on the cognitive processes involved in the experimental task used in this study: audition (i), memory (ii), evaluation and decision-making (iii). As shown by **Figures 4** and **S8**, the selected AAL ROIs were left and right Heschl's gyrus (LHG, RHG), left and right hippocampus (LHP, RHP), bilateral anterior cingulate gyrus (ACC) and bilateral medial cingulate gyrus (MC), respectively. These six regions were also used for the Dynamic Causal Modelling (DCM) analysis described later.

In conclusion, using the AAL parcellation offered a refined spatiotemporal extent of the ROIs and their time series and thus provided results which were improved and more easily interpretable and comparable with previous literature. However, it is still worth noting that the AAL results aligned with the ones obtained using the functional parcellation (FP). This can be further noticed by comparing the provided figures showing the results for the AAL and the functional parcellation: e.g. **Figure 4** (AAL) – **Figure S7a-g** and **Figure S11**(FP); **Figure 6** (AAL) – **Figure S16** (FP); **Figure 7** (AAL) – **Figure S17** (FP); **Figure S12** (AAL) – **Figure S18** (FP).

For your convenience, we report as follows the revised Results section (line 349):

“Automated anatomical labelling (AAL) time series

After evaluating the neural sources of the different brain activity associated with the experimental conditions using the fine-grained 8-mm parcellation of the brain (**Figure 1e**, “8-mm parcellation”), we focused on a set of anatomically defined regions of interest (ROIs) (**Figure 1e**, “AAL parcellation”). Here, we used a standard anatomical parcellation method known as automated anatomical labelling (AAL)²⁰ and calculated a time series for each of the 90 non-cerebellar ROIs of AAL (**Figure 1f**, left, “Evoked responses”).

As specified above, our source reconstruction was conducted in an 8-mm space, resulting in 3559 brain voxels, each with its own time series. Thus, for each of the 90 ROIs, we identified the corresponding brain voxels and then averaged the time series of these associated voxels. This allowed us to obtain a final time series for each AAL ROI.

Then, we computed one t-test for each AAL ROI, each time-point and each combination of M versus Ns. Finally, we corrected for multiple comparisons using 1D cluster-based MCS (MCS, $\alpha = .05$, MCS p -value = .001). Confirming the results from the 8-mm brain parcellation shown in **Figure 3**, this analysis returned several significant clusters of differential brain activity over time between M and Ns. These were primarily localised in auditory cortices (e.g. Heschl's gyrus, superior temporal gyrus) hippocampus and para-hippocampal gyrus, inferior temporal cortex, anterior, medial and posterior cingulate gyrus, ventromedial prefrontal cortex. **Figure S7a-g** and **Tables S6** and **S7** show all AAL ROIs and the significant differences between conditions. **Figures 4** and **S8** shows instead a selected array of ROIs which were particularly relevant for this study. Here, we identified the two ROIs that showed the maximum activity in absolute value among auditory regions (i), medial temporal lobe (ii) and cingulate and prefrontal cortices (iii). We selected these broad regions based on the cognitive processes involved in the experimental task used in this study: audition (i), memory (ii), evaluation and decision-making (iii). As shown by **Figures 4** and **S8**, the selected AAL ROIs were left and right Heschl's gyrus (LHG, RHG), left and right hippocampus (LHP, RHP), anterior cingulate gyrus (ACC, left and right averaged together) and medial cingulate gyrus (MC, ACC, left and right averaged together), respectively.

In essence, the contrasts showed that M versus N was characterized by stronger activity in RHG, LHP, RHP and ACC at 350 – 450 ms after the onset of each tone. Similarly, M presented stronger negative activity than N in the MC at 400 – 500 ms after the onset of each tone (**Figure 4**). Conversely,

late N100 responses localised in LHG were stronger for N versus M. Moreover, LHP, RHP and ACC showed a stronger response for N versus M occurring at 250 – 300 ms after altering the original sequences (**Figure 4**). **Table 3** reports detailed statistics of the largest significant clusters, while complete statistical results of these six ROIs are described in **Table S7**.

Contrast	ROI	Temporal extent of the largest clusters from the 1st tone of the sequence	Peak t-value	P-value
Positive activity				
M versus NT1 (onset deviation NT1: 350 ms)	ACC	580 – 830	7.45	< .001
	LHG	520 – 630	7.17	< .001
	RHP	580 – 780	7.66	< .001
M versus NT2 (onset deviation NT2: 700 ms)	ACC	940 – 1160	7.48	< .001
	LHG	890 – 970	6.26	< .001
	RHP	950 – 1160	6.24	< .001
M versus NT3 (onset deviation NT3: 1005 ms)	ACC	1310 – 1540	6.13	< .001
	RHG	1660 – 1760	5.37	< .001
	RHP	1320 – 1520	6.72	< .001
M versus NT4 (onset deviation NT4: 1400 ms)	ACC	1700 – 1890	6.54	< .001
	LHG	1640 – 1720	6.58	< .001
	RHP	1680 – 1890	6.56	< .001
Negative activity				
M versus NT1	MC	680 – 860	-6.06	< .001
M versus NT2	MC	980 – 1180	-6.17	< .001
M versus NT3	MC	1360 – 1550	-5.96	< .001
M versus NT4	MC	1720 – 1920	-6.17	< .001

Table 3. Largest clusters of stronger activity of memorised (M) versus novel sequences (Ns).

Largest clusters of significantly stronger activity of M versus Ns computed for the six selected automated anatomical (AAL) regions of interest (ROIs): left Heschl's gyrus (LHG), right Heschl's gyrus (RHG), left hippocampus (LHP), right hippocampus (RHP), anterior cingulate gyrus (ACC), medial cingulate gyrus (MC). The table shows the contrast (including the onset of the first varied note in the N sequence), the correspondent ROI, the temporal extent (in ms) of the largest cluster, the peak t-value of the cluster and the associated Monte-Carlo simulations (MCS) p-value. The MC shows stronger negativity since the polarity of the MC signal was primarily negative, thus stronger activity in the MC was reflected in a more pronounced negativity. All the other significant clusters, for both the selected six and the remaining AAL ROIs, are reported in detail in **Tables S6** and **S7**.

In addition, we applied the multivariate source leakage correction method proposed by Colclough and colleagues²⁵ (see Methods for details) to the ROIs time series and computed the same statistical

analysis described above. As shown in **Figure S9** and **Table S8**, the results aligned with our initial findings, thereby corroborating their robustness.

Finally, as a further measure to enhance the reliability of our findings, we reported in supplementary material the description of an alternative, functional parcellation derived from the data. As shown by supplementary Figures S7_{a-g}, S10 and S11 and Tables S9, the contrasts across experimental conditions when using the functional parcellation returned results which were highly comparable with AAL, demonstrating that the significance of our findings did not depend on the chosen parcellation. All the subsequent analyses reported in the manuscript were computed on the AAL parcellation. Moreover, we complemented them by performing the same analyses using the functional parcellation, as reported in the supplementary material (**Figures S15, S16, S17, S18** and **Tables S14, S15**)”.

Additional changes were made in the Methods section (line 1043).

Finally, while we shifted our primary focus toward the ROIs of the anatomical AAL parcellation, in the interest of enhancing the scientific discussion following your thorough and helpful revision, we have also improved the description and performed additional analyses related to the procedures required for obtaining the functional parcellation.

However, it is important to state that this detailed information is now available exclusively in the supplementary material for the interest of you and potentially the readers, but it should no longer be considered a primary outcome of our work.

- The results in Table 3 show a stronger activity for M vs N (i.e. $M > N$) for most ROIs, except in the cingulate cortex. This result seems contradictory with the predictive coding hypothesis (the responses to surprising notes in N are expected to show a greater amplitude than in M), and with the interpretation of the results shown in Fig. 3 (for instance: “late N100 responses localized in AC were stronger for N versus M”, p.12 l.266). This should be addressed in the discussion.

In the table, the “Peak t-value” for MC is negative because the polarity of the signal was negative for both M and N. Since in absolute terms the signal was stronger for M than N (i.e. it was ‘more negative’ for M), when computing the contrasts, the emerging t-values were negative. This indeed indicates that M had a stronger signal in absolute terms.

We have clarified this in the revised manuscript.

Moreover, the improved analyses reported throughout the manuscript refined the evidence that this region also played a role in detecting the novel sequences. This is shown by the prediction error signal peaking at approximately 250 ms after the onset of the varied tones. Here, the prediction error of the MC was expressed by a positive component, in contrast to the negative signals originated in the ACC, LHP and RHP. To be noted, although different in polarity, the signal originated in MC, ACC, LHP and RHP were simultaneous and in response to the same varied tones. This result is illustrated in **Figure 5** and described in detail in response to one of your following comments.

- In Fig. 3:

- The y-axis label “Neural activity index” is not described (in legend, main text or method). This is concerning, since it is one of the main results of the paper.

Thank you for highlighting this issue.

The “Neural activity index” is simply the output of the source reconstruction, corrected for the spatially varying bias which leads to an overestimation of the variance in the centre of the brain ^{26, 27}. The correction consists of normalising the weights computed by the beamforming algorithm by using a noise covariance matrix between the MEG sensors. This matrix can either be measured or estimated. We used the weights normalization technique described by Luckhoo and colleagues ²⁷, which is an effective and standard solution to correct for the bias towards the centre of the brain during source reconstruction.

Please, note that “Neural activity index” is a pure number (i.e. it does not have any unit of measurement).

Following your suggestion, we have specified in the source reconstruction sections that the time series outputted by the source reconstruction procedure were named “Neural activity index”.

Line 986:

“Finally, the weights were applied to each brain source and time-point. To be noted, the covariance matrix C was computed on the continuous signal, which was estimated by concatenating the trials of all experimental conditions. The weights were applied to the brain data associated with each condition and normalised according to Luekhoo and colleagues²⁶⁻²⁹. For counterbalancing the reconstruction bias towards the centre of the head. The weights were applied to the neural activity averaged over trials for the evoked responses and to the neural activity of each independent trial for the induced responses. This procedure returned a time series for each of the 3559 brain sources (and each trial in the case of induced responses), commonly referred to as “neural activity index”^{26, 27}”.

And line 277:

“Neural sources of the MEG channels peak activity

First, we computed source reconstruction by employing a local-spheres forward model and a beamforming approach as inverse solution in an 8-mm grid (corresponding to 3559 brain voxels), using magnetometers only (**Figure 1e**). This procedure returned a time series for each of the reconstructed brain voxel, which is commonly referred to as neural activity index (see Methods for extensive details on this standard procedure)”.

- In the legend, the M and NT1 curves are described as “M vs NT1” and “NT1 vs M” contrasts. Shouldn’t these two curves be symmetrical, or similar in some ways? Or is it simply a wrong legend?

We have rewritten the caption to avoid this misleading description.

Now, we have clarified that this figure (now coded as **Figure 4**) illustrates the source-localised brain activity averaged over participants for each experimental condition (M, NT1, NT2, NT3, NT4) within six selected AAL ROIs. We have also improved the graphical depiction of the significant time-points emerged contrasting M versus each category of N (i.e. M versus NT1, M versus NT2, M versus NT3, M versus NT4) and correcting for multiple comparisons. Now, different hues of blue-black horizontal lines highlight specific M versus N condition comparisons. For example, the lightest blue represents M versus NT1, the second lightest blue represents M versus NT2, and so on.

We have reported the revised caption as follows.

“**Figure 4. Source-localised differences in evoked responses across experimental conditions**

*This figure illustrates the source-localised brain activity averaged over participants for each experimental condition (memorised [M], novel T1 [NT1], NT2, NT3, NT4) within six selected automated anatomical labelling (AAL) regions of interest (ROIs): left Heschl’s gyrus (LHG), right Heschl’s gyrus (RHG), left hippocampus (LHP), right hippocampus (RHP), anterior cingulate gyrus (ACC), and medial cingulate gyrus (MC). The shaded areas indicate standard errors. Musical tone sketches indicate the onset of the sounds forming the sequences. Brain templates depict the spatial extent of the selected ROIs. Blue-black lines highlight the temporal extent of the significant differences between M versus each category of Ns (i.e. M versus NT1, M versus NT2, M versus NT3, M versus NT4). Different hues of blue-black correspond to specific M versus N condition comparisons. For a comprehensive depiction of all AAL ROIs, please refer to **Figure S7a-g**. For a detailed statistical report on significant differences between experimental conditions in all ROIs, consult **Table S6**.*

- The timing of the “late N100” mentioned in the text is not clearly shown on the figure. Plotting vertical lines at the exact timing of note onsets may help.

Since we appreciate your suggestion, we attempted to add vertical lines at the timing of note onsets for clarity. However, it resulted in a crowded and less visually clear representation. Therefore, in this particular instance, we have opted to maintain the current format.

- All regions of interest display an initial auditory burst after the first note onset. The fact that it is observed in all ROIs (even the cingulate cortex) raises suspicions about the source reconstruction procedure.

The initial auditory burst observed in all regions of interest can be attributed to a phenomenon known as source leakage, which is a recognised challenge in the source reconstruction procedure²⁵. The leakage of the signal happens because all MEG channels record activity from all brain regions. Thus, when there is a very strong neural response which originates in the true source X (e.g. the N100 to the first sounds in our study, originating in the auditory cortex), that response is not recorded only by the MEG channels very close to X, but by all MEG channels (with decreasing amplitude as a function of the distance of the true source X from the MEG channels). Since the source reconstruction is a linear transformation from sensor to source space, this phenomenon is linearly reflected also in the neural sources which are reconstructed.

For example, the strong N100 response to the first sound in our study, originating in the auditory cortex, was reconstructed not only in the auditory cortex, but also in surrounding regions. However, it is essential to emphasize that the amplitude of the reconstructed N100 in the auditory cortex was significantly stronger than in other brain regions. This demonstrates that our source reconstruction correctly estimated the auditory cortices as the primary sources of the N100. Moreover, the additional analysis that we performed using the AAL ROIs showed even more clearly how the focus of the reconstruction of the N100 was precisely the Heschl's gyrus, aligning very well with notions from previous literature^{7, 38, 39}.

It is worth noting that source leakage is a well-known challenge in MEG research, and various quantitative methods have been proposed to address it. For instance, Colclough et al. (2015)²⁵ introduced a symmetric orthogonalization method to correct for artificial correlations between brain ROIs introduced during source reconstruction. While this correction effectively reduces the artificial leakage of brain signals into different regions, it also removes some genuine neural signal, leading to an ongoing debate in the field about the necessity of such corrections.

Consequently, many MEG (and EEG) studies abstain from applying source leakage correction methods due to the potential risk of inadvertently removing genuine neural signal. However, here, motivated by your suggestion, we computed an additional analysis for the supplementary material of the manuscript which shows that our results remain consistent whether or not source leakage correction is applied, further supporting the robustness of our procedures.

The results of this procedure are illustrated in the following figures and tables:

Figure 4 and **Table S7** – Original time series AAL

Figure S9 and **Table S8** – Source leakage corrected time series AAL

Figure S11 and **Table S9** – Original time series functional parcellation

Figure S15 and **Table S14** – Source leakage corrected time series functional parcellation

Additional details are available in the Results and Methods sections.

Results (line 399):

“In addition, we applied the multivariate source leakage correction method proposed by Colclough and colleagues²⁵ (see Methods for details) to the ROIs time series and computed the same statistical analysis described above. As shown in **Figure S9** and **Table S8**, the results aligned with our initial findings, thereby corroborating their robustness”.

Methods (line 1066):

“In addition, since it has been shown that the source reconstruction can produce leakage of the brain signals between the ROIs²⁵, we took additional measures to ensure the robustness of our results. Specifically, we applied the multivariate source leakage correction method proposed by Colclough and colleagues²⁵ to the ROIs time series. This method consists of removing the zero-lag correlations between the ROIs time series through a symmetric, multivariate orthogonalization process. After

computing this correction, we performed the exact same statistical tests described above. The outcomes of this analysis are illustrated in **Figure S9** and described in **Table S8**".

In addition, in the discussion, we have acknowledged source leakage as a common limitation of MEG and have suggested that future studies employing stereo-electroencephalography (SEEG) could help corroborate our findings (line 692):

" [...] further research employing MEG and additional tools such as stereo-electroencephalography (SEEG) is needed to expand our experimental design and conduct cross-frequency coupling analysis, testing whether gamma-theta coupling is connected to long-term recognition of auditory sequences. This would also provide supporting evidence of the quality of the source reconstruction performed in this study since SEEG data is less sensitive to source leakage artifacts than MEG source reconstruction algorithms ⁷⁶".

• In Fig. 4:

• While I appreciate the visual clarity of Fig. 4.a, the "brain network" shown is not supported by any connectivity measures and is purely descriptive. Moreover, the "high-order regions" and "low-order regions" labels are interpretative, and not derived from data nor analyses. Finally, since the ROIs may not be defined properly, the only part of this network that is based on data may be compromised. This is especially important, as one of the main claims of the paper is based on that figure.

• Similarly, Fig. 4.b and 4.c are only descriptive, and do not show any inferential statistics to support the finding of a temporal hierarchy of processing. Moreover, the fixed delay in response to surprising notes is shown only between left AC and VMPFC (only 2 of the 6 ROIs displayed in the network in Fig. 4.a). It would have been relevant to replicate the analysis along the connections hypothesized above (and at the very least with right AC and VMPFC). Finally, the authors say that the "strength of the signal increases over time for VMPFC" in the legend and in the discussion (p.18, l.368), but this not visible from the figure and not supported by any analysis (descriptive or inferential). Again, this is especially important, as the second main claims of the paper is based on that figure. Finally, a delay in response between "low" and "high" order regions of the auditory system (i.e. from auditory to frontal) is not a novel finding (for instance, see Fig. 7 from Rocchi et al., 2021).

You are right. We acknowledge that the original version of the manuscript provided little support for our claim of a brain hierarchical processing underlying long-term recognition of auditory sequences. For this reason, we have computed a novel analysis using Dynamic Causal Modelling (DCM).

Extensive details on the DCM procedure, results and discussions are provided in the revised manuscript, in **Figure 6**, and in this letter in response to some of your previous comments. Moreover, we have inserted the reference to Rocchi et al., 2021 in the revised manuscript (e.g. line 65 and line 625).

• In Fig. 5, the time-frequency analysis shows similar response patterns in all ROIs, which again casts some doubts on the validity of the source reconstruction procedure. Furthermore, the figures do not show which data points are significant or not (or all the points are significant?), which is not acceptable.

The significant results were reported in **Table ST5** of the original manuscript; however, you are right, it is important to also show the significant time-points and frequencies in the figures. For this reason, we have added to the previous panels new figures showing only the significant time windows and frequencies emerging after contrasting the experimental conditions. This highlighted especially the stronger power in alpha and beta bands in the 'novel' conditions compared to the 'memorised' condition after the end of the musical sequence.

To show that the source reconstruction was accurate, we have performed two additional procedures:

1. We have computed time-frequency analysis on eight MEG channels which covered fronto-temporal and occipital regions. The results were highly consistent with the findings reported for the ROIs. They are illustrated in **Figure S13**, while statistical details are reported in **Table S12**.
2. First, we have computed time-frequency analysis on the six main AAL ROIs that we reported in the revised version of the manuscript. Moreover, we have computed the time-frequency analysis on an additional ROI, the occipital superior lobe (both left and right hemisphere from AAL parcellation). As commonly shown by previous literature, this region is one of the main generators of alpha waves⁷⁷. Coherently, our analysis showed that the frequency band characterised by the strongest power in the occipital superior lobe was alpha. Conversely, the other ROIs were characterised by power spectra which did not primarily show alpha power. Interestingly, when contrasting the power spectrum of M and NT1 conditions in the occipital superior lobe, the results were similar to the ones for the other six ROIs. A similar phenomenon occurred for the time-frequency contrasts computed for the eight MEG channels, highlighting a particularly stronger power in alpha and beta bands for N vs M which was spread across the whole brain/scalp. Finally, it is worth noting that when conducting the time-frequency analysis on the six main AAL ROIs, the outcomes exhibited more distinct characteristics compared to the results obtained from the six ROIs that we functionally derived from the data. This divergence can be attributed to the more focalised spatial specificity of the AAL ROIs, encompassing smaller brain regions, ultimately yielding more interpretable and precise findings.

These additional analyses show that the time-frequency decomposition was not affected by the source reconstruction algorithms. Moreover, they suggest that the differences between experimental conditions in term of power in different frequency bands were rather similar across the whole brain.

We have reported this information in the revised manuscript.

Results (line 533):

“Time-frequency analysis for induced responses

We computed time-frequency analysis using complex Morlet wavelet transform (from 1 to 60 Hz with 1-Hz intervals). This analysis was conducted for induced responses. First, we estimated independent time-frequency decompositions for each trial and each voxel of the six ROIs described above (LHG, RHG, LHP, RHP, ACC, MC) plus left and right occipital superior lobe. Then, the computed power spectra were averaged over voxels and over trials within each ROI. Finally, in line with the previous analyses, we calculated four contrasts (M versus NT1, M versus NT2, M versus NT3, and M versus NT4). Specifically, we computed a t-test for each frequency and time-point and corrected for multiple comparisons using 2D cluster-based MCS (MCS, $\alpha = .05$, MCS p-value = .001). As shown in **Figure 7**, results were similar across ROIs and displayed a generalised stronger power for N versus M between 2 and 20 Hz (corresponding to, approximately, theta, alpha, and beta bands), in the time window 1.0 – 3.0 seconds ($p < .001$). In addition, a few significant clusters showed a moderate yet significant increase of power for M versus N ($p < .001$) for frequencies higher than 30 Hz, which are normally referred to as gamma⁵⁰. Detailed statistical results about this procedure are extensively reported in **Table S11** and depicted in **Figure S12**”.

Discussion (line 682):

“Finally, the induced-response analysis showed that, after the end of the sequence, alpha and beta bands were stronger for the varied compared to the memorised sequences. Considering the modulation of alpha and beta activity in encoding and working memory tasks described in the scientific literature⁵¹, this finding may suggest that post-stimulus increase of alpha-beta power might be relevant to process novel information. In addition, this analysis revealed a weak yet significant increase in gamma power for the memorised compared to the varied sequences. This is coherent with previous studies which reported increased gamma power during recognition of target stimuli^{52, 53} and, more generally, a modulation of the brain oscillations associated with memory load and complex cognitive functions⁵⁴. Although this study provided initial insights on the time-frequency modulation occurring in the brain during long-term recognition of auditory sequences, further research employing MEG and additional

tools such as stereo-electroencephalography (SEEG) is needed to expand our experimental design and conduct cross-frequency coupling analysis, testing whether gamma-theta coupling is connected to long-term recognition of auditory sequences”.

Additional changes were made in the Methods section (line 1232).

In addition, please refer to **Figures 7** (AAL ROIs time-frequency), **S13** (MEG channels time-frequency) and **S17** (functional parcellation time-series) for a graphical depiction of the results.

- In the discussion, many interpretations are based on descriptive results, or on results that are not shown in the result section:

You are right.

In the revised manuscript, we have made sure to remove all the claims that were not precisely supported by inferential (or at least descriptive) statistics.

Moreover, in relation to the specific points that you mentioned, we have carried out the revisions described below.

- the claim of “a rapid hierarchical pathway progressing from the auditory cortex to the ventromedial prefrontal cortex” (p.18 l.364) and that there is “information flowing from auditory cortex to ventromedial prefrontal cortex and hippocampal regions” (p.21, l.455) should be based on directed connectivity measures.

As described in detail above, we have computed DCM to provide quantitative evidence of functional connectivity and hierarchy between the brain regions. As you could read below in response to your previous comments, we have refined our claims in the discussion accordingly.

- the claim that “the later responses in the ventromedial prefrontal cortex and hippocampus occurred only after the tone that disrupted the original sequences” (p.18 l.374) could be explicitly tested by grouping the responses to notes in M, grouping the first novel note in N, and looking at the contrast between them.

Thank you for this relevant comment. We acknowledge the importance of grounding our assertions in quantitative evidence. Thus, we have now addressed this concern in the revised manuscript. In response to your comment, we have omitted the original claim and, instead, presented two pertinent claims substantiated by quantitative evidence derived from inferential statistics.

They are reported as follows:

- 1) The responses in ventromedial prefrontal cortex, anterior cingulate gyrus and hippocampus occurring at approximately 250 ms after the varied tones were significantly stronger for the N condition where the variations were introduced compared to M.

This was tested by conducting t-tests for each time-point between M and the N condition where the variations were introduced. Then, these tests were cluster based MCS corrected for multiple comparisons. **Figures 4** and **S7_{a-g}** illustrate all the significant time-points emerging from the analyses, while complete statistical reports are available in **Tables S6** and **S7**.

- 2) Within NT1, NT2 and NT3, the responses in anterior and medial cingulate gyrus and hippocampus occurring approximately at 250 ms after the varied tones were significantly stronger for the first varied tones compared to all the other tones. Since NT4 comprised only one varied tone, it was obviously not possible to extend this analysis to this experimental condition. Notably, the responses in Heschl's gyrus occurring at approximately 150 ms after the varied tones were the same across all tones.

To test this, we computed the average neural activity within a ± 20 ms time window centred around the designated time index (150 ms for Heschl's gyri and 250 ms for the hippocampus and cingulate gyrus). This is depicted in **Figure 5** using red circles. Subsequently, we performed separate one-way analyses of variance (ANOVAs) for each

ROI and selected condition, with the sequential neural peaks as independent variable and the neural activity as the dependent variable. Since this approach led to a total of 18 ANOVAs (six ROIs for NT1, NT2, NT3), we applied FDR correction to account for multiple comparisons. Moreover, the Tukey-Kramer correction was applied to the post-hoc analysis computed for each ANOVA (**Table S10**).

The discussion of the revised manuscript has been modified accordingly, making sure that our claims are now backed by robust statistical evidence.

Claim 1)

Results (line 360):

“Then, we computed one t-test for each AAL ROI, each time-point and each combination of M versus Ns. Finally, we corrected for multiple comparisons using 1D cluster-based MCS (MCS, $\alpha = .05$, MCS *p-value* = .001). Confirming the results from the 8-mm brain parcellation shown in **Figure 3**, this analysis returned several significant clusters of differential brain activity over time between M and Ns. These were primarily localised in auditory cortices (e.g. Heschl’s gyrus, superior temporal gyrus) hippocampus and para-hippocampal gyrus, inferior temporal cortex, anterior, medial and posterior cingulate gyrus, ventromedial prefrontal cortex”.

Discussion (line 638):

“In addition, while the hierarchical architecture involved in recognising both memorised and varied sequences remained largely unchanged, the temporal dynamics, strength, and polarity of the brain signal sharply differed. Here, when the upcoming sound of the sequence was matched with the predicted sound based on the previously stored memory trace, first the auditory cortex and then hippocampus, anterior and medial cingulate gyri responded with positive components. Conversely, when the upcoming sound was incoherent with the prediction made by the brain, a pathway of primarily negative and faster components emerged within the same brain network”.

Claim 2)

Results (line 428):

“Prediction error across brain regions of interest

After differentiating the brain activity of M versus N, we conducted an additional analysis to investigate the peak responses elicited by the varied tones within the N conditions. Regarding Heschl’s gyrus (left and right), the peaks corresponded to the late N100, occurring approximately at 150 ms after the onset of each varied tone. Regarding the hippocampus (left and right), anterior and medial cingulate gyrus, the peaks occurred approximately at 250 ms after the onset of each varied tone (these responses were negative for the hippocampus and anterior cingulate gyrus and positive for the medial cingulate gyrus). We computed the average neural activity within a ± 20 ms time window centred around the designated time index (depicted in **Figure 5** by red circles). Subsequently, we performed separate one-way analyses of variance (ANOVAs) for each ROI and selected condition, with the sequential neural peaks as independent variable and the neural activity as the dependent variable. Since this approach led to a total of 18 ANOVAs (six ROIs for NT1, NT2, NT3), we applied FDR correction to account for multiple comparisons. Moreover, the Tukey-Kramer correction was applied to the post-hoc analysis computed for each ANOVA (**Table S10**). Results revealed that the responses in Heschl’s gyri to subsequent varied tones were not significantly different: LHG_{NT1}: $F(3, 331) = 0.87, p = .51$; LHG_{NT2}: $F(2, 248) = 1.33, p = .37$; LHG_{NT3}: $F(1, 165) = 0.68, p = .51$; RHG_{NT1}: $F(3, 331) = 0.90, p = .51$; RHG_{NT2}: $F(2, 248) = 0.18, p = .83$; RHG_{NT3}: $F(1, 165) = 0.33, p = .60$. In contrast, for the hippocampus and cingulate gyrus, the responses to different varied tones were significantly different, highlighting the stronger activity following the initial tone which introduced the sequence variation: LHP_{NT1}: $F(3, 331) = 2.82, p = .05$; LHP_{NT2}: $F(2, 248) = 7.29, p = .002$; LHP_{NT3}: $F(1, 165) = 7.86, p = .01$; RHP_{NT1}: $F(3, 331) = 7.40, p = 2.9e-04$; RHP_{NT2}: $F(2, 248) = 14.33, p = 1.3e-06$; RHP_{NT3}: $F(1, 165) = 27.16, p = 5.56e-07$; ACC_{NT1}: $F(3, 331) = 4.98, p = .005$; ACC_{NT2}: $F(2, 248) = 11.54, p = 1.6e-05$; ACC_{NT3}: $F(1, 165) = 17.55, p = 2.05e-04$; MC_{NT1}: $F(3, 331) = 3.21, p = .02$; MC_{NT2}: $F(2, 248) = 8.40, p = 2.9e-04$; MC_{NT3}: $F(1, 165) = 6.24, p = .02$ ”.

Discussion (line 587):

“Conversely, in our study we revealed an initial recruitment of the auditory cortex (100 – 150 ms) and then of the anterior and medial cingulate gyrus, ventromedial prefrontal cortex and bilateral hippocampus (250 – 500 ms). Notably, our results showed that Heschl’s gyrus discriminated memorised melodies versus systematic variations but did not distinguish the strength of the errors (i.e., the response was not significantly different across varied tones). Conversely, the prediction error observed

in the bilateral hippocampus, anterior and medial cingulate gyrus was significantly stronger in response to the tone that introduced the variations compared to all other tones. Given the different brain responses observed in our study in comparison to previous literature on automatic prediction error, we argue that the brain signature underlying the awareness of the variation might be represented by the responses recorded in the anterior cingulate gyrus, ventromedial prefrontal cortex and hippocampus and their specific temporal dynamics”.

- the claim that “The strength of the responses was progressively greater depending on the tone used to introduce the variation” (p.18, l.378) also has to be empirically tested.

Following your comment, we have removed this claim from the manuscript as it was supported only by descriptive rather than inferential statistics.

Instead, as described in response to your previous comment, we have emphasised a related yet distinct phenomenon that our study highlighted, and which is supported by inferential analysis.

Indeed, here, we investigated the prediction error in the context of conscious long-term memory recognition, revealing an initial recruitment of the auditory cortex (100 – 150 ms) and then of the anterior and medial cingulate gyrus, ventromedial prefrontal cortex and bilateral hippocampus (250 – 500 ms). Notably, our results showed that Heschl’s gyrus discriminated memorised melodies versus systematic variations but did not distinguish the strength of the errors (i.e., the response was not significantly different across varied tones). Conversely, the prediction error observed in the bilateral hippocampus, anterior cingulate gyrus and medial cingulate gyrus was significantly stronger in response to the tone that introduced the variations compared to all other tones. Given the different brain responses observed in our study in comparison to previous literature on automatic prediction error, we argue that the brain signature underlying the awareness of the variation might be represented by the responses recorded in the anterior cingulate gyrus, ventromedial prefrontal cortex and hippocampus and their specific temporal dynamics.

We have reported the results and updated discussion in the revised manuscript (e.g. line 587), as you can see from the response to your previous comment.

- the claim that “... we observed a distinct, faster response to each tone of the sequence. The frequency of these responses was of approximately 1 – 1.5 Hz” (p.19, l.403)” is also not supported by any analysis, even descriptive.

You are right and thus we have removed this claim.

Minor Comments:

- There is no information about the musical background of the participants. If this data is not available for some reason, it would be preferable to state it clearly rather than omitting it.

Initially, our focus was not on exploring the connection between musical background and neural data. Our primary objective was to employ music as a tool to probe the brain's responses, with less emphasis on understanding the relationship between musical training and neural signal.

However, your observation raised a pertinent point. Consequently, we conducted additional analyses to investigate the link between musical training and brain responses during the recognition of previously memorised and novel sequences.

During data collection, we assessed participants' musical backgrounds using the Goldsmith Musical Sophistication Index (GOLD-MSI)⁷⁸, a widely used standardized test that evaluates various musical abilities, including the Musical Training subscale. This subscale encompasses data related to music education, formal training, average time spent playing musical instruments at different life stages, and more.

This measure was correlated with the minimum and maximum peaks of the neural activity for tones 2-5 (the ones reported in the previous analyses and illustrated in **Figures 2 and 3**). The output of the correlations was corrected for multiple comparisons by using the FDR correction.

The results of these analyses returned several significant relationships, highlighting that musical training was associated with enhanced brain activity during the recognition of previously memorised and novel sequences. This was especially evident for the neural signals in the right hemisphere indexing the prediction error caused by the tones which introduced the variations of the original sequences.

The results of this analysis are reported in the revised manuscript.

Results (line 1033):

“Furthermore, the brain activity recorded by the magnetometers for each peak and hemisphere was correlated with the measure of participants’ musical training provided by the Goldsmith Musical Sophistication Index (GOLD-MSI) ⁷⁸ and corrected for multiple comparisons using FDR.

Tone #	Tone 2		Tone 3		Tone 4		Tone 5	
Peak	Max	Min	Max	Min	Max	Min	Max	Min
Magnetometers in the left hemisphere								
Time (s)	0.708 (M)	0.612 (NT1)	1.092 (M)	0.992 (NT2)	1.452 (M)	1.340 (NT3)	1.804 (M)	1.736 (NT4)
Musical expertise (FDR-corrected)	$r = .093$ $p = .473$	$r = -.218$ $p = .084$	$r = .239$ $p = .066$	$r = -.186$ $p = .132$	$r = .286$ $p = .040$	$r = -.261$ $p = .050$	$r = .344$ $p = .014$	$r = -.034$ $p = .768$
Contrasts in MEG source space								
Conditions	M versus NT1	M versus NT1	M versus NT2	M versus NT2	M versus NT3	M versus NT3	M versus NT4	M versus NT4
Max t-val	5.79	n.s	5.45	-6.09	5.81	-6.56	5.1	-6.03
FDR-adjusted threshold p-value	.003	n.s	.009	.008	.01	.01	.01	.02
MNI coordinates	-6 -22 64	n.s	-6 -30 40	2 34 0	-6 -38 32	2 18 -16	26 -30 -8	-6 26 -16
AAL region	Supp Motor Ar	n.s	Cingulu m Mid	Cingulum Ant	Cingulum Mid	Subgenua l	Hippocampus	Front Med Orb
Magnetometers in the right hemisphere								
Time (s)	0.620 (NT1)	0.832 (M)	0.988 (NT2)	1.088 (M)	1.352 (NT3)	1.456 (M)	1.732 (NT4)	1.920 (M)
Musical expertise (FDR-corrected)	$r = .242$ $p = .061$	$r = .051$ $p = .819$	$r = .428$ $p < .001$	$r = -.034$ $p = .819$	$r = .459$ $p < .001$	$r = -.050$ $p = .819$	$r = .257$ $p = .050$	$r = -.026$ $p = .819$
Contrasts in MEG source space								
Conditions	M versus NT1	M versus NT1	M versus NT2	M versus NT2	M versus NT3	M versus NT3	M versus NT4	M versus NT4
Max t-val	n.s	4.41	-6.14	5.50	-7.13	5.61	-6.17	5.24

FDR-adjusted threshold p-value	n.s	.001	.008	.01	.02	.01	.02	.01
MNI coordinates	n.s	58 -14 -16	2 34 -8	-6 -30 40	2 18 -16	-6 -38 32	-6 26 -16	10 -46 48
AAL region	n.s	Temporal Mid	Cingulum Ant	Cingulum Mid	Subgenual I	Cingulum Mid	Front Med Orb	Precuneus

Table 2. Neural activity peaks recorded by magnetometers, musical expertise and contrasts in MEG 8-mm source space

The table shows the time index of the maximum and minimum peaks recorded by the magnetometers (refer to **Figure 3**) for each tone (excluding the 1st one). Additionally, the table indicates the condition for which the strongest peaks were observed (in parenthesis). The table also illustrates the correlations between the absolute value of the neural peaks and the participants' musical training ($r = \text{Pearson's correlation coefficient}$, p -values are adjusted by the false discovery rate [FDR] correction for multiple comparisons). Finally, the table provides details of the MEG source contrasts in 8-mm space. This includes the conditions which were contrasted, t -value, FDR-adjusted p -value, Montreal Neurological Institute (MNI) coordinates, and automated anatomical labelling (AAL) region of the brain voxel which showed the strongest difference between the experimental conditions.

Our results align with previous literature, which has underscored the correlation between musical expertise and neural responses to musical stimuli. Notably, our findings extend beyond the existing literature, offering novel evidence of the relationship between musical training and the neural mechanisms governing the conscious recognition of previously memorised and novel musical sequences.

Thank you for raising this point, which has contributed to increase the novelty of our work. These findings have been discussed in detail in the revised manuscript (line 661): "Our findings revealed a positive overall correlation between musical training and neural activity associated with the recognition of memorised and varied musical sequences. This correlation was particularly evident with regards to the neural responses to the sounds that introduced variations. This aligns with previous research indicating a connection between musical expertise and altered brain characteristics, including enhanced neural responses (e.g. enhanced auditory automatic prediction error) ^{2, 14, 74} and structural changes ⁷⁵. In this study, what distinguishes our findings was the extension of this relationship to the domain of long-term recognition of musical sequences and the underlying brain mechanisms".

Additional changes were made to the Methods section (line 1033).

- For the sake of clarity, I would suggest reducing the size of the figure's legends by removing all the interpretation of the results that they contain. There are also some mistakes to correct (for instance: wrong colors described in Fig. 3).

We have reduced the size of the captions, according to your suggestion and to the editorial guidelines provided by *Nature Communications* and removed the interpretation of the results. Likewise, we corrected the typos.

- Before showing the MVPA results in Fig. 2, and to help the reader getting a better idea of what the data look like, I would suggest to show some simple univariate analysis, or at least in the supplementary figures (e.g. a topographic map of the general auditory response to all conditions, or simple contrasts between conditions at the sensor level).

Following your recommendation, we have computed additional univariate analyses. This was done in two ways:

1. On a selected cluster of MEG channels (the ones that provided the strongest contribution to the MVPA results) (Please, refer to **Figure 2** and **Table S3** for the results).

2. On every single MEG channel (Please, refer to **Figure S5_{a-z2}** and **Table S4** for the results).

Moreover, we have provided in supplementary material, topographic maps of the main auditory responses at the MEG channels level (**Figure S6**).

These additional analyses and plots provided full disclosure of our data and clarify the extent of the neural signal and the differences between experimental conditions.

However, in the manuscript, we start by showing the MVPA analysis. The rationale for that is that MVPA uses all MEG channels in a multivariate manner, and it does not make any assumption on specific channels of interest. For this reason, we always prefer to have it as the first analysis of our works.

- The temporal generalization of the decoding analysis is interesting but not mentioned in the discussion. For an example of interpretation, see King & Dehaene (2014).

Following your comments and the excellent reference you recommended, we have discussed this finding (line 651):

“An additional finding pertains to our utilisation of multivariate pattern analysis and temporal generalisation techniques, as described by King and Dehaene⁷³. Here, we revealed that when decoding memorised from novel sequences, the algorithm could differentiate the conditions starting from the first tone that was varied in the novel sequences. This learning could then be generalised by the algorithm with significant (yet reduced) accuracy for all the subsequent tones. This pattern was consistently observed across all categories of variations, and it implies that the differential processes between the two conditions remained consistent over time, indicating that the brain not only identifies the initial varied tone but also monitors the entire auditory sequence. This monitoring might have served the purpose of assessing whether the varied tone represented a transient mistake or the beginning of a proper varied sequence”.

- In the section “Neural sources of the differential brain activity between M and N”, the table 2 is hard to read. Plotting the clusters on a 3D brain (grouped by time window or by contrast) may help.

We have heavily reshaped that section, including the table. Now, **Table 2** and **Figure 3** provide much clearer representation of the results as well as the relationship between the results for MEG sensors and MEG sources.

- Although it is good to be enthusiastic and positive about one’s results, I would suggest avoiding wording like “This study represents the acme of our recent investigations”, and any sentence written to highlight your work without reporting facts or interpretations. You are right; we have gone through the manuscript with great care, refining the language and eliminating any interpretations that were not rigorous enough.

Bibliography:

- King, J. R., & Dehaene, S. (2014). Characterizing the dynamics of mental representations: the temporal generalization method. *Trends in cognitive sciences*, 18(4), 203-210.
- Rocchi, F., Oya, H., Balezeau, F., Billig, A. J., Kocsis, Z., Jenison, R. L., ... & Petkov, C. I. (2021). Common fronto-temporal effective connectivity in humans and monkeys. *Neuron*, 109(5), 852-868.

We have discussed these excellent references in the revised manuscript.

For instance, line 651:

“An additional finding pertains to our utilisation of multivariate pattern analysis and temporal generalisation techniques, as described by King and Dehaene⁷³. Here, we revealed that when decoding memorised from novel sequences, the algorithm could differentiate the conditions starting from the first tone that was varied in the novel sequences. This learning could then be generalised by the algorithm with significant (yet reduced) accuracy for all the subsequent tones. This pattern was consistently observed across all categories of variations, and it implies that the differential processes between the

two conditions remained consistent over time, indicating that the brain not only identifies the initial varied tone but also monitors the entire auditory sequence. This monitoring might have served the purpose of assessing whether the varied tone represented a transient mistake or the beginning of a proper varied sequence”.

Line 625:

“These findings align with previous research demonstrating the flow of information from sensory cortices to the medial temporal lobe, associative areas, and prefrontal regions of the brain ^{23, 52}. Moreover, our results are coherent with the several studies which used PCT to investigate perceptual ⁵³ and automatic memory mechanisms in the brain ²², as reviewed by Spratling ⁴⁵”.

Line 65:

“Expanding upon these investigations, Rocchi and colleagues provided evidence of hierarchical organisation in the auditory system using direct electrical brain stimulation. They showed effective connectivity from the auditory cortex to regions in the medial temporal lobe and prefrontal cortex, including the hippocampus and ventro-lateral prefrontal cortex ⁴³”.

References

1. Bonetti, L., *et al.* Brain predictive coding processes are associated to COMT gene Val158Met polymorphism. *NeuroImage* **233**, 117954 (2021).
2. Bonetti, L., *et al.* Whole-brain computation of cognitive versus acoustic errors in music: A mismatch negativity study. *Neuroimage: Reports* **2**, 100145 (2022).
3. Bonetti, L., *et al.* Auditory sensory memory and working memory skills: association between frontal MMN and performance scores. *Brain research* **1700**, 86-98 (2018).
4. Bonetti, L., Haumann, N., Vuust, P., Kliuchko, M. & Brattico, E. Risk of depression enhances auditory Pitch discrimination in the brain as indexed by the mismatch negativity. *Clinical Neurophysiology* **128**, 1923-1936 (2017).
5. Brattico, E., Winkler, I., Naatanen, R., Paavilainen, P. & Tervaniemi, M. Simultaneous storage of two complex temporal sound patterns in auditory sensory memory. *Neuroreport* **13**, 1747-1751 (2002).
6. Conley, E.M., Michalewski, H.J. & Starr, A. The N100 auditory cortical evoked potential indexes scanning of auditory short-term memory. *Clin Neurophysiol* **110**, 2086-2093 (1999).
7. Naatanen, R. & Picton, T. The N1 wave of the human electric and magnetic response to sound: a review and an analysis of the component structure. *Psychophysiology* **24**, 375-425 (1987).
8. Koelsch, S. Towards a neural basis of processing musical semantics. *Phys Life Rev* **8**, 89-105 (2011).
9. Koelsch, S. Music-syntactic processing and auditory memory: Similarities and differences between ERAN and MMN. *Psychophysiology* **46**, 179-190 (2009).
10. Brattico, E., *et al.* Neural discrimination of nonprototypical chords in music experts and laymen: an MEG study. *J Cogn Neurosci* **21**, 2230-2244 (2009).
11. Näätänen, R., Paavilainen, P., Rinne, T. & Alho, K. The mismatch negativity (MMN) in basic research of central auditory processing: a review. *Clinical neurophysiology* **118**, 2544-2590 (2007).
12. Friston, K. Predictive coding, precision and synchrony. *Cognitive neuroscience* **3**, 238-239 (2012).
13. Koelsch, S., Vuust, P. & Friston, K. Predictive processes and the peculiar case of music. *Trends in cognitive sciences* **23**, 63-77 (2019).
14. Vuust, P., Heggli, O.A., Friston, K.J. & Kringelbach, M.L. Music in the brain. *Nature Reviews Neuroscience* **23**, 287-305 (2022).

15. Spratling, M.W. Predictive coding as a model of cognition. *Cognitive processing* **17**, 279-305 (2016).
16. Bonetti, L., *et al.* Brain recognition of previously learned versus novel temporal sequences: a differential simultaneous processing. *Cerebral Cortex*, bhac439 (2022).
17. Fernandez-Rubio, G., *et al.* Magnetoencephalography recordings reveal the spatiotemporal dynamics of recognition memory for complex versus simple auditory sequences. *Commun Biol* **5**, 1272 (2022).
18. Fernández-Rubio, G., Carlomagno, F., Vuust, P., Kringelbach, M.L. & Bonetti, L. Associations between abstract working memory abilities and brain activity underlying long-term recognition of auditory sequences. *PNAS Nexus* **1**, pgac216 (2022).
19. Hoegholt, N.F., *et al.* A magnetoencephalography study of first-time mothers listening to infant cries. *Cereb Cortex* (2022).
20. Tzourio-Mazoyer, N., *et al.* Automated anatomical labeling of activations in SPM using a macroscopic anatomical parcellation of the MNI MRI single-subject brain. *Neuroimage* **15**, 273-289 (2002).
21. Bonetti, *et al.* Brain predictive coding processes are associated to COMT gene Val158Met polymorphism. *Neuroimage* **233**, 117954 (2021).
22. Bonetti, L., *et al.* Spatiotemporal brain dynamics during recognition of the music of Johann Sebastian Bach. *Biorxiv* (2020).
23. Bruzzone, S.E.P., *et al.* Dissociated brain functional connectivity of fast versus slow frequencies underlying individual differences in fluid intelligence: a DTI and MEG study. *Sci Rep* **12**, 4746 (2022).
24. Cabral, J., *et al.* Exploring mechanisms of spontaneous functional connectivity in MEG: how delayed network interactions lead to structured amplitude envelopes of band-pass filtered oscillations. *Neuroimage* **90**, 423-435 (2014).
25. Colclough, G.L., Brookes, M.J., Smith, S.M. & Woolrich, M.W. A symmetric multivariate leakage correction for MEG connectomes. *Neuroimage* **117**, 439-448 (2015).
26. Brookes, M.J., *et al.* Beamformer reconstruction of correlated sources using a modified source model. *Neuroimage* **34**, 1454-1465 (2007).
27. Luckhoo, H.T., Brookes, M.J. & Woolrich, M.W. Multi-session statistics on beamformed MEG data. *Neuroimage* **95**, 330-335 (2014).

28. Hall, E.L., Woolrich, M.W., Thomaz, C.E., Morris, P.G. & Brookes, M.J. Using variance information in magnetoencephalography measures of functional connectivity. *Neuroimage* **67**, 203-212 (2013).
29. Van Veen, B.D., van Drongelen, W., Yuchtman, M. & Suzuki, A. Localization of brain electrical activity via linearly constrained minimum variance spatial filtering. *IEEE Trans Biomed Eng* **44**, 867-880 (1997).
30. Dietz, M.J., Nielsen, J.F., Roepstorff, A. & Garrido, M.I. Reduced effective connectivity between right parietal and inferior frontal cortex during audiospatial perception in neglect patients with a right-hemisphere lesion. *Hearing Research* **399**, 108052 (2021).
31. Garrido, M.I., *et al.* The functional anatomy of the MMN: a DCM study of the roving paradigm. *Neuroimage* **42**, 936-944 (2008).
32. Stephan, K.E., *et al.* Ten simple rules for dynamic causal modeling. *Neuroimage* **49**, 3099-3109 (2010).
33. David, O., *et al.* Dynamic causal modeling of evoked responses in EEG and MEG. *Neuroimage* **30**, 1255-1272 (2006).
34. Penny, W.D. Comparing dynamic causal models using AIC, BIC and free energy. *Neuroimage* **59**, 319-330 (2012).
35. Penny, W.D., *et al.* Comparing families of dynamic causal models. *PLoS Comput Biol* **6**, e1000709 (2010).
36. Merkley, T.L., Larson, M.J., Bigler, E.D., Good, D.A. & Perlstein, W.M. Structural and functional changes of the cingulate gyrus following traumatic brain injury: relation to attention and executive skills. *J Int Neuropsychol Soc* **19**, 899-910 (2013).
37. Kennerley, S.W., Walton, M.E., Behrens, T.E., Buckley, M.J. & Rushworth, M.F. Optimal decision making and the anterior cingulate cortex. *Nat Neurosci* **9**, 940-947 (2006).
38. Brattico, E. & Pearce, M. The neuroaesthetics of music. *Psychology of Aesthetics, Creativity, and the Arts* **7**, 48 (2013).
39. Zatorre, R.J., Belin, P. & Penhune, V.B. Structure and function of auditory cortex: music and speech. *Trends Cogn Sci* **6**, 37-46 (2002).
40. Hasselmo, M.E. A Handbook for Modeling Hippocampal Circuits. (Frontiers Research Foundation, 2011).
41. Stern, C.E., Sherman, S.J., Kirchhoff, B.A. & Hasselmo, M.E. Medial temporal and prefrontal contributions to working memory tasks with novel and familiar stimuli. *Hippocampus* **11**, 337-346 (2001).

42. Plakke, B. & Romanski, L.M. Auditory connections and functions of prefrontal cortex. *Frontiers in neuroscience* **8**, 199 (2014).
43. Rocchi, F., *et al.* Common fronto-temporal effective connectivity in humans and monkeys. *Neuron* **109**, 852-868 e858 (2021).
44. Denham, S.L. & Winkler, I. Predictive coding in auditory perception: challenges and unresolved questions. *Eur J Neurosci* **51**, 1151-1160 (2020).
45. Garrido, M.I., Kilner, J.M., Stephan, K.E. & Friston, K.J. The mismatch negativity: a review of underlying mechanisms. *Clin Neurophysiol* **120**, 453-463 (2009).
46. Denison, R.N., Piazza, E.A. & Silver, M.A. Predictive Context Influences Perceptual Selection during Binocular Rivalry. *Front Hum Neurosci* **5**, 166 (2011).
47. Jehee, J.F. & Ballard, D.H. Predictive feedback can account for biphasic responses in the lateral geniculate nucleus. *PLoS computational biology* **5**, e1000373 (2009).
48. Spratling, M.W. Predictive coding as a model of response properties in cortical area V1. *Journal of neuroscience* **30**, 3531-3543 (2010).
49. Spratling, M.W. A single functional model of drivers and modulators in cortex. *J Comput Neurosci* **36**, 97-118 (2014).
50. Fitzgibbon, S.P., Pope, K.J., Mackenzie, L., Clark, C.R. & Willoughby, J.O. Cognitive tasks augment gamma EEG power. *Clin Neurophysiol* **115**, 1802-1809 (2004).
51. Proskovec, A.L., Wiesman, A.I., Heinrichs-Graham, E. & Wilson, T.W. Load effects on spatial working memory performance are linked to distributed alpha and beta oscillations. *Hum Brain Mapp* **40**, 3682-3689 (2019).
52. Lenz, D., Schadow, J., Thaerig, S., Busch, N.A. & Herrmann, C.S. What's that sound? Matches with auditory long-term memory induce gamma activity in human EEG. *International Journal of Psychophysiology* **64**, 31-38 (2007).
53. Slobounov, S., Tutwiler, R., Slobounova, E., Rearick, M. & Ray, W. Human oscillatory brain activity within gamma band (30–50 Hz) induced by visual recognition of non-stable postures. *Cognitive Brain Research* **9**, 177-192 (2000).
54. Sauseng, P., Griesmayr, B., Freunberger, R. & Klimesch, W. Control mechanisms in working memory: a possible function of EEG theta oscillations. *Neurosci Biobehav Rev* **34**, 1015-1022 (2010).
55. Gross, J., *et al.* Good practice for conducting and reporting MEG research. *Neuroimage* **65**, 349-363 (2013).

56. Hansen, P., Kringelbach, M. & Salmelin, R. *MEG: An introduction to methods* (Oxford university press, 2010).
57. Albouy, P., Weiss, A., Baillet, S. & Zatorre, R.J. Selective entrainment of theta oscillations in the dorsal stream causally enhances auditory working memory performance. *Neuron* **94**, 193-206. e195 (2017).
58. Mantini, D., *et al.* A signal-processing pipeline for magnetoencephalography resting-state networks. *Brain Connect* **1**, 49-59 (2011).
59. Deco, G., Vidaurre, D. & Kringelbach, M.L. Revisiting the global workspace orchestrating the hierarchical organization of the human brain. *Nat Hum Behav* **5**, 497-511 (2021).
60. Mesulam, M.M. From sensation to cognition. *Brain* **121** (Pt 6), 1013-1052 (1998).
61. Preti, M.G., Bolton, T.A. & Van De Ville, D. The dynamic functional connectome: State-of-the-art and perspectives. *Neuroimage* **160**, 41-54 (2017).
62. Scalabrini, A., Mucci, C. & Northoff, G. The nested hierarchy of self and its trauma: In search for a synchronic dynamic and topographical re-organization. *Front Hum Neurosci* **16**, 980353 (2022).
63. Kringelbach, M.L., *et al.* Dynamic coupling of whole-brain neuronal and neurotransmitter systems. *Proc Natl Acad Sci U S A* **117**, 9566-9576 (2020).
64. Ip, I.B. & Bridge, H. Investigating the neurochemistry of the human visual system using magnetic resonance spectroscopy. *Brain Struct Funct* **227**, 1491-1505 (2022).
65. King, J.-R. & Wyart, V. The human brain encodes a chronicle of visual events at each instant of time through the multiplexing of traveling waves. *Journal of Neuroscience* **41**, 7224-7233 (2021).
66. Zatorre, R.J. Sound analysis in auditory cortex. *Trends Neurosci* **26**, 229-230 (2003).
67. Moore, J.K. Organization of the human superior olivary complex. *Microsc Res Tech* **51**, 403-412 (2000).
68. Pandya, D.N. Anatomy of the auditory cortex. *Rev Neurol (Paris)* **151**, 486-494 (1995).
69. Peretz, I. & Zatorre, R.J. *The cognitive neuroscience of music* (OUP Oxford, 2003).
70. Bonetti, L., *et al.* Rapid encoding of musical tones discovered in whole-brain connectivity. *NeuroImage* **245**, 118735 (2021).
71. Zatorre, R.J., Evans, A.C. & Meyer, E. Neural mechanisms underlying melodic perception and memory for pitch. *J Neurosci* **14**, 1908-1919 (1994).

72. Zatorre, R.J., Perry, D.W., Beckett, C.A., Westbury, C.F. & Evans, A.C. Functional anatomy of musical processing in listeners with absolute pitch and relative pitch. *Proc Natl Acad Sci U S A* **95**, 3172-3177 (1998).
73. King, J.R. & Dehaene, S. Characterizing the dynamics of mental representations: the temporal generalization method. *Trends Cogn Sci* **18**, 203-210 (2014).
74. Vuust, P., Brattico, E., Seppänen, M., Naatanen, R. & Tervaniemi, M. The sound of music: differentiating musicians using a fast, musical multi-feature mismatch negativity paradigm. *Neuropsychologia* **50**, 1432-1443 (2012).
75. Criscuolo, A., Pando-Naude, V., Bonetti, L., Vuust, P. & Brattico, E. An ALE meta-analytic review of musical expertise. *Sci Rep* **12**, 11726 (2022).
76. Afnan, J., *et al.* Validating MEG source imaging of resting state oscillatory patterns with an intracranial EEG atlas. *Neuroimage* **274**, 120158 (2023).
77. Shevelev, I.A., Kosteljanetz, N.B., Kamenkovich, V.M. & Sharaev, G.A. EEG alpha-wave in the visual cortex: check of the hypothesis of the scanning process. *Int J Psychophysiol* **11**, 195-201 (1991).
78. Müllensiefen, D., Gingras, B., Stewart, L. & Musil, J.J. Goldsmiths Musical Sophistication Index (Gold-MSI) v1. 0: Technical Report and Documentation Revision 0.3. (2013).

Reviewer #2 (Remarks to the Author):

The authors have responded extremely well to the comments that I raised and I recommend acceptance

Reviewer #3 (Remarks to the Author):

In their manuscript, Bonetti and colleagues highlight the brain network involved in recognizing memorized versus novel melodies.

The authors employed whole-brain decoding analyzes to demonstrate the overall impact of the task and utilized ROI-based dynamic causal modeling to reveal information flow between 3 regions. Univariate analyses corroborated that novel (surprising) notes elicit heightened activation in most regions, aligning with predictive coding theory expectations. These findings contribute to the predictive coding literature in music listening, expanding beyond 'low-order' automatic predictions to reveal the neural correlates of conscious error monitoring within a memory task context.

Firstly, I'd like to emphasize that the manuscript has undergone drastic changes since the initial version (refer to the text, mostly red in the revised version). While the manuscript has seen considerable improvement, for reviewers it was like another first round of review, involving substantial work.

It's important to also note that we lack expertise in DCM and, therefore, cannot provide a detailed review of this aspect of the manuscript.

Comments:

The manuscript has been significantly enhanced. The introduction aptly frames hypotheses within existing literature, and the discussion provides detailed and well-reasoned interpretations. Descriptive claims without supporting data have been addressed. Figures are now corrected. Overall, this paper offers valuable results for the music cognition and predictive coding community. We only have relatively minor comments for this revised manuscript:

- Please baseline the time-frequency panels to get rid of the 1/f pattern that currently dominate them.
- It is unclear how the t-test mentioned in the caption of figure 1d was performed. It may help to mention in the caption that t-tests were done for each time-point, and/or to show the significance bars from figure 2.
- In Figure 2b, it is unclear what the 2 topographies correspond to. The caption could be more precise.
- The axes & labels in Figure 5 are very small and difficult to read.

REVIEWERS' COMMENTS

Reviewer #2 (Remarks to the Author):

The authors have responded extremely well to the comments that I raised and I recommend acceptance

Thank you very much for your kind words.

Reviewer #3 (Remarks to the Author):

In their manuscript, Bonetti and colleagues highlight the brain network involved in recognizing memorized versus novel melodies.

The authors employed whole-brain decoding analyzes to demonstrate the overall impact of the task and utilized ROI-based dynamic causal modeling to reveal information flow between 3 regions. Univariate analyses corroborated that novel (surprising) notes elicit heightened activation in most regions, aligning with predictive coding theory expectations. These findings contribute to the predictive coding literature in music listening, expanding beyond 'low-order' automatic predictions to reveal the neural correlates of conscious error monitoring within a memory task context.

Firstly, I'd like to emphasize that the manuscript has undergone drastic changes since the initial version (refer to the text, mostly red in the revised version). While the manuscript has seen considerable improvement, for reviewers it was like another first round of review, involving substantial work.

It's important to also note that we lack expertise in DCM and, therefore, cannot provide a detailed review of this aspect of the manuscript.

Comments:

The manuscript has been significantly enhanced. The introduction aptly frames hypotheses within existing literature, and the discussion provides detailed and well-reasoned interpretations. Descriptive claims without supporting data have been addressed. Figures are now corrected. Overall, this paper offers valuable results for the music cognition and predictive coding community.

Thank you so much for your very thorough comments which have helped us to improve the manuscript even more.

We only have relatively minor comments for this revised manuscript:

- Please baseline the time-frequency panels to get rid of the 1/f pattern that currently dominate them.

Following your suggestion, we have now baseline corrected the time-frequency panels. After a careful evaluation of the available methods, we have opted for subtracting, for each frequency, the average power in the baseline interval from the power at all time points. This method returns the absolute change in power with respect to the baseline interval and represents one of the most popular methods, providing interpretable results (see, for instance, ¹⁻³). As expected, this baseline correction does not alter our results, but, as you predicted, slightly attenuates the 1/f pattern. Moreover, it better highlights the absolute change in power from the baseline, making our results even clearer. Once again, thank you for your comment.

We have reported this in the revised manuscript.

Results (line 398):

“First, we estimated independent time-frequency decompositions for each trial and each voxel of the six ROIs described above (LHG, RHG, LHP, RHP, ACC, MC) plus left and right occipital superior lobe for comparison purposes. Here, baseline correction was applied by subtracting, for each frequency, the average power in the baseline interval from the power at all time points ¹”.

Methods (line 1091):

“This analysis was conducted for induced responses, independently for the six ROIs previously described and for the four contrasts considered in this study (i.e., M versus NT1, M versus NT2, M versus NT3, M versus NT4). Specifically, the time-frequency decomposition was done independently for each trial, brain voxel, and participant. Moreover, baseline correction was applied by subtracting, for each frequency, the average power in the baseline interval from the power at all time points ¹⁻³”.

Moreover, we have updated the time-frequency Figures by replacing the previous time-frequency plots with the baseline corrected ones.

- It is unclear how the t-test mentioned in the caption of figure 1d was performed. It may help to mention in the caption that t-tests were done for each time-point, and/or to show the significance bars from figure 2.

We have specified in the caption that the t-tests were computed independently for each time-point (line 1438):

“**d** – Univariate analyses on the MEG channels that primarily contributed to the decoding algorithm were performed using t-tests, independently for each time-point, to compare the M condition with each of the N conditions (i.e. M versus NT1, M versus NT2, etc.)”.

- In Figure 2b, it is unclear what the 2 topographies correspond to. The caption could be more precise.

In the caption, we have better clarified what are the activation patterns depicted in the topographic maps.

“**b** – Activation patterns estimated by the decoding weights (without unit of measure). They represent the relative contribution of the MEG channels to the decoding algorithm when it successfully distinguished the two experimental conditions from the neural data (as indicated in **a**)”.

- The axes & labels in Figure 5 are very small and difficult to read.

We have increased the size of axes and labels in **Figure 5**.

1. Gyurkovics, M., Clements, G.M., Low, K.A., Fabiani, M. & Gratton, G. The impact of 1/f activity and baseline correction on the results and interpretation of time-frequency analyses of EEG/MEG data: A cautionary tale. *Neuroimage* **237**, 118192 (2021).
2. Herrmann, C.S., Rach, S., Voskuhl, J. & Struber, D. Time-frequency analysis of event-related potentials: a brief tutorial. *Brain Topogr* **27**, 438-450 (2014).
3. Hu, L., Xiao, P., Zhang, Z.G., Mouraux, A. & Iannetti, G.D. Single-trial time-frequency analysis of electrocortical signals: baseline correction and beyond. *Neuroimage* **84**, 876-887 (2014).